# Recovering Simultaneously Structured Data via Non-Convex Iteratively Reweighted Least Squares

**Christian Kümmerle**
Department of Computer Science
University of North Carolina at Charlotte
Charlotte, NC 28223, USA
kuemmerle@uncc.edu

**Johannes Maly**
Department of Mathematics
Ludwig-Maximilians-Universität München
80799 Munich, Germany and
Munich Center for Machine Learning (MCML)
maly@math.lmu.de

## Abstract

We propose a new algorithm for the problem of recovering data that adheres to multiple, heterogeneous low-dimensional structures from linear observations. Focusing on data matrices that are simultaneously row-sparse and low-rank, we propose and analyze an iteratively reweighted least squares (IRLS) algorithm that is able to leverage both structures. In particular, it optimizes a combination of non-convex surrogates for row-sparsity and rank, a balancing of which is built into the algorithm. We prove locally quadratic convergence of the iterates to a simultaneously structured data matrix in a regime of minimal sample complexity (up to constants and a logarithmic factor), which is known to be impossible for a combination of convex surrogates. In experiments, we show that the IRLS method exhibits favorable empirical convergence, identifying simultaneously row-sparse and low-rank matrices from fewer measurements than state-of-the-art methods.

## 1 Introduction

Reconstructing an image from (noisy) linear observations is maybe the most relevant inverse problem for modern image processing and appears in various applications like medical imaging and astronomy [7]. If the latent image is $n$-dimensional, for $n \in \mathbb{N}$, it is well-known that $\Omega(n)$ observations are required for robust identification in general. In practice, imaging problems are however often ill-posed, i.e., the number of observations is smaller than $n$ or the operator creating the observations is defective [88, 93]. In such situations, the fundamental lower bound of $\Omega(n)$ can be relaxed by leveraging structural priors of the latent image in the reconstruction process.

Of the various priors that are used for solving ill-posed inverse problems in the literature, sparsity[1] and low-rankness are most prevalent. This prominent role can be explained with their competitive performance in imaging tasks and the rigorous mathematical analysis they allow [10, 68]. For instance, consider the recovery of an $n_1 \times n_2$-dimensional image $\mathbf{X}_\star \in \mathbb{R}^{n_1 \times n_2}$ from linear observations

$$\mathbf{y} = \mathcal{A}(\mathbf{X}_\star) + \boldsymbol{\eta} \in \mathbb{R}^m, \tag{1}$$

where $\mathcal{A} \colon \mathbb{R}^{n_1 \times n_2} \to \mathbb{R}^m$ is a linear operator modeling the impulse response of the sensing device and $\boldsymbol{\eta} \in \mathbb{R}^m$ models additive noise. Whereas this problem is ill-posed for $m < n_1 n_2$, it has been established [38, 15, 78] that it becomes well-posed if $\mathbf{X}_\star$ is sparse or of low rank. The aforementioned works prove that $m = \Omega(s_1 s_2)$ observations suffice for robust reconstruction if $\mathbf{X}_\star$ is $s_1$-row-sparse

---

[1]A vector $\mathbf{x} \in \mathbb{R}^n$ is called $s$-sparse if $\mathbf{x}$ has at most $s$ non-zero entries. For a matrix $\mathbf{X} \in \mathbb{R}^{n_1 \times n_2}$ there are various ways to count the level of sparsity. In this work, we use the most common definition and call $\mathbf{X}$ $s$-row-sparse (resp. -column-sparse) if at most $s$ rows (resp. columns) are non-zero.

37th Conference on Neural Information Processing Systems (NeurIPS 2023).

and $s_2$-column-sparse, and that $m = \Omega(r(n_1 + n_2))$ observations suffice if $\mathbf{X}_\star$ is a rank-$r$ matrix. These bounds, which relax the general lower bound of $m = \Omega(n_1 n_2)$, agree with the degrees of freedom of sparse and low-rank matrices, respectively.

A number of computationally challenging problems in signal processing and machine learning can be formulated as instances of (1) with $\mathbf{X}_\star$ being *simultaneously structured*, i.e., $\mathbf{X}_\star$ is both of rank $r$ and $s_1$-row-sparse/$s_2$-column-sparse. Examples encompass sparse phase retrieval [51, 46, 11, 47, 85], sparse blind deconvolution [59, 83], hyperspectral imaging [43, 41, 89, 102], sparse reduced-rank regression [22, 100], and graph denoising and refinement [80, 103]. In these settings, the hope is that due to leveraging the simultaneous structure, $\Omega(r(s_1 + s_2))$ observations suffice to identify the data matrix. For $r \ll s_1, s_2 \ll n_1, n_2$, these bounds are significantly smaller than the bounds for single-structured data matrices.

From an algorithmic point of view, however, the simultaneously structured recovery problem poses obstacles that are *not* present for problems where $\mathbf{X}_\star$ is only of low-rank *or* (group) sparse: In the latter case, variational methods [6] that formulate the reconstruction method in terms of optimizing a suitable objective with a structural regularization term involving $\ell_p/\ell_{2,p}$-(quasi-)norms and and $S_p$-Schatten (quasi-)norms have been well-understood, leading to tractable algorithms in the information theoretically optimal regime [28, 17, 38].
For simultaneously structured problems, on the other hand, Oymak et al. showed in [74] that a mere linear combination of *convex* regularizers for different sparsity structures — in our case, nuclear and $\ell_{2,1}$-norms — cannot outperform recovery guarantees of the "best" one of them alone. While this indicates that leveraging two priors at once is a way more intricate problem than leveraging a single prior, it was also shown in [74] that minimizing a linear combination of rank and row sparsity *can* indeed lead to guaranteed recovery from $\Omega(r(s_1 + s_2))$ measurements. The downside is that the combination of these *non-convex and discontinuous* quantities does not lend itself directly to practical optimization algorithms, and to the best of our knowledge, so far, there have been no works directly tackling the optimization of a combination of *non-convex surrogates* that come with any sort of convergence guarantees.

## 1.1 Contribution

In this work, we approach the reconstruction of simultaneously sparse and low-rank matrices by leveraging the positive results of [74] for non-convex regularizers. To this end, we introduce a family of non-convex, but continuously differentiable regularizers that are tailored to the recovery problem for simultaneously structured data. The resulting objectives lend themselves to efficient optimization by a novel algorithm from the class of *iteratively reweighted least squares (IRLS)* [26, 35, 70, 5, 1, 54, 55], the convergence of which we analyze in the information theoretically (near-)optimal regime. Specifically, our main contributions are threefold:

(i) In Algorithm 1, we propose a novel IRLS method that is tailored to leveraging both structures of the latent solution, sparsity and low-rankness, at once. The core components of the algorithm are the weight operator defined in Definition 2.1 and the update of the smoothing parameters in (12). Notably, the algorithm automatically balances between its low-rank and its sparsity promoting terms, leading to a reliable identification of $s_1$-row-sparse and rank-$r$ ground truths[2].

(ii) Under the assumption that $\mathcal{A}$ behaves almost isometrically on the set of row-sparse and low-rank matrices, we show in Theorem 2.5 that locally Algorithm 1 exhibits quadratic convergence towards $\mathbf{X}_\star$. Note that if $\mathcal{A}$ is, e.g., a Gaussian operator, the isometry assumption is (up to log factors) fulfilled in the information theoretic (near-)optimal regime $m = \Omega(r(s_1 + n_2))$ [60].

(iii) Finally, in Section 2.3 we identify the underlying family of objectives that are minimized by Algorithm 1. To make this precise, we define for $\tau > 0$ and $e$ denoting Euler's number the real-valued function $f_\tau : \mathbb{R} \to \mathbb{R}$ such that

$$f_\tau(t) = \begin{cases} \frac{1}{2}\tau^2 \log(et^2/\tau^2), & \text{if } |t| > \tau, \\ \frac{1}{2}t^2, & \text{if } |t| \leq \tau, \end{cases} \tag{2}$$

---

[2]To enhance readability of the presented proofs and results, we restrict ourselves to row-sparsity of $\mathbf{X}_\star$ here. It is straight-forward to generalize the arguments to column-sparsity as well.

which is quadratic around the origin and otherwise logarithmic in its argument. Using this definition, we define for $\varepsilon > 0$ the $(\varepsilon-)$*smoothed* log-*determinant* objective $\mathcal{F}_{lr,\varepsilon}$ : $\mathbb{R}^{n_1 \times n_2} \to \mathbb{R}$ and for $\delta > 0$ the $(\delta-)$*smoothed sum of logarithmic row-wise $\ell_2$-norms* objective $\mathcal{F}_{sp,\delta} : \mathbb{R}^{n_1 \times n_2} \to \mathbb{R}$ such that

$$\mathcal{F}_{lr,\varepsilon}(\mathbf{X}) = \sum_{r=1}^{\min\{n_1,n_2\}} f_\varepsilon(\sigma_r(\mathbf{X})), \qquad \mathcal{F}_{sp,\delta}(\mathbf{X}) = \sum_{i=1}^{n_1} f_\delta(\|\mathbf{X}_{i,:}\|_2). \qquad (3)$$

Combining the above, we further define the $(\varepsilon, \delta-)$*smoothed logarithmic surrogate* objective $\mathcal{F}_{\varepsilon,\delta} : \mathbb{R}^{n_1 \times n_2} \to \mathbb{R}$ as

$$\mathcal{F}_{\varepsilon,\delta}(\mathbf{X}) := \mathcal{F}_{lr,\varepsilon}(\mathbf{X}) + \mathcal{F}_{sp,\delta}(\mathbf{X}). \qquad (4)$$

In Theorem 2.6, we prove for *any* $\mathcal{A}$ that the iterates of Algorithm 1 minimize quadratic majorizations of $\mathcal{F}_{\varepsilon,\delta}$ and form a non-increasing sequence on $\mathcal{F}_{\varepsilon,\delta}$. To the best of our knowledge, the proposed method is the so far only approach for recovering simultaneously sparse and low-rank matrices which combines local (quadratic) convergence with a rigorous variational interpretation.

The numerical simulations in Section 4 support our theoretical findings and provide empirical evidence for the efficacy of the proposed method.

## 1.2 Related Work

**Sparse and Low-Rank Recovery.** Whereas leveraging a single matrix structure like sparsity *or* low-rankness in the reconstruction process can easily be obtained by convex regularizers [78, 16], Oymak et al. [74] showed that, if one is interested in near-optimal sampling rates, one cannot expect comparably simple solutions for identifying simultaneously structured objects; a minimization of (4) with *convex* terms $\mathcal{F}_{lr,\varepsilon}(\cdot)$ and $\mathcal{F}_{sp,\delta}(\cdot)$ would be only as good as using the one structure that is information theoretically more favorable. A closely related problem[3] that appears in statistical literature under the name *Sparse Principal Component Analysis* (SPCA) [104, 25] is known to be NP-hard in general [67]. Despite the the intrinsic hardness of simultaneously structured recovery problems, promising empirical results for hyperspectral image demixing were shown in [40], where minimization problems involving the sum of *reweighted* convex surrogates are solved by a proximal scheme based on ADMM for the case of simultaneously sparse, low-rank and non-negative matrices. For the problem of simultaneously sparse and low-rank matrix recovery, there exist only a handful approaches that come with rigorous theoretical analysis. The first line of works [4, 37] aims to overcome the aforementioned limitations of purely convex methods in a neat way. They assume that the operator $\mathcal{A}$ has a nested structure such that basic solvers for low-rank resp. row/column-sparse recovery can be applied in two consecutive steps. Despite being an elegant idea, this approach clearly restricts possible choices for $\mathcal{A}$ and is of hardly any practical use.

In a second line of work, Lee et al. [60] consider general impulse response operators that satisfy a suitable restricted isometry property for $s_1$-row- and $s_2$-column-sparse rank-$r$ matrices. They propose and analyze a highly efficient, greedy method, the so-called *Sparse Power Factorization* (SPF) which is a modified version of power factorization [48] and uses hard thresholding pursuit [36] to enforce sparsity in addition. In particular, they show that if $\mathbf{X}_\star$ is rank-$R$, has $s_1$-sparse columns and $s_2$-sparse rows, then $m \gtrsim R(s_1 + s_2) \log(\max\{en_1/s_1, en_2/s_2\})$ Gaussian observations suffice for robust recovery, which is up to the log-factor at the information theoretical limit we discussed above. The result however assumes a low noise level and requires that SPF is initialized by a, in general, intractable method. Only in the special case that $\mathbf{X}_\star$ is spiky, which means that the norms of non-zero rows/columns exhibit a fast decay, a tractable substitute for the initialization method works provably. The analysis of SPF has been extended to the blind deconvolution setup in [59]. In [100], a related approach that combines gradient descent of a smooth objective with hard thresholding is considered, for which the authors show linear convergence from a suitable intialization if the measurement operator satisfies a restricted strong convexity and smoothness assumption.

A third line of work, approaches the problem from a variational point of view. In [33, 69] the

---

[3]Since observations in SPCA are provided from noisy samples of the underlying distribution, whereas in our case the matrix itself is observed indirectly, it is however hard to directly compare results from sparse and low-rank matrix reconstruction with corresponding results for SPCA.

authors aim at enhancing robustness of recovery by alternating minimization of an $\ell_1$-norm based multi-penalty functional. In essence, the theoretical results bound the reconstruction error of global minimizers of the proposed functional depending on the number of observations. Although the authors only provide local convergence guarantees for the proposed alternating methods, the theoretical error bounds for global minimizers hold for arbitrarily large noise magnitudes and a wider class of ground-truth matrices than the one considered in [60].

The works [37, 29], which build upon generalized projection operators to modify iterative hard thresholding to the simultaneous setting, share the lack of global convergence guarantees.

In [79], the authors examine the use of atomic norms to perform recovery of simultaneously sparse and low-rank matrices, which uses a related, but different sparsity assumption compared to the row or column sparsity studied here. From a practical point of view, the such norms are hard to compute and the paper only proposes a heuristic polynomial time algorithm for the problem.

Finally, the alternative approach of using optimally weighted sums or maxima of convex regularizers [52] requires optimal tuning of the parameters under knowledge of the ground-truth.

**Iteratively Reweighted Least Squares.** The herein proposed iteratively reweighted least squares algorithm builds on a long line of research on IRLS going back to Weiszfeld's algorithm proposed in the 1930s for a facility location problem [95, 5]. IRLS is a practical framework for the optimization of non-smooth, possibly non-convex, high-dimensional objectives that minimizes quadratic models which majorize these objectives. Due to its ease of implementation and favorable data-efficiency, it has been widely used in compressed sensing [42, 18, 26, 57, 34, 55], robust statistics [45, 2, 72], computer vision [19, 61, 84], low-rank matrix recovery and completion [35, 70, 56, 54], and in inverse problems involving group sparsity [21, 101, 20]. Recently, it has been shown [62] that dictionary learning techniques can be incorporated into IRLS schemes for sparse and low-rank recovery to allow the learning of a sparsifying dictionary while recovering the solution. Whereas IRLS can be considered as a type of majorize-minimize algorithm [58], optimal performance is achieved if intertwined with a smoothing strategy for the original objective, in which case globally linear (for convex objectives) [26, 1, 72, 55, 75] and locally superlinear (for non-convex objectives) [26, 56, 54, 75] convergence rates have been shown under suitable conditions on the linear operator $\mathcal{A}$.

However, there has only been little work on IRLS optimizing a sum of heterogenous objectives [81] — including the combination of low-rank promoting and sparsity-promoting objectives — nor on the convergence analysis of any such methods. The sole algorithmically related approach for our setting has been studied in [23], where a method has been derived in a sparse Bayesian learning framework, the main step of which amounts to the minimization of weighted least squares problems. Whereas the algorithm of [23] showcases that such a method can empiricially identify simultaneously structured matrices from a small number of measurements, no convergence guarantees or rates have been provided in the information-theoretically optimal regime. Furthermore, [23] only focuses on general sparsity rather than row or column sparsity.

## 1.3 Notation

We denote matrices and vectors by bold upper- and lower-case letters to distinguish them from scalars and functions. We furthermore denote the $i$-th row of a matrix $\mathbf{Z} \in \mathbb{R}^{n_1 \times n_2}$ by $\mathbf{Z}_{i,:}$ and the $j$-th column of $\mathbf{Z}$ by $\mathbf{Z}_{:,j}$. We abbreviate $n = \min\{n_1, n_2\}$. We denote the $r$-th singular value of a matrix $\mathbf{Z} \in \mathbb{R}^{n_1 \times n_2}$ by $\sigma_r(\mathbf{Z})$. Likewise, we denote the in $\ell_2$-norm $s$-largest row of $\mathbf{Z}$ by $\rho_s(\mathbf{Z})$. (To determine $\rho_s(\mathbf{Z})$, we form the in $\ell_2$-norm non-increasing rearrangement of the rows of $\mathbf{Z}$ and, by convention, sort rows with equal norm according to the row-index.) We use $\circ$ to denote the Hadamard (or Schur) product, i.e., the entry-wise product of two vectors/matrices.

We denote the Euclidean $\ell_2$-norm of a vector $\mathbf{z} \in \mathbb{R}^n$ by $\|\mathbf{z}\|_2$. For $\mathbf{Z} \in \mathbb{R}^{n_1 \times n_2}$, the matrix norms we use encompass the operator norm $\|\mathbf{Z}\| := \sup_{\|\mathbf{w}\|_2=1} \|\mathbf{Zw}\|_2$, the row-sum $p$-quasinorm $\|\mathbf{Z}\|_{p,2} := \left( \sum_{i=1}^{n_1} \|\mathbf{Z}_{i,:}\|_2^p \right)^{1/p}$, the row-max norm $\|\mathbf{Z}\|_{\infty,2} := \max_{i \in [n_1]} \|\mathbf{Z}_{i,:}\|_2$, and the Schatten-p quasinorm $\|\mathbf{Z}\|_{S_p} := \left( \sum_{r=1}^{n} \sigma_r(\mathbf{Z})^p \right)^{1/p}$. Note that two special cases of Schatten quasinorms are the nuclear norm $\|\mathbf{Z}\|_* := \|\mathbf{Z}\|_{S_1}$ and the Frobenius norm $\|\mathbf{Z}\|_F := \|\mathbf{Z}\|_{S_2}$.

## 2 IRLS for Sparse and Low-Rank Reconstruction

Recall that we are interested in recovering a rank-$r$ and $s$-row-sparse matrix $\mathbf{X}_\star \in \mathbb{R}^{n_1 \times n_2}$ from $m$ linear observations

$$\mathbf{y} = \mathcal{A}(\mathbf{X}_\star) \in \mathbb{R}^m, \tag{5}$$

i.e., $\mathcal{A} \colon \mathbb{R}^{n_1 \times n_2} \to \mathbb{R}^m$ is linear. We write $\mathbf{X}_\star \in \mathcal{M}_{r,s}^{n_1,n_2} := \mathcal{M}_r^{n_1,n_2} \cap \mathcal{N}_s^{n_1,n_2}$, where $\mathcal{M}_r^{n_1,n_2} \subset \mathbb{R}^{n_1 \times n_2}$ denotes the set of matrices with rank at most $r$ and $\mathcal{N}_s^{n_1,n_2} \subset \mathbb{R}^{n_1 \times n_2}$ denotes the set of matrices with at most $s$ non-zero rows. For convenience, we suppress the indices $n_1$ and $n_2$ whenever the ambient dimension is clear from the context. In particular, we know that $\mathbf{X}_\star = \mathbf{U}_\star \mathbf{\Sigma}_\star \mathbf{V}_\star^*$, where $\mathbf{U}_\star \in \mathcal{N}_s^{n_1,r}$, $\mathbf{\Sigma}_\star \in \mathbb{R}^{r \times r}$, and $\mathbf{V}_\star \in \mathbb{R}^{n_2 \times r}$ denote the reduced SVD of $\mathbf{X}_\star$. Furthermore, the row supports of $\mathbf{X}_\star$ and $\mathbf{U}_\star$ (the index sets of non-zero rows of $\mathbf{X}_\star$ resp. $\mathbf{U}_\star$) are identical, i.e., $\mathrm{supp}(\mathbf{X}_\star) = \mathrm{supp}(\mathbf{U}_\star) = S_\star \subset [n_1] := \{1, \ldots, n_1\}$.

### 2.1 How to Combine Sparse and Low-Rank Weighting

As discussed in Section 1, the challenge in designing a reconstruction method for (5) lies in simultaneously leveraging both structures of $\mathbf{X}_\star$ in order to achieve the optimal sample complexity of $m \approx r(s + n_2)$. To this end, we propose a novel IRLS-based approach in Algorithm 1. The key ingredient for computing the $(k + 1)$-st iterate $\mathbf{X}^{(k+1)} \in \mathbb{R}^{n_1 \times n_2}$ in Algorithm 1 is the multi-structural weight operator $W_{\mathbf{X}^{(k)}, \varepsilon_k, \delta_k} \colon \mathbb{R}^{n_1 \times n_2} \to \mathbb{R}^{n_1 \times n_2}$ of Definition 2.1, which depends on the current iterate $\mathbf{X}^{(k)}$.

---

**Definition 2.1.** *For $\delta_k, \varepsilon_k > 0$, $\sigma_i^k := \sigma_i(\mathbf{X}^{(k)})$, and $\mathbf{X}^{(k)} \in \mathbb{R}^{n_1 \times n_2}$, let*

$$r_k := |\{i \in [n] : \sigma_i^{(k)} > \varepsilon_k\}| \tag{6}$$

*denote the number of singular values $\sigma^{(k)} = (\sigma_i^{(k)})_{i=1}^{r_k}$ of $\mathbf{X}^{(k)}$ larger than $\varepsilon_k$ and furthermore be*

$$s_k := \left|\{i \in [n_1] : \|\mathbf{X}_{i,:}^{(k)}\|_2 > \delta_k\}\right| \tag{7}$$

*the number of rows of $\mathbf{X}^{(k)}$ with $\ell_2$-norm larger than $\delta_k$. Define the $r_k$ left and right singular vectors of $\mathbf{X}^{(k)}$ as columns of $\mathbf{U} \in R^{n_1 \times r_k}$ and $\mathbf{V} \in \mathbb{R}^{n_2 \times r_k}$, respectively, corresponding to the leading singular values $\boldsymbol{\sigma}^{(k)} = (\sigma_i^{(k)})_{i=1}^{r_k} := (\sigma_i(\mathbf{X}^{(k)}))_{i=1}^{r_k}$.*

*We define the weight operator $W_{\mathbf{X}^{(k)}, \varepsilon_k, \delta_k} \colon \mathbb{R}^{n_1 \times n_2} \to \mathbb{R}^{n_1 \times n_2}$ at iteration $k$ of Algorithm 1 as*

$$W_{\mathbf{X}^{(k)}, \varepsilon_k, \delta_k}(\mathbf{Z}) = W_{\mathbf{X}^{(k)}, \varepsilon_k}^{lr}(\mathbf{Z}) + \mathbf{W}_{\mathbf{X}^{(k)}, \delta_k}^{sp} \mathbf{Z} \tag{8}$$

*where $W_{\mathbf{X}^{(k)}, \varepsilon_k}^{lr} \colon \mathbb{R}^{n_1 \times n_2} \to \mathbb{R}^{n_1 \times n_2}$ is its low-rank promoting part*

$$W_{\mathbf{X}^{(k)}, \varepsilon_k}^{lr}(\mathbf{Z}) = [\mathbf{U} \quad \mathbf{U}_\perp] \, \mathbf{\Sigma}_{\varepsilon_k}^{-1} \begin{bmatrix} \mathbf{U}^* \\ \mathbf{U}_\perp^* \end{bmatrix} \mathbf{Z} \, [\mathbf{V} \quad \mathbf{V}_\perp] \, \mathbf{\Sigma}_{\varepsilon_k}^{-1} \begin{bmatrix} \mathbf{V}^* \\ \mathbf{V}_\perp^* \end{bmatrix} \tag{9}$$

*with $\mathbf{\Sigma}_{\varepsilon_k} = \max(\sigma_i^{(k)}/\varepsilon_k, 1)$ and $\mathbf{W}_{\mathbf{X}^{(k)}, \delta_k}^{sp} \in \mathbb{R}^{n_1 \times n_1}$ is its sparsity-promoting part, which is diagonal with*

$$\left(\mathbf{W}_{\mathbf{X}^{(k)}, \delta_k}^{sp}\right)_{ii} = \max\left(\|(\mathbf{X}^{(k)})_{i,:}\|_2^2/\delta_k^2, 1\right)^{-1}, \qquad \text{for all } i \in [n_1]. \tag{10}$$

*The matrices $\mathbf{U}_\perp$ and $\mathbf{V}_\perp$ are arbitrary complementary orthogonal bases for $\mathbf{U}$ and $\mathbf{V}$ that do do not need to be computed in Algorithm 1.*

---

**Remark 2.2.** *For the sake of conciseness, we only consider row-sparsity here. Algorithm 1 and its analysis can however be modified to cover row- and column-sparse matrices as well. For instance, in the symmetric setting $\mathbf{X}_\star = \mathbf{X}_\star^T$ (naturally occurring in applications like sparse phase retrieval) one would define the weight operator $W_{\mathbf{X}^{(k)}, \varepsilon_k, \delta_k}$ as in (8), but with an additional term that multiplies $\mathbf{W}_{\mathbf{X}^{(k)}, \delta_k}^{sp}$ from the right to $\mathbf{Z}$, which corresponds to minimizing the sum of three smoothed logarithmic surrogates. In this case, the solving modified weighted least squares problem (11) will have similar complexity (potentially smaller complexity, as additional symmetries can be exploited).*

---

**Algorithm 1** IRLS for simultaneously low-rank rand row-sparse matrices

---

1: **Input:** Linear operator $\mathcal{A}\colon \mathbb{R}^{n_1\times n_2}\to\mathbb{R}^m$, data $\mathbf{y}\in\mathbb{R}^m$, rank and sparsity estimates $\widetilde{r}$ and $\widetilde{s}$.
2: Initialize $k=0$, $W_{\mathbf{X}^{(0)},\varepsilon_0,\delta_0}=\mathbf{Id}$ and set $\delta_k,\varepsilon_k=\infty$.
3: **for** $k=1$ to $K$ **do**
4:     **Weighted Least Squares:** Update iterate $\mathbf{X}^{(k)}$ by

$$\mathbf{X}^{(k)}=\operatorname*{arg\,min}_{\mathbf{X}:\mathcal{A}(\mathbf{X})=\mathbf{y}}\langle\mathbf{X},W_{\mathbf{X}^{(k-1)},\varepsilon_{k-1},\delta_{k-1}}(\mathbf{X})\rangle. \tag{11}$$

5:     **Update Smoothing :** Compute $\widetilde{r}+1$-st singular value $\sigma^{(k)}_{\widetilde{r}+1}:=\sigma_{\widetilde{r}+1}(\mathbf{X}^{(k)})$ and $(\widetilde{s}+1)$-st
    largest row $\ell_2$-norm $\rho_{\widetilde{s}+1}(\mathbf{X}^{(k)})$ of $\mathbf{X}^{(k)}$, update

$$\varepsilon_k=\min\left(\varepsilon_{k-1},\sigma^{(k)}_{\widetilde{r}+1}\right),\quad \delta_k=\min\left(\delta_{k-1},\rho_{\widetilde{s}+1}(\mathbf{X}^{(k)})\right). \tag{12}$$

6:     **Update Weight Operator:** For $r_k$ and $s_k$ as in (6) and (7),
      •   compute first $r_k$ singular triplets $\sigma^{(k)}\in\mathbb{R}^{r_k}$, $\mathbf{U}\in\mathbb{R}^{n_1\times r_k}$ and $\mathbf{V}\in\mathbb{R}^{n_2\times r_k}$,
      •   compute $W_{\mathbf{X}^{(k)},\varepsilon_k,\delta_k}$ in (8) via $W^{lr}_{\mathbf{X}^{(k)},\varepsilon_k}$ in (9) and $\mathbf{W}^{sp}_{\mathbf{X}^{(k)},\delta_k}$ in (10).
7: **end for**
8: **Output:** $\mathbf{X}^{(K)}$.

---

Recall $\mathcal{F}_{\varepsilon_k,\delta_k}$, $\mathcal{F}_{lr,\varepsilon_k}$, and $\mathcal{F}_{sp,\delta_k}$ from (3)-(4). The high-level idea of Algorithm 1, as for other IRLS methods, is to minimize quadratic functionals, which we call $\mathcal{Q}_{lr,\varepsilon_k}(\,\cdot\,|\mathbf{X}^{(k)})\colon\mathbb{R}^{n_1\times n_2}\to\mathbb{R}$ and $\mathcal{Q}_{sp,\delta_k}(\,\cdot\,|\mathbf{X}^{(k)})\colon\mathbb{R}^{n_1\times n_2}\to\mathbb{R}$ and define them by

$$\mathcal{Q}_{lr,\varepsilon_k}(\mathbf{Z}|\mathbf{X}^{(k)}):=\mathcal{F}_{lr,\varepsilon_k}(\mathbf{X}^{(k)})+\langle\nabla\mathcal{F}_{lr,\varepsilon_k}(\mathbf{X}^{(k)}),\mathbf{Z}-\mathbf{X}^{(k)}\rangle+\frac{1}{2}\langle\mathbf{Z}-\mathbf{X}^{(k)},W^{lr}_{\mathbf{X}^{(k)},\varepsilon_k}(\mathbf{Z}-\mathbf{X}^{(k)})\rangle,$$

$$\mathcal{Q}_{sp,\delta_k}(\mathbf{Z}|\mathbf{X}^{(k)}):=\mathcal{F}_{sp,\delta_k}(\mathbf{X}^{(k)})+\langle\nabla\mathcal{F}_{sp,\delta_k}(\mathbf{X}^{(k)}),\mathbf{Z}-\mathbf{X}^{(k)}\rangle+\frac{1}{2}\langle\mathbf{Z}-\mathbf{X}^{(k)},\mathbf{W}^{sp}_{\mathbf{X}^{(k)},\delta_k}(\mathbf{Z}-\mathbf{X}^{(k)})\rangle, \tag{13}$$

that majorize $\mathcal{F}_{\varepsilon_k,\delta_k}(\cdot)$ (see Theorem 2.6 below) for any iteration $k$. This minimization leads to the weighted least squares problem (11) in Algorithm 1. This step can be implemented by standard numerical linear algebra (see the supplementary material for a discussion of its computational complexity). As a second ingredient of the method, the smoothing parameters $\varepsilon_k$ and $\delta_k$ of $\mathcal{F}_{\varepsilon_k,\delta_k}$ are updated (i.e., decreased) in step (12) before the weight operator is updated according to the current iterate information. For the weight operator update, it is only necessary to compute row norms and leading singular triplets of $\mathbf{X}^{(k)}$.

**Remark 2.3.** *The particular form of the low-rank promoting part of the weight operator $W^{lr}_{\mathbf{X}^{(k)},\varepsilon_k}$ in (9) is due to [53, 54] and captures optimally spectral information both in the column and row space, unlike prior work on low-rank IRLS [35, 70], while retaining the property that the induced quadratic model $\mathcal{Q}_{lr,\varepsilon_k}(\,\cdot\,|\mathbf{X}^{(k)})$ majorizes $\mathcal{F}_{lr,\varepsilon_k}(\cdot)$ (see proof of Theorem 2.6). This choice is critical to enable a fast local rate as established in Theorem 2.5.*

## 2.2 Local Quadratic Convergence of IRLS

Our first main result states that Algorithm 1 exhibits quadratic convergence in a local neighborhood of $\mathbf{X}_\star$, a property Algorithm 1 shares with several methods from the IRLS family. We only need to assume that $\mathcal{A}$ acts almost isometrically on the set $\mathcal{M}_{r,s}$.

**Definition 2.4.** *We say that a linear operator $\mathcal{A}\colon\mathbb{R}^{n_1\times n_2}\to\mathbb{R}^m$ satisfies the rank-$r$ and row-$s$-sparse restricted isometry property (or $(r,s)$-RIP) with RIP-constant $\delta\in(0,1)$ if*

$$(1-\delta)\|\mathbf{Z}\|_F^2\leq\|\mathcal{A}(\mathbf{Z})\|_2^2\leq(1+\delta)\|\mathbf{Z}\|_F^2,$$

*for all $\mathbf{Z}\in\mathcal{M}_{r,s}$.*

It is worth highlighting that Gaussian operators satisfy the above RIP with high-probability if $m\geq cr(s+n_2)\log(en_1/s)$, for some absolute constant $c>0$, see for instance [60]. Up to log-factors, this is at the information theoretic limit which we discussed in the beginning. The convergence result for Algorithm 1 now reads as follows.

**Theorem 2.5** (Local Quadratic Convergence). *Let $\mathbf{X}_\star \in \mathcal{M}_{r,s}$ be a fixed ground-truth matrix that is $s$-row-sparse and of rank $r$. Let linear observations $\mathbf{y} = \mathcal{A}(\mathbf{X}_\star)$ be given and assume that $\mathcal{A}$ has the $(r,s)$-RIP with $\delta \in (0,1)$. Assume that the $k$-th iterate $\mathbf{X}^{(k)}$ of Algorithm 1 with $\widetilde{r} = r$ and $\widetilde{s} = s$ updates the smoothing parameters in (12) such that one of the statements $\varepsilon_k = \sigma_{r+1}(\mathbf{X}^{(k)})$ or $\delta_k = \rho_{s+1}(\mathbf{X}^{(k)})$ is true, and that $r_k \geq r$ and $s_k \geq s$.*

*If $\mathbf{X}^{(k)}$ satisfies*

$$\|\mathbf{X}^{(k)} - \mathbf{X}_\star\| \leq \frac{1}{48\sqrt{n}c_{\|\mathcal{A}\|_{2\to2}}^3} \min\left\{\frac{\sigma_r(\mathbf{X}_\star)}{r}, \frac{\rho_s(\mathbf{X}_\star)}{s}\right\} \tag{14}$$

*where $c_{\|\mathcal{A}\|_{2\to2}} = \sqrt{1 + \frac{\|\mathcal{A}\|_{2\to2}^2}{(1-\delta)}}$ and $n = \min\{n_1, n_2\}$, then the local convergence rate is quadratic in the sense that*

$$\|\mathbf{X}^{(k+1)} - \mathbf{X}_\star\| \leq \min\{\mu\|\mathbf{X}^{(k)} - \mathbf{X}_\star\|^2, 0.9\|\mathbf{X}^{(k)} - \mathbf{X}_\star\|\},$$

*for*

$$\mu = 4.179 c_{\|\mathcal{A}\|_{2\to2}}^2 \left(\frac{5r}{\sigma_r(\mathbf{X}_\star)} + \frac{2s}{\rho_s(\mathbf{X}_\star)}\right), \tag{15}$$

*and $\mathbf{X}^{(k+\ell)} \overset{\ell\to\infty}{\to} \mathbf{X}_\star$.*

The proof of Theorem 2.5 is presented in the supplementary material. To the best of our knowledge, so far no other method exists for recovering simultaneously sparse and low-rank matrices that exhibits local quadratic convergence. In particular, the state-of-the-art competitor methods [60, 37, 69, 29] reach a local linear error decay at best.

On the other hand, (14) is rather pessimistic since for Gaussian $\mathcal{A}$ the constant $c_{\|\mathcal{A}\|_{2\to2}}$ scales like $\sqrt{(n_1 n_2)/m}$, which means that the right-hand side of (14) behaves like $m^{3/2}/(n(n_1 n_2)^{3/2})$, whereas we observe quadratic convergence in experiments within an empirically much larger convergence radius. Closing this gap between theory and practical performance is future work.

It is noteworthy that the theory in [60] — to our knowledge the only other related work explicitly characterizing the convergence radius — holds on a neighborhood of $\mathbf{X}_\star$ that is independent of the ambient dimension. The authors of [60] however assume that the RIP-constant decays with the conditioning number $\kappa$ of $\mathcal{A}$, a quantity that might be large in applications. Hence, ignoring log-factors the sufficient number of measurements in [60] scales like $m = \Omega(\kappa^2 r(s_1 + n_2))$. In contrast, Theorem 2.5 works for any RIP-constant less than one which means for $m = \Omega(r(s_1 + n_2))$.

## 2.3 `IRLS` as Quadratic Majorize-Minimize Algorithm

With Theorem 2.5, we have provided a local convergence theorem that quantifies the behavior of Algorithm 1 in a small neighbourhood of the simultaneously row-sparse and low-rank ground-truth $\mathbf{X}_\star \in \mathbb{R}^{n_1 \times n_2}$. The result is based on sufficient regularity of the measurement operator $\mathcal{A}$, which in turn is satisfied with high probability if $\mathcal{A}$ consists of sufficiently generic random linear observations that concentrate around their mean.

In this section we establish that, for *any* measurement operator $\mathcal{A}$, Algorithm 1 can be interpreted within the framework of iteratively reweighted least squares (`IRLS`) algorithms [26, 70, 75], which implies a strong connection to the minimization of a suitable smoothened objective function. In our case, the objective $\mathcal{F}_{\varepsilon,\delta}$ in (4) is a linear combination of sum-of-logarithms terms penalizing both non-zero singular values [31, 70, 14] as well as non-zero rows of a matrix $\mathbf{X}$ [50].

We show in Theorem 2.6 below that the `IRLS` algorithm Algorithm 1 studied in this paper is based on minimizing at each iteration quadratic models that majorize $\mathcal{F}_{\varepsilon,\delta}$, and furthermore, that the iterates $(\mathbf{X}^{(k)})_{k\geq1}$ of Algorithm 1 define a non-increasing sequence $\left(\mathcal{F}_{\varepsilon_k,\delta_k}(\mathbf{X}^{(k)})\right)_{k\geq1}$ with respect to the objective $\mathcal{F}_{\varepsilon,\delta}$ of (4). The proof combines the fact that for fixed smoothing parameters $\varepsilon_k$ and $\delta_k$, the weighted least squares and weight update steps Algorithm 1 can be interpreted as a step of a Majorize-Minimize algorithm [86, 58], with a decrease in the underlying objective (4) for updated smoothing parameters.

**Theorem 2.6.** *Let $\mathbf{y} \in \mathbb{R}^m$, let the linear operator $\mathcal{A}: \mathbb{R}^{n_1 \times n_2} \to \mathbb{R}^m$ be arbitrary. If $(\mathbf{X}^{(k)})_{k\geq1}$ is a sequence of iterates of Algorithm 1 and $(\delta_k)_{k\geq1}$ and $(\varepsilon_k)_{k\geq1}$ are the sequences of smoothing parameters as defined therein, then the following statements hold.*

1. *The quadratic model functions $\mathcal{Q}_{lr,\varepsilon_k}(\cdot|\mathbf{X}^{(k)})$ and $\mathcal{Q}_{sp,\delta_k}(\cdot|\mathbf{X}^{(k)})$ defined in (13) globally majorize the $(\varepsilon_k, \delta_k)$−smoothed logarithmic surrogate objective $\mathcal{F}_{\varepsilon_k, \delta_k}$, i.e., for any $\mathbf{Z} \in \mathbb{R}^{n_1 \times n_2}$, it holds that*

$$\mathcal{F}_{\varepsilon_k, \delta_k}(\mathbf{Z}) \leq \mathcal{Q}_{lr,\varepsilon_k}(\mathbf{Z}|\mathbf{X}^{(k)}) + \mathcal{Q}_{sp,\delta_k}(\mathbf{Z}|\mathbf{X}^{(k)}). \tag{16}$$

2. *The sequence $\left(\mathcal{F}_{\varepsilon_k, \delta_k}(\mathbf{X}^{(k)})\right)_{k \geq 1}$ is non-increasing.*

3. *If $\overline{\varepsilon} := \lim_{k \to \infty} \varepsilon_k > 0$ and $\overline{\delta} := \lim_{k \to \infty} \delta_k > 0$, then $\lim_{k \to \infty} \|\mathbf{X}^{(k)} - \mathbf{X}^{(k+1)}\|_F = 0$. Furthermore, in this case, every accumulation point of $(\mathbf{X}^{(k)})_{k \geq 1}$ is a stationary point of*

$$\min_{\mathbf{X}:\mathcal{A}(\mathbf{X})=\mathbf{y}} \mathcal{F}_{\overline{\varepsilon}, \overline{\delta}}(\mathbf{X}).$$

The proof of Theorem 2.6 is presented in the supplementary material.

## 3 Discussion of Computational Complexity

It is well-known that the solution of the linearly constrained weighted least squares problem (11) can be written as

$$\mathbf{X}^{(k)} = W_{k-1}^{-1}\mathcal{A}^* \left(\mathcal{A}W_{k-1}^{-1}\mathcal{A}^*\right)^{-1} \mathbf{y} \tag{17}$$

where $W_{k-1} := W_{\mathbf{X}^{(k-1)},\varepsilon_{k-1},\delta_{k-1}}$ is the weight operator (8) of iteration $k-1$ [26, 54]. In [54, Theorem 3.1 and Supplementary Material], it was shown that in the case of low-rank matrix completion without the presence of a row-sparsity inducing term, this weighted least squares problem can be solved by solving an equivalent, well-conditioned linear system via an iterative solver that uses the application of a system matrix whose matrix-vector products have time complexity of $O(mr + r^2 \max(n_1, n_2))$.

In the case of Algorithm 1, the situations is slightly more involved as we cannot provide an explicit formula for the inverse of the weight operator $W_{k-1}$ as it amounts to the sum of the weight operators $W_{\mathbf{X}^{(k-1)},\varepsilon_{k-1}}^{lr}$ and $\mathbf{W}_{\mathbf{X}^{(k-1)},\delta_{k-1}}^{sp}$ that are diagonalized by different, mutually incompatible bases. However, computing this inverse is facilitated by the *Sherman-Morrison-Woodbury* formula [97]

$$(\mathbf{E}\mathbf{C}\mathbf{F}^* + \mathbf{B})^{-1} = \mathbf{B}^{-1} - \mathbf{B}^{-1}\mathbf{E}(\mathbf{C}^{-1} + \mathbf{F}^*\mathbf{B}^{-1}\mathbf{E})^{-1}\mathbf{F}^*\mathbf{B}^{-1}$$

for suitable matrices of compatible size $\mathbf{E}, \mathbf{F}$ and invertible $\mathbf{C}, \mathbf{B}$ and the fact that both $W_{\mathbf{X}^{(k-1)},\varepsilon_{k-1}}^{lr}$ and $\mathbf{W}_{\mathbf{X}^{(k-1)},\delta_{k-1}}^{sp}$ exhibit a "low-rank plus (scaled) identity" or a "sparse diagonal plus (scaled) identity" structure. After a simple application of the SMW formula, (17) can be rewritten such that the computational bottleneck becomes the assembly and inversion of a $O(r_k \max(n_1, n_2))$ linear system. We note that in general, this can be done exactly in a time complexity of $O(r_k^3 max(n_1, n_2)^3)$ using standard linear algebra. A crucial factor in the computational cost of the method is also the structure of the measurement operator $\mathcal{A}$ defining the problem, as the application of itself and its adjoint can significantly influence the per-iteration cost of IRLS; for dense Gaussian measurements, just processing the information of $\mathcal{A}$ amounts to $mn_1n_2$ flops. If rank-one or Fourier-type measurements are taken, this cost can significantly be reduced, see [29, Table 1 and Section 3] for an analogous discussion.

We refer to the MATLAB implementation available in the repository https://github.com/ckuemmerle/simirls for further details. While our implementation is not optimized for large-scale problems, the computational cost of Algorithm 1 was observed to be comparable to the implementations of SPF or RiemAdaIHT provided by the authors [69, 29]. We leave further improvements and adaptations to large-scale settings to future work.

## 4 Numerical Evaluation

In this section, we explore the empirical performance of IRLS in view of the theoretical results of Theorems 2.5 and 2.6, and compare its ability to recover simultaneous low-rank and row-sparse data matrices with the state-of-the-art methods Sparse Power Factorization (SPF) [60] and Riemannian adaptive iterative hard thresholding (RiemAdaIHT) [29], which are

among the methods with the best empirical performance reported in the literature. The method `ATLAS` [33] and its successor [69] are not used in our empirical studies since they are tailored to robust recovery and yield suboptimal performance when seeking high-precision reconstruction in low noise scenarios. We use spectral initialization for `SPF` and `RiemAdaIHT`. The weight operator of IRLS is initialized by the identity as described in Algorithm 1, solving an unweighted least squares problem in the first iteration. A detailed description of the experimental setup can be found in Appendix A.1.

**Performance in Low-Measurement Regime.** Figures 1 and 2 show the empirical probability of successful recovery when recovering $s$-row sparse ground-truths $\mathbf{X}_\star \in \mathbb{R}^{256 \times 40}$ of rank $r = 1$ (resp. $r = 5$) from Gaussian measurements under oracle knowledge on $r$ and $s$. The results are averaged over 64 random trials. As both figures illustrate, the region of success of `IRLS` comes closest to the information theoretic limit of $r(s + n_2 - r)$ which is highlighted by a red line, requiring a significantly lower oversampling factor than the baseline methods.

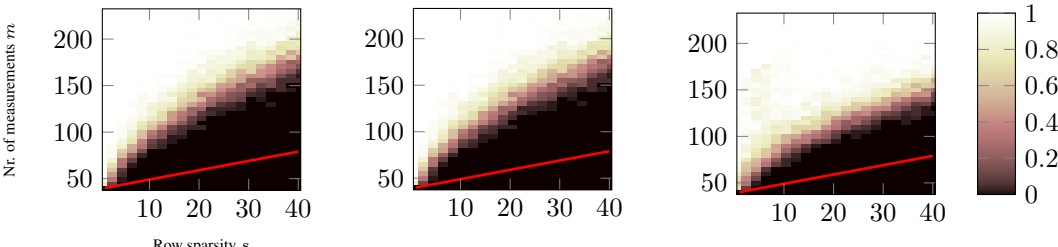

Figure 1: Left column: `RiemAdaIHT`, center: `SPF`, right: `IRLS`. Phase transition experiments with $n_1 = 256$, $n_2 = 40$, $r = 1$, Gaussian measurements. Algorithmic hyperparameters informed by model order knowledge (i.e., $\widetilde{r} = r$ and $\widetilde{s} = s$ for `IRLS`). White corresponds to empirical success rate of $1$, black to $0$.

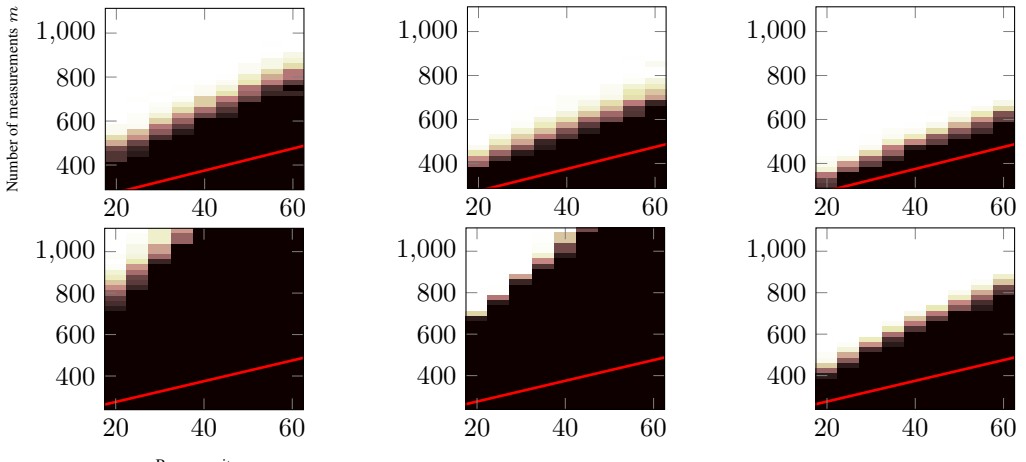

Figure 2: Left column: `RiemAdaIHT`, center: `SPF`, right: `IRLS`. First row: As in Figure 1, but for data matrix $\mathbf{X}_\star$ of rank $r = 5$. Second row: As first row, but hyper-parameters $r$ and $s$ are overestimated as $\widetilde{r} = 2r = 10$, $\widetilde{s} = \lfloor 1.5s \rfloor$

In Appendix A.2 and Appendix A.3 in the supplementary material, we report on similar experiments conducted for other measurement operators than dense Gaussians, in which cases the empirical relative behavior of the methods is comparable.

**Sensitivity to Parameter Choice.** In applications of our setting, the quantities $r$ and $s$ might be unknown or difficult to estimate. In the second row of Figure 2, we repeat the experiment of the first row (rank-5 ground truth), but run the algorithms with rank and sparsity estimates of $\widetilde{r} = 2r$ and $\widetilde{s} = \lfloor 1.5s \rfloor$. Whereas all considered methods suffer a deterioration of performance, we observe that `IRLS` deteriorates relatively the least by a large margin. Furthermore, we observe that even if `IRLS` does not recovery $\mathbf{X}_\star$, it converges typically to a matrix that is still low-rank and row-sparse (with

larger $r$ and $s$) satisfying the data constraint, while the other methods fail to convergence to such a matrix.

**Convergence Behavior.** Finally, we examine the convergence rate of the iterates to validate the theoretical prediction of Theorem 2.5 in the setting of Figure 2. Figure 3 depicts in log-scale the approximation error over the iterates of SPF, RiemAdaIHT, and IRLS. We observe that the IRLS indeed exhibits empirical quadratic convergence within a few iterations (around 10), whereas the other methods clearly only exhibit linear convergence. The experiment further suggests that the rather pessimistic size of the convergence radius established by Theorem 2.5 could possibly be improved by future investigations.

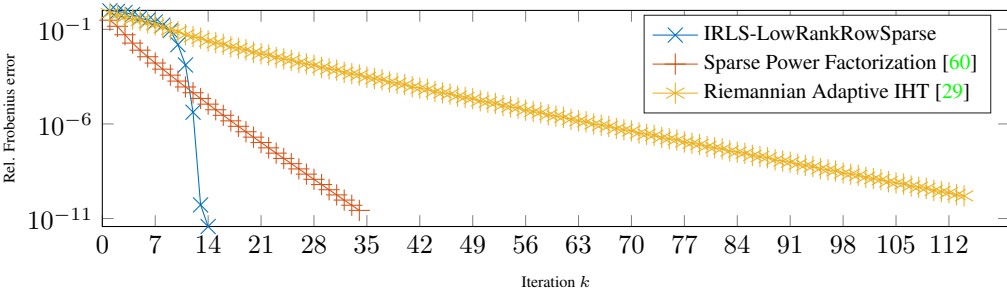

Figure 3: Comparison of convergence rate. Setting as in Figure 2 with $s = 40$ and $m = 1125$.

**Further experiments.** In Appendix A.4, we provide additional experiments investigating the self-balancing property of the objective (4), as well as the experiments on the noise robustness of the method in Appendix A.5.

## 5 Conclusion, Limitations and Future Work

**Conclusion.** In this paper, we adapted the IRLS framework to the problem of recovering simultaneously structured matrices from linear observations focusing on the special case of row sparsity and low-rankness. Our convergence guarantee Theorem 2.5 is hereby the first one for any method minimizing combinations of structural surrogate objectives that holds in the information-theoretic near-optimal regime and exhibits local quadratic convergence. The numerical experiments we conducted for synthetic data suggest that, due to its weak dependence on the choice of hyperparameters, IRLS in the form of Algorithm 1 can be a practical method for identifying simultaneously structured data even in difficult problem instances.

**Limitations and Future Work.** As in the case of established IRLS methods that optimize non-convex surrogate objectives representing a single structure [26, 56, 54, 75], the radius of guaranteed quadratic convergence in Theorem 2.5 is the most restrictive assumption. Beyond the interpretation in terms of surrogate minimization as presented in Theorem 2.6, which holds without any assumptions on the initialization, our method shares a lack of a global convergence guarantees with other non-convex IRLS algorithms [70, 54, 75].

The generalization and application of our framework to combinations of structures beyond rank and row- (or column-)sparsity lies outside the scope of the present paper, but could involve subspace-structured low-rankness [31, 24, 99, 87] or analysis sparsity [30, 77]. A generalization of the presented IRLS framework to higher-order objects such as low-rank tensors is of future interest as convexifications of structure-promoting objectives face similar challenges [71, 74, 98] in this case.

In parameter-efficient deep learning, both sparse [39, 44, 32, 91] and low-rank [96, 94, 82] weight parameter models have gained considerable attention due to the challenges of training and storing, e.g., in large transformer-based models [92, 9]. It will be of interest to study whether in this non-linear case IRLS-like preconditioning of the parameter space can find network weights that are simultaneously sparse and low-rank, and could potentially lead to further increases in efficiency.

## Acknowledgements

The authors thank Max Pfeffer for providing their implementation of the algorithm `RiemAdaIHT` as considered in [29].

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

**Supplementary material for** *Recovering Simultaneously Structured Data via Non-Convex Iteratively Reweighted Least Squares*

This supplement is structured as follows.

## A    Experimental Setup and Supplementary Experiments

In this section, we elaborate on the detailed experimental setup that was used in Section 4 of the main paper. Furthermore, we provide additional experiments comparing the behavior of the three methods studied in Section 4 for linear measurement operators $\mathcal{A}$ that are closer to operators that can be encountered in applications of simultaneous low-rank and group-sparse recovery. Finally, we shed light on the evolution of the objective function (4) of IRLS (Algorithm 1), including in situations where the algorithm does not manage to recover the ground truth.

### A.1    Experimental Setup

The experiments of Section 4 were conducted using MATLAB implementations of the three algorithms on different Linux machines using MATLAB versions R2019b or R2022b. In total, the preparation and execution of the experiments used approximately 1200 CPU hours. The CPU models used in the simulations are Dual 18-Core Intel Xeon Gold 6154, Dual 24-Core Intel Xeon Gold 6248R, Dual 8-Core Intel Xeon E5-2667, 28-Core Intel Xeon E5-2690 v3, 64-Core Intel Xeon Phi KNL 7210-F. For Sparse Power Factorization (SPF) [60], we used our custom implementation of [60, Algorithm 4 "rSPF_HTP"] and for Riemannian adaptive iterative hard thresholding (RiemAdaIHT) [29], we used an implementation provided to us by Max Pfeffer in private communications. We refer to Section 3 for implementation details for the IRLS method Algorithm 1.

In all phase transition experiments, we define *successful recovery* such that the relative Frobenius error $\frac{\left\|\mathbf{X}^{(K)}-\mathbf{X}_\star\right\|_F}{\left\|\mathbf{X}_\star\right\|_F}$ of the iterate $\mathbf{X}^{(K)}$ returned by the algorithm relative to the simultaneously low-rank and row-sparse ground truth matrix $\mathbf{X}_\star$ is smaller than the threshold $10^{-4}$. As stopping criteria, we used the criterion that the relative change of Frobenius norm satisfies $\frac{\left\|\mathbf{X}^{(k)}-\mathbf{X}^{(k-1)}\right\|_F}{\left\|\mathbf{X}^{(k)}\right\|_F} < \mathrm{tol}$ for IRLS, the change in the matrix factors norms satisfy $\|\mathbf{U}_k - \mathbf{U}_{k-1}\| < \mathrm{tol}$ and $\|\mathbf{V}_k - \mathbf{V}_{k-1}\| < \mathrm{tol}$ for SPF, and the norm of the Riemannian gradient in RiemAdaIHT being smaller than $\mathrm{tol}$ for $\mathrm{tol} = 10^{-10}$, or if a maximal number of iterations is reached. This iteration threshold was chosen as $\mathrm{max\_iter} = 250$ for IRLS and SPF and as $\mathrm{max\_iter} = 2000$ for RiemAdaIHT, reflecting the fact that RiemAdaIHT is a gradient-type method which might need many iterations to reach a high-accuracy solution. The parameters were chosen so that the stopping criteria do not prevent a method's iterates reaching the recovery threshold if they were to reach $\mathbf{X}_\star$ eventually.

In the experiments, we chose random ground truths $\mathbf{X}_\star \in \mathbb{R}^{n_1 \times n_2}$ of rank $r$ and row-sparsity $s$ such that $\mathbf{X}_\star = \tilde{\mathbf{X}}_\star / \left\|\tilde{\mathbf{X}}_\star\right\|_F$, where $\tilde{\mathbf{X}}_\star = \mathbf{U}_\star \mathrm{diag}(\mathbf{d}_\star) \mathbf{V}_\star^*$, and where $\mathbf{U}_\star \in \mathbb{R}^{n_1 \times r}$ is a matrix with $s$ non-zero rows whose location is chosen uniformly at random and whose entries are drawn from i.i.d. standard Gaussian random variables, $\mathbf{d}$ has i.i.d. standard Gaussian entries and $\mathbf{V}_\star \in \mathbb{R}^{n_2 \times r}$ has likewise i.i.d. standard Gaussian entries.

### A.2    Random Rank-One Measurements

In Section 4, we considered only measurement operator $\mathcal{A}: \mathbb{R}^{n_1 \times n_2} \to \mathbb{R}^m$ whose matrix representation consists of i.i.d. Gaussian entries, i.e., operators such that there are independent matrices

$\mathbf{A}_1, \ldots \mathbf{A}_m$ with i.i.d. standard Gaussian entries such that

$$\mathcal{A}(\mathbf{X})_j = \langle \mathbf{A}_j, \mathbf{X} \rangle_F$$

for any $\mathbf{X} \in \mathbb{R}^{n_1 \times n_2}$. While it is known that such Gaussian measurement operators satisfy the $(r, s)$-RIP of Section 4, which is the basis of our convergence theorem Theorem 2.5, in a regime of a near-optimal number of measurements with high probability, practically relevant measurement operators are often more structured; another downside of dense Gaussian measurements is that it is computationally expensive to implement their action on matrices.

In relevant applications of our setup, however, e.g., in sparse phase retrieval [46, 11, 47] or blind deconvolution [59, 83], the measurement operator consists of rank-one measurements. For this reason, we now conduct experiments in settings related to the ones depicted in Figure 1 and Figure 2 Section 4, but for *random rank-one measurements* where the action of $\mathcal{A} : \mathbb{R}^{n_1 \times n_2} \to \mathbb{R}^m$ on $\mathbf{X}$ can be written as

$$\mathcal{A}(\mathbf{X})_j = \langle \mathbf{a}_j \mathbf{b}_j^*, \mathbf{X} \rangle_F$$

for each $j = 1, \ldots, m$, where $\mathbf{a}_j, \mathbf{b}_j$ are independent random standard Gaussian vectors. In Figure 4, we report the phase transition performance of RiemAdaIHT, SPF and IRLS for $(256 \times 40)$-dimensional ground truths of different row-sparsities and different ranks if we are given such random rank-one measurements.

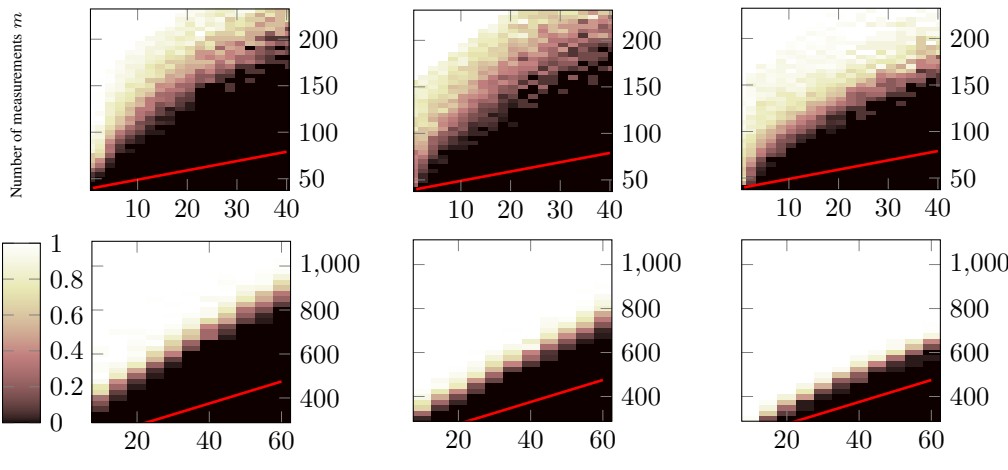

Figure 4: Left column: RiemAdaIHT, center: SPF, right: IRLS. Success rates for the recovery of low-rank and row-sparse matrices from random rank-one measurements. First row: Rank-1 ground truth $\mathbf{X}_\star$ (cf. Figure 1. Second row: Rank-5 ground truth $\mathbf{X}_\star$ (cf. Figure 2).

We observe in Figure 4 that compared to the setting of dense Gaussian measurements, the phase transitions of all three algorithms deteriorate slightly; especially for $r = 1$, one can observe that the transition between no success and high empirical success rate is extends across a larger area. IRLS performs clearly best for both $r = 1$ and $r = 5$, whereas SPF has the second best performance for $r = 5$. For $r = 1$, it is somewhat unclear whether RiemAdaIHT or SPF performs better.

### A.3 Discrete Fourier Rank-One Measurements

We now revisit the experiments of Appendix A.2 for a third measurement setup motivated from blind deconvolution problems [3, 59, 64, 66, 83, 29], which are prevalent in astronomy, medical imaging and communications engineering [49, 13]. In particular, in these settings, if $\mathbf{z} \in \mathbb{R}^m$ is an (unknown) signal and $\mathbf{w} \in \mathbb{R}^m$ is an (unknown) convolution kernel, assume we are given the entries of their convolution $\widetilde{y} = \mathbf{z} * \mathbf{w}$. If we know that $\mathbf{z} = \mathbf{A}\mathbf{u}$ for some known matrix $\mathbf{A} \in \mathbb{R}^{m \times n_1}$ and an $s$-sparse vector $\mathbf{u} \in \mathbb{R}^{n_1}$ and $\mathbf{w} = \mathbf{B}\mathbf{v}$ for some known matrix $\mathbf{B} \in \mathbb{R}^{m \times n_2}$ and arbitrary vector $\mathbf{v} \in \mathbb{R}^{n_2}$, applying the discrete Fourier transform (represented via the DFT matrix $\mathbf{F} \in \mathbb{C}^{m \times m}$), we can write the coordinates of

$$\mathbf{y} = \mathbf{F}\widetilde{y} = \mathrm{diag}(\mathbf{F}\mathbf{z})\mathbf{F}\mathbf{w} = \mathrm{diag}(\mathbf{F}\mathbf{A}\mathbf{u})\mathbf{F}\mathbf{B}\mathbf{v}$$

as

$$\mathbf{y}_j = \mathcal{A}(\mathbf{uv}^*)_j = \langle (\mathbf{FA})^*_{j,:}, \overline{\mathbf{FB}}_{j,:}, \mathbf{uv}^* \rangle_F$$

for each $j = 1, \ldots, m$, which allows us to write the problem as a simultaneously rank-1 and $s$-row sparse recovery problem from Fourier-type measurements.

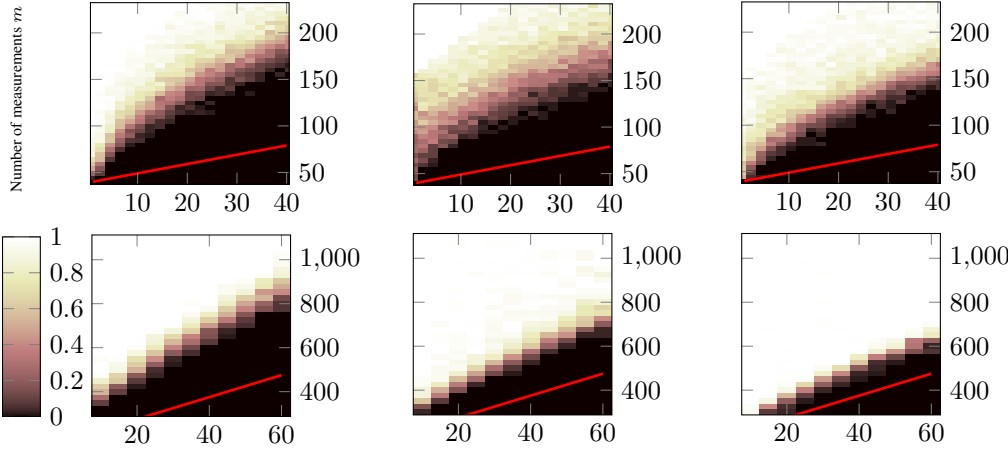

Figure 5: Left column: `RiemAdaIHT`, center: `SPF`, right: `IRLS`. Success rates for the recovery of low-rank and row-sparse matrices from Fourier rank-one measurements. First row: Rank-1 ground truth $\mathbf{X}_\star$. Second row: Rank-5 ground truth $\mathbf{X}_\star$ (cf. Figure 2).

In Figure 5, we report the results of simulations with $\mathbf{A}$ and $\mathbf{B}$ chosen generically as standard real Gaussians for these Fourier-based rank-1 measurements (including for rank-5 ground truths, which goes beyond a blind deconvolution setting). We observe that the transition from no recovery to exact recovery for an increasing number of measurement (with fixed dimension parameters $s$, $n_1$ and $n_2$) happens earlier than for the random Gaussian rank-one measurements of Appendix A.2, but slightly later than for dense Gaussian measurements. Again, `IRLS` exhibits the best empirical data-efficiency with sharpest phase transition curves.

As a summary, we observe that `IRLS` is able to recovery simultaneously low-rank and row-sparse matrices empirically from fewer measurements than state-of-the-art methods for a variety of linear measurement operators, including in cases where the RIP assumption of Definition 2.4 is not satisfied and in cases that are relevant for applications.

### A.4 Evolution of Objective Values

While Theorem 2.5 guarantees local convergence if the measurement operator $\mathcal{A}$ is generic enough and contains enough measurements (RIP-assumption), it is instructive to study the behavior of Algorithm 1 in situations where there are *not* enough measurements available to identify a specific low-rank and row-sparse ground truth $\mathbf{X}_\star$ which respect to which the measurements have been taken.

In this setting, Theorem 2.6 guarantees that the behavior of the `IRLS` methods is still benign as the sequence of $\varepsilon$- and $\delta$-smoothed log-objectives $\left( \mathcal{F}_{\varepsilon_k, \delta_k}(\mathbf{X}^{(k)}) \right)_{k \geq 1}$ from (4) is non-increasing. In Figure 6, we illustrate the evolution of the relative Frobenius error of an iterate to the ground truth $\mathbf{X}_\star$, the $(\varepsilon_k, \delta_k)$-smoothed logarithmic surrogates $\mathcal{F}_{\varepsilon_k, \delta_k}(\mathbf{X}^{(k)})$ as well as of the rank and row-sparsity parts $\mathcal{F}_{lr, \varepsilon_k}(\mathbf{X}^{(k)})$ and $\mathcal{F}_{sp, \delta_k}(\mathbf{X}^{(k)})$ of the objective, respectively, in two typical situations.

In particular, we can see the evolution of these four quantities in the setting of data of dimensionality $n_1 = 128$, $n_2 \in \{20, 40\}$, $s = 20$ and $r = 5$ created as in the other experiments, where a number of $m = 875$ and $m = 175$ (corresponding to an oversampling factor of 3.0 and 1.0, respectively) dense Gaussian measurements are provided to Algorithm 1.

In the left plot of Figure 6, which corresponds to setting of abundant measurements, we observe that the four quantities all track each other relatively well on a semilogarithmic scale (note that we plot the square roots of the objective values to match the order of the (unsquared) relative Frobenius error),

converging to values between $10^{-13}$ and $10^{-11}$ (at which point the stopping criterion of the method applies) within 12 iterations.

In the second plot of Figure 6, the number of measurements exactly matches the number of degrees of freedom of the ground truth, in which case the $\mathbf{X}^{(k)}$ does *not* converge to $\mathbf{X}_\star$. However, it can be seen that Algorithm 1 still finds very meaningful solutions: It can be seen that within 86 iterations, $\mathcal{F}_{\varepsilon_k,\delta_k}(\mathbf{X}^{(k)})$ converges to $\approx 10^{-12}$ (since $\sqrt{\mathcal{F}_{\varepsilon_k,\delta_k}(\mathbf{X}^{(k)})} \approx 10^{-6}$) in a manner that is partially "staircase-like": After 20 initial iterations where $\mathcal{F}_{\varepsilon_k,\delta_k}(\mathbf{X}^{(k)})$ decreases significantly at each iteration, its decrease is dominated by relatively sudden, alternating drops of the (blue) sparsity objective $\mathcal{F}_{sp,\delta_k}(\mathbf{X}^{(k)})$ and the (red) rank objective $\mathcal{F}_{lr,\varepsilon_k}(\mathbf{X}^{(k)})$, which typically do not occur simultaneously.

This illustrates the *self-balancing* property of the two objective terms in the IRLS objective $\mathcal{F}_{\varepsilon_k,\delta_k}(\mathbf{X}^{(k)})$: while the final iterate at iteration $k = 86$ is not of the target row-sparsity $s = 20$ and $r = 5$, it is still 20-row sparse and has essentially rank 6. This means that Algorithm 1 has found an alternative parsimonious solution to the simultaneous low-rank and row-sparse recovery problem that is just slightly less parsimonious.

Arguably, this robust performance in the low-data regime of `IRLS` is rather unique, and to the best of our knowledge, not shared by methods such as `SPF` or `RiemAdaIHT`, which typically breakdown in such a regime.

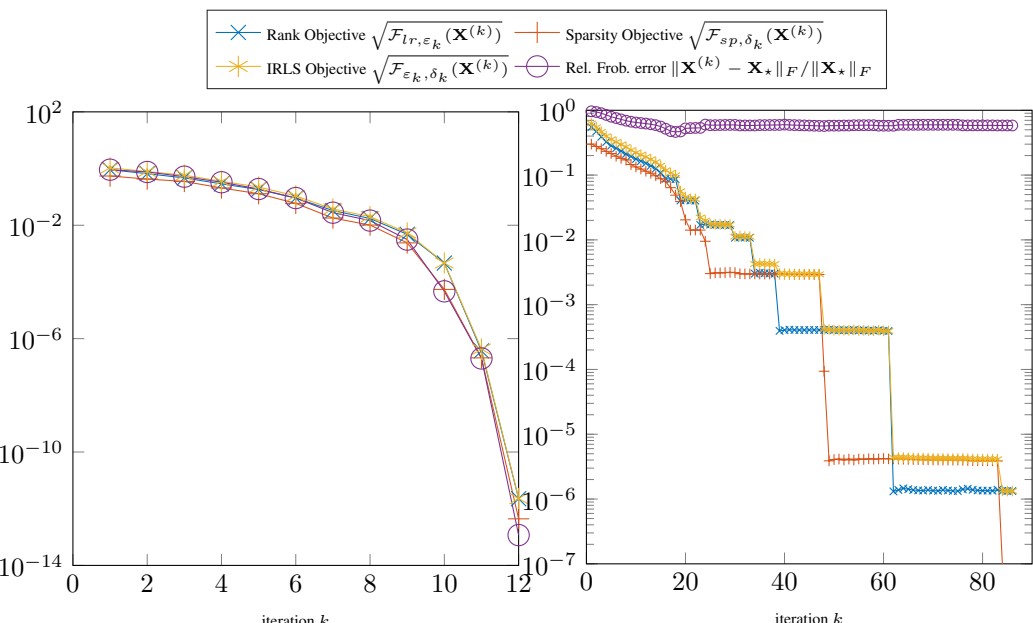

Figure 6: Objective/ error quantities of iterates $\mathbf{X}^{(k)}$ for iterations $k$. Left: Typical result for $n_1 = 128$, $n = 40$, $m = 875$. Right: Typical result for $n_1 = 128$, $n = 20$, $m = 175$.

### A.5 Robustness under Noisy Measurements

The convergence theory for the `IRLS` method Algorithm 1 established in Theorem 2.5 assume that *exact* linear measurements $\mathbf{y} = \mathcal{A}(\mathbf{X}_\star)$ of a row-sparse and low-rank ground truth $\mathbf{X}_\star$ are provided to the algorithm. However, in practice, one would expect that the linear measurement model is only approximately accurate. For IRLS for sparse vector recovery, theoretical guarantees have been established for this case in [26, 57]. We do not extend such results to the simultaneously structured case, but we provide numerical evidence that IRLS as defined in Algorithm 1 can be used directly also for noisy measurements.

To this end, we conduct an experiment in the problem setup of Figure 1 in Section 4 for a fixed row-sparsity of $s = 40$, in which the measurements provided to the algorithms IRLS, RiemAdaIHT

and SPF are such that

$$\mathbf{y} = \mathcal{A}(\mathbf{X}_\star) + \mathbf{w},$$

where $\mathbf{w}$ is a Gaussian vector (i.i.d. entries) with standard deviation of $\sigma = \sqrt{\frac{\|\mathcal{A}(\mathbf{X}_\star)\|_2^2}{m \cdot \text{SNR}}}$ and where SNR is a varying signal-to-noise ratio. We consider SNRs between $10$ and $10^{12}$, and report the resulting relative Frobenius error statistics in Figure 7.

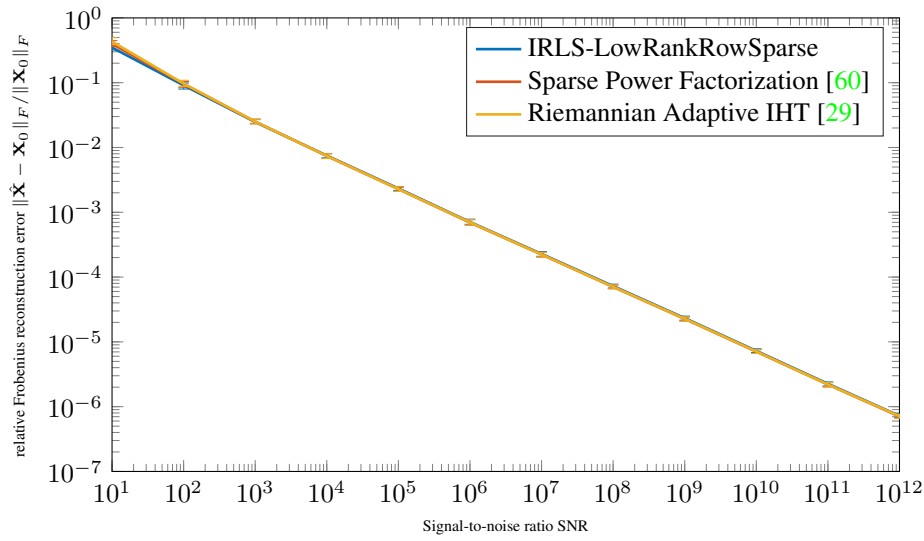

Figure 7: Median relative Frobenius reconstruction errors of different algorithms given noisy Gaussian measurements, $n_1 = 256$, $n_2 = 40$, row-sparsity $s = 40$ and rank $r = 1$, oversampling factor of 3. Error bars correspond to 25% and 75% percentiles.

We observe that the reconstruction error is consistently roughly proportional to the inverse square root of the signal-to-noise ratio, for all three algorithms considered. This suggests that IRLS is as noise robust as comparable algorithms, and expected to be return estimates of the original ground truth that has a reconstruction error that is of the order of the norm of the noise.

## B  Proofs

The following two sections contain the proofs of our main results. Let us begin with some helpful observations.

First note that the low-rank promoting part $W^{lr}_{\mathbf{X}^{(k)},\varepsilon_k} : \mathbb{R}^{n_1 \times n_2} \to \mathbb{R}^{n_1 \times n_2}$ of our weight operator can be re-written as

$$W^{lr}_{\mathbf{X}^{(k)},\varepsilon_k}(\mathbf{Z}) = [\mathbf{U} \quad \mathbf{U}_\perp] \left( \mathbf{H}(\boldsymbol{\sigma}^{(k)}, \varepsilon_k) \circ \left( \begin{bmatrix} \mathbf{U}^* \\ \mathbf{U}_\perp^* \end{bmatrix} \mathbf{Z} [\mathbf{V} \quad \mathbf{V}_\perp] \right) \right) \begin{bmatrix} \mathbf{V}^* \\ \mathbf{V}_\perp^* \end{bmatrix}, \qquad (18)$$

where

$$\mathbf{H}(\boldsymbol{\sigma}^{(k)}, \varepsilon_k) := \left[ \min\left( \varepsilon_k/\sigma_i^{(k)}, 1 \right) \min\left( \varepsilon_k/\sigma_j^{(k)}, 1 \right) \right]_{i,j=1}^{n_1, n_2}$$

$$= \left[ \begin{array}{c|c} \left( \frac{\varepsilon_k^2}{\sigma_i^{(k)} \sigma_j^{(k)}} \right)_{i,j=1}^{r_k} & \left( \frac{\varepsilon_k}{\sigma_i^{(k)}} \right)_{i,j=1}^{r_k, d_2} \\ \hline \left( \frac{\varepsilon_k}{\sigma_j^{(k)}} \right)_{i,j=1}^{d_1, r_k} & \mathbf{1} \end{array} \right] \in \mathbb{R}^{n_1 \times n_2}.$$

Consequently, all weight operators in Definition 2.1 are self-adjoint and positive. Whereas for $\mathbf{W}^{sp}_{\mathbf{X}^{(k)},\delta_k}$ this is obvious, for $W^{lr}_{\mathbf{X}^{(k)},\varepsilon_k}$ it follows from the matrix representation

$$W^{lr}_{\mathbf{X}^{(k)},\varepsilon_k} = \left( [\mathbf{U} \quad \mathbf{U}_\perp] \otimes [\mathbf{V} \quad \mathbf{V}_\perp] \right) \mathbf{D}_{\mathbf{H}(\boldsymbol{\sigma}^{(k)},\varepsilon_k)} \left( [\mathbf{U} \quad \mathbf{U}_\perp]^* \otimes [\mathbf{V} \quad \mathbf{V}_\perp]^* \right)$$

where $\mathbf{D}_{\mathbf{H}(\boldsymbol{\sigma}^{(k)}, \varepsilon_k)} \in \mathbb{R}^{n_1 n_2 \times n_1 n_2}$ is a diagonal matrix with the entries of $\mathbf{H}(\boldsymbol{\sigma}^{(k)}, \varepsilon_k)$, which are all positive, on its diagonal.

## B.1 Proof of Theorem 2.5

Before approaching the proof of Theorem 2.5, let us collect various important observations. In order to keep the presentation concise, we defer part of the proofs to Appendix C.

For a rank-$r$ matrix $\mathbf{Z} = \mathbf{U}\boldsymbol{\Sigma}\mathbf{V}^*$, we define the tangent space of the manifold of rank-$r$ matrices at $\mathbf{Z}$ as

$$T_{\mathbf{U},\mathbf{V}} := \{\mathbf{U}\mathbf{Z}_1^* + \mathbf{Z}_2\mathbf{V}^* \colon \mathbf{Z}_1 \in \mathbb{R}^{n_2 \times r}, \mathbf{Z}_2 \in \mathbb{R}^{n_1 \times r}\}. \tag{19}$$

In a similar manner, we can define for $\mathbf{Z} = \mathbf{U}\boldsymbol{\Sigma}\mathbf{V}^* \in \mathcal{M}_{r,s}$ and $S = \operatorname{supp}(\mathbf{Z}) = \operatorname{supp}(\mathbf{U}) \subset [n_1]$ the tangent space of $\mathcal{M}_r$ restricted to $S$ as

$$T_{\mathbf{U},\mathbf{V},S} := \{\mathbf{U}\mathbf{Z}_1^* + \mathbf{Z}_2\mathbf{V}^* \colon \mathbf{Z}_1 \in \mathbb{R}^{n_2 \times r}, \mathbf{Z}_2 \in \mathbb{R}^{n_1 \times r} \text{ with } \operatorname{supp}(\mathbf{Z}_2) = S\}. \tag{20}$$

As the following lemma shows, orthogonal projections onto the sets $\mathcal{M}_r^{n_1,n_2}$, $\mathcal{N}_s^{n_1,n_2}$, $T_{\mathbf{U},\mathbf{V}}$, and $T_{\mathbf{U},\mathbf{V},S}$ can be efficiently computed.

**Lemma B.1.** *We denote the projection operators onto $\mathcal{M}_r^{n_1,n_2}$ and $\mathcal{N}_s^{n_1,n_2}$ by $\mathsf{T}_r$ and $\mathsf{H}_s$. $\mathsf{T}_r$ truncates a matrix to the $r$ dominant singular values; $\mathsf{H}_s$ sets all but the $s$ in $\ell_2$-norm largest rows to zero. In case of ambiguities (multiple singular values/rows of same magnitude), by convention we choose the $r$ (respectively $s$) with smallest index.*

*For $\mathbf{U}$ and $\mathbf{V}$ fixed, the orthogonal projection onto $T_{\mathbf{U},\mathbf{V}}$ is given by*

$$\mathbb{P}_{\mathbf{U},\mathbf{V}} := \mathbb{P}_{T_{\mathbf{U},\mathbf{V}}}\mathbf{Z} = \mathbf{U}\mathbf{U}^*\mathbf{Z} + \mathbf{Z}\mathbf{V}\mathbf{V}^* - \mathbf{U}\mathbf{U}^*\mathbf{Z}\mathbf{V}\mathbf{V}^*.$$

*For $S \subset [n_1]$ and $\mathbf{U}, \mathbf{V}$ fixed with $\operatorname{supp}(\mathbf{U}) = S$, the orthogonal projection onto $T_{\mathbf{U},\mathbf{V},S}$ is given by*

$$\mathbb{P}_{\mathbf{U},\mathbf{V},S} := \mathbb{P}_{T_{\mathbf{U},\mathbf{V},S}}\mathbf{Z} = \mathbb{P}_S(\mathbf{U}\mathbf{U}^*\mathbf{Z} + \mathbf{Z}\mathbf{V}\mathbf{V}^* - \mathbf{U}\mathbf{U}^*\mathbf{Z}\mathbf{V}\mathbf{V}^*)$$
$$= \mathbf{U}\mathbf{U}^*\mathbf{Z} + \mathbb{P}_S\mathbf{Z}\mathbf{V}\mathbf{V}^* - \mathbf{U}\mathbf{U}^*\mathbf{Z}\mathbf{V}\mathbf{V}^*,$$

*where $\mathbb{P}_S$ projects to the row support $S$, i.e., it sets all rows to zero which are not indexed by $S$.*

The proof of Lemma B.1 is provided in Appendix C.2. In contrast to the above named projections, the projection onto $\mathcal{M}_{r,s}^{n_1,n_2}$ is not tractable. However, [29, Lemma 2.4] shows that locally $\mathbb{P}_{\mathcal{M}_{r,s}}$ can be replaced by the concatenation of $\mathsf{T}_r$ and $\mathsf{H}_s$, i.e., for $\mathbf{Z}_\star \in \mathcal{M}_{r,s}$ and $\mathbf{Z} \approx \mathbf{Z}_\star$, one has that

$$\mathbb{P}_{\mathcal{M}_{r,s}}(\mathbf{Z}) = \mathsf{T}_r(\mathsf{H}_s(\mathbf{Z})).$$

For a matrix $\mathbf{X} \in \mathbb{R}^{n_1 \times n_2}$ and $i \in [n_1]$, we set $\rho_i(\mathbf{X}) = \|(\mathbf{X})_{i',:}\|_2$ where $i'$ is a row index corresponding to the $i$-th largest row of $\mathbf{X}$ in $\ell_2$-norm. More precisely, if $\overline{\mathbf{X}}$ is a decreasing rearrangement of $\mathbf{X}$ with rows ordered by magnitude in $\ell_2$-norm, then $\rho_i(\mathbf{X}) = \|(\overline{\mathbf{X}})_{i,:}\|_2$. As the following lemma shows, the quantity $\rho_s(\mathbf{X})$ determines a local neighborhood of $\mathbf{X}$ on which $\mathsf{H}_s$ preserves the row-support.

**Lemma B.2.** *Let $\mathbf{X} \in \mathbb{R}^{n_1 \times n_2}$ be a matrix with row-support $S \subset [n_1]$ and $|S| = s$. Then, for any $\mathbf{Z} \in \mathbb{R}^{n_1 \times n_2}$ with $\|\mathbf{X} - \mathbf{Z}\|_{\infty,2} := \max_{i \in [n_1]} \|\mathbf{X}_{i,:} - \mathbf{Z}_{i,:}\|_2 \leq \frac{1}{2}\rho_s(\mathbf{X})$ the matrix $\mathsf{H}_s(\mathbf{Z})$ has row-support $S$.*

**Proof:** Note that

$$\max_{i \in [n_1]} \|(\mathbf{Z})_{i,:} - (\mathbf{X})_{i,:}\|_2 \leq \frac{1}{2}\rho_s(\mathbf{X})$$

implies that any non-zero row of $\mathbf{X}$ corresponds to a non-zero row of $\mathsf{H}_s(\mathbf{Z})$ and hence yields the claim. ∎

A first important observation is that if $\mathcal{A}$ has the $(r,s)$-RIP, then the norm of kernel elements of $\mathcal{A}$ is bounded in the following way.

**Lemma B.3.** *If $\mathcal{A}$ has the $(r,s)$-RIP with $\delta \in (0,1)$ and $\mathbf{U} \in \mathbb{R}^{n_1 \times r}, \mathbf{V} \in \mathbb{R}^{n_2 \times r}$ with $\operatorname{supp}(\mathbf{U}) = S$, $|S| \leq s$, then*

$$\|\boldsymbol{\Xi}\|_F \leq \sqrt{1 + \frac{\|\mathcal{A}\|_{2\to2}^2}{(1-\delta)}} \left\|\mathbb{P}_{T_{\mathbf{U},\mathbf{V},S}}^{\perp}(\boldsymbol{\Xi})\right\|_F,$$

*for all $\boldsymbol{\Xi} \in \ker(\mathcal{A})$.*

The proof of Lemma B.3 is presented in Appendix C.3.

**Remark B.4.** *If $\mathcal{A}$ is a Gaussian operator with standard deviation $\sqrt{\frac{1}{m}}$, one has with high probability that $\|\mathcal{A}\|_{2\to2}^2 \approx \frac{n_1 n_2}{m}$.*

We use of the following lemma to characterize the solution of the weighted least squares problem (11). Its proof is analogous to [54, Lemma B.7] and [26, Lemma 5.2].

**Lemma B.5.** *Let $\mathcal{A}: \mathbb{R}^{n_1\times n_2} \to \mathbb{R}^m$ and $\mathbf{y} \in \mathbb{R}^m$. Let $W_{\mathbf{X}^{(k)},\varepsilon_k,\delta_k}: \mathbb{R}^{n_1\times n_2} \to \mathbb{R}^{n_1\times n_2}$ be the weight operator (8) defined based on the information of $\mathbf{X}^{(k)} \in \mathbb{R}^{n_1\times n_2}$. Then the solution of the weighted least squares step (11) of Algorithm 1*

$$\mathbf{X}^{(k+1)} = \underset{\mathcal{A}(\mathbf{X})=\mathbf{y}}{\arg\min} \langle \mathbf{X}, W_{\mathbf{X}^{(k)},\varepsilon_k,\delta_k}(\mathbf{X})\rangle, \tag{21}$$

*is unique and solves (21) if and only if*

$$\mathcal{A}(\mathbf{X}^{(k+1)}) = \mathbf{y} \qquad and \qquad \langle W_{\mathbf{X}^{(k)},\varepsilon_k,\delta_k}(\mathbf{X}^{(k+1)}), \boldsymbol{\Xi}\rangle = 0 \ for\ all\ \boldsymbol{\Xi} \in \ker \mathcal{A}. \tag{22}$$

For any iterate $\mathbf{X}^{(k)}$ of Algorithm 1, we furthermore abbreviate the tangent space (20) of the fixed rank-$r$ manifold $\mathcal{M}_r$ restricted to $S$ at $\mathsf{H}_s(\mathbf{X}^{(k)})$ by

$$T_k = T_{\widetilde{\mathbf{U}},\widetilde{\mathbf{V}},S}, \tag{23}$$

where $\widetilde{\mathbf{U}} \in \mathbb{R}^{n_1\times r}$ and $\widetilde{\mathbf{V}} \in \mathbb{R}^{n_2\times r}$ are matrices with leading[4] $r$ singular vectors of $\mathsf{H}_s(\mathbf{X}^{(k)})$ as columns, and $S \in [n_1]$ is the support set of the $s$ rows of $\mathbf{X}^{(k)}$ with largest $\ell_2$-norm.

The following lemma is the first crucial tool for showing local quadratic convergence of Algorithm 1.

**Lemma B.6.** *Let $\mathbf{X}_\star \in \mathcal{M}_{r,s}$ and let $\mathbf{X}^{(k)}$ be the $k$-th iterate of Algorithm 1 with rank and sparsity parameters $\widetilde{r} = r$ and $\widetilde{s} = s$, let $\delta_k, \varepsilon_k$ be such that $s_k$ and $r_k$ from Definition 2.1 satisfy $s_k \geq s$ and $r_k \geq r$. Assume that there exists a constant $c > 1$ such that*

$$\|\boldsymbol{\Xi}\|_F \leq c \left\|\mathbb{P}_{T_k^\perp}(\boldsymbol{\Xi})\right\|_F \qquad for\ all \quad \boldsymbol{\Xi} \in \ker(\mathcal{A}), \tag{24}$$

*where $T_k = T_{\widetilde{\mathbf{U}},\widetilde{\mathbf{V}},S}$ is as defined in (23) for matrices $\widetilde{\mathbf{U}} \in \mathbb{R}^{n_1\times r}$ and $\widetilde{\mathbf{V}} \in \mathbb{R}^{n_2\times r}$ of leading $r$ left and right singular vectors of $\mathsf{H}_s(\mathbf{X}^{(k)})$ and $S \subset [n_1]$ is the support set of $\mathsf{H}_s(\mathbf{X}^{(k)})$. Assume furthermore that*

$$\|\mathbf{X}^{(k)} - \mathbf{X}_\star\| \leq \min\left\{\frac{1}{2}\rho_s(\mathbf{X}_\star), \min\left\{\frac{1}{48}, \frac{1}{19c}\right\}\sigma_r(\mathbf{X}_\star)\right\}. \tag{25}$$

*Then,*

$$\left\|\mathbf{X}^{(k+1)} - \mathbf{X}_\star\right\| \leq 4c^2 \min\left\{\frac{\sigma_{r+1}(\mathbf{X}^{(k)})}{\varepsilon_k}, \frac{\rho_{s+1}(\mathbf{X}^{(k)})}{\delta_k}\right\}^2$$
$$\cdot \left(\left\|W_{\mathbf{X}^{(k)},\varepsilon_k}^{lr}(\mathbf{X}_\star)\right\|_* + \left\|\mathbf{W}_{\mathbf{X}^{(k)},\delta_k}^{sp} \cdot \mathbf{X}_\star\right\|_{1,2}\right),$$

*where $\|\mathbf{M}\|_{1,2} = \sum_i \|\mathbf{M}_{i,:}\|_2$ denotes the row-sum norm of a matrix $\mathbf{M}$, and $W_{\mathbf{X}^{(k)},\varepsilon_k}^{lr}$ and $\mathbf{W}_{\mathbf{X}^{(k)},\delta_k}^{sp}$ are the weight operators (9) and (10) from Definition 2.1.*

The proof of Lemma B.6 is presented in Appendix C.4.

**Remark B.7.** *By revisiting the proof of Lemma B.6 (omit the bound in (48) and keep the term $\langle\boldsymbol{\Xi}, \bar{W}\boldsymbol{\Xi}\rangle$ until the end), one can show under the same assumptions as in Lemma B.6 that*

$$\|\boldsymbol{\Xi}\|_F^2 \leq 4c^2 \min\left\{\frac{\sigma_{r+1}(\mathbf{X}^{(k)})}{\varepsilon_k}, \frac{\rho_{s+1}(\mathbf{X}^{(k)})}{\delta_k}\right\}^2 \left\langle\boldsymbol{\Xi}, \left(\mathbb{P}_{\mathbf{U},\mathbf{V}}^\perp W_{\mathbf{X}^{(k)},\varepsilon_k}^{lr}\mathbb{P}_{\mathbf{U},\mathbf{V}}^\perp + \mathbb{P}_{S^c}\mathbf{W}_{\mathbf{X}^{(k)},\delta_k}^{sp}\mathbb{P}_{S^c}\right)\boldsymbol{\Xi}\right\rangle,$$

*where $\mathbf{U}$ and $\mathbf{V}$ are containing the left and right singular vectors of $\mathbf{X}^{(k)}$, see Definition 2.1.*

---

[4]As $\widetilde{\mathbf{U}}$ and $\widetilde{\mathbf{V}}$ might not be unique, any set of $r$ leading singular vectors can be chosen in this definition.

The contribution of the norms of the weighted $\mathbf{X}_\star$ terms in Lemma B.6 can be controlled by Lemmas B.8 and B.9 below.

**Lemma B.8.** *Let $W^{lr}_{\mathbf{X}^{(k)},\varepsilon_k} : \mathbb{R}^{n_1 \times n_2} \to \mathbb{R}^{n_1 \times n_2}$ be the rank-based weight operator* (9) *that uses the spectral information of $\mathbf{X}^{(k)}$ and let $\mathbf{X}_\star \in \mathbb{R}^{n_1 \times n_2}$ be a rank-$r$ matrix. Assume that there exists $0 < \zeta < \frac{1}{2}$ such that*

$$\max\{\varepsilon_k, \|\mathbf{X}^{(k)} - \mathbf{X}_\star\|\} \leq \zeta \sigma_r(\mathbf{X}_\star). \tag{26}$$

*Then for each $1 \leq q \leq \infty$,*

$$\left\| W^{lr}_{\mathbf{X}^{(k)},\varepsilon_k}(\mathbf{X}_\star) \right\|_{S_q} \leq \frac{r^{1/q}}{(1-\zeta)\sigma_r(\mathbf{X}_\star)} \left( \frac{1}{1-\zeta}\varepsilon_k^2 + \varepsilon_k K_q \|\mathbf{X}^{(k)} - \mathbf{X}_\star\| + 2\|\mathbf{X}^{(k)} - \mathbf{X}_\star\|^2 \right)$$

*and*

$$\left\| W^{lr}_{\mathbf{X}^{(k)},\varepsilon_k}(\mathbf{X}_\star) \right\|_{S_q} \leq \frac{1}{(1-\zeta)\sigma_r(\mathbf{X}_\star)} \left( \frac{r^{1/q}}{1-\zeta}\varepsilon_k^2 + 2\left\| \mathbf{X}^{(k)} - \mathbf{X}_\star \right\|_{S_q} \left( \varepsilon_k + \|\mathbf{X}^{(k)} - \mathbf{X}_\star\| \right) \right)$$

*where $K_q$ is such that $K_q = 2^{1/q}$ for $1 \leq q \leq 2$ and $4 \leq q$, $K_q = \sqrt{2}$ for $2 < q \leq 4$ and $K_q = 1$ for $q = \infty$.*

**Lemma B.9.** *Let $\mathbf{W}^{sp}_{\mathbf{X}^{(k)},\delta_k} \in \mathbb{R}^{n_1 \times n_1}$ be the row-sparsity-based weight operator* (10) *that uses the current iterate $\mathbf{X}^{(k)}$ with $\delta_k = \min\left(\delta_{k-1}, \rho_{s+1}(\mathbf{X}^{(k)})\right)$ and let $\mathbf{X}_\star \in \mathbb{R}^{n_1 \times n_2}$ be an $s$-row-sparse matrix. Assume that there exists $0 < \zeta < \frac{1}{2}$ such that*

$$\|\mathbf{X}^{(k)} - \mathbf{X}_\star\|_{\infty,2} = \max_{i \in [n_1]} \|(\mathbf{X}^{(k)})_{i,:} - (\mathbf{X}_\star)_{i,:}\|_2 \leq \zeta \rho_s(\mathbf{X}_\star), \tag{27}$$

*where $\rho_s(\mathbf{M})$ denotes the $\ell_2$-norm of the in $\ell_2$-norm $s$-largest row of $\mathbf{M}$. Then*

$$\|\mathbf{W}^{sp}_{\mathbf{X}^{(k)},\delta_k} \cdot \mathbf{X}_\star\|_{1,2} \leq \frac{s\delta_k^2}{(1-\zeta)^2 \rho_s(\mathbf{X}_\star)}$$

Lemma B.8 is a refined version of [54, Lemma B.9] the proof of which we omit here.[5] The proof of Lemma B.9 is provided in Appendix C.5. Finally, the following lemma will allow us to control the decay of the IRLS parameters $\delta_k$ and $\varepsilon_k$.

**Lemma B.10** ([54, Lemma B.5]). *Let $\mathbf{X}_\star \in \mathcal{M}_{r,s}$, assume that $\mathcal{A}$ has the $(r,s)$-RIP with $\delta \in (0,1)$, and let us abbreviate $n = \min\{n_1, n_2\}$.*

*Assume that the $k$-th iterate $\mathbf{X}^{(k)}$ of Algorithm 1 with $\widetilde{r} = r$ and $\widetilde{s} = r$ updates the smoothing parameters in* (12) *such that one of the statements $\varepsilon_k = \sigma_{r+1}(\mathbf{X}^{(k)})$ or $\delta_k = \rho_{s+1}(\mathbf{X}^{(k)})$ is true, and that $r_k \geq r$ and $s_k \geq s$. Furthermore, let*

$$\varepsilon_k \leq \frac{1}{48}\sigma_r(\mathbf{X}_\star)$$

*with $c_{\|\mathcal{A}\|_{2\to2}} = \sqrt{1 + \frac{\|\mathcal{A}\|_{2\to2}^2}{(1-\delta)^2}}$, let $\mathbf{\Xi}^{(k)} := \mathbf{X}^{(k)} - \mathbf{X}_\star$ satisfy*

$$\|\mathbf{\Xi}^{(k)}\| \leq \min\left\{ \frac{1}{2}\rho_s(\mathbf{X}_\star), \min\left\{ \frac{1}{48}, \frac{1}{21c_{\|\mathcal{A}\|_{2\to2}}} \right\}\sigma_r(\mathbf{X}_\star) \right\}. \tag{28}$$

*Then*

$$\|\mathbf{\Xi}^{(k)}\|_F \leq 2\sqrt{2}\sqrt{n}c_{\|\mathcal{A}\|_{2\to2}}\sqrt{4\varepsilon_k^2 + \delta_k^2}.$$

The proof of Lemma B.10 is provided in Appendix C.6. We finally have all the tools to prove Theorem 2.5. Note that (14) implies

$$\|\mathbf{X}^{(k)} - \mathbf{X}_\star\| \leq \min\left\{ \frac{1}{48c_{\|\mathcal{A}\|_{2\to2}}^2} \min\left\{ \frac{\sigma_r(\mathbf{X}_\star)}{r}, \frac{\rho_s(\mathbf{X}_\star)}{s} \right\}, \frac{1}{4\mu\sqrt{5n}c_{\|\mathcal{A}\|_{2\to2}}} \right\}, \tag{29}$$

---

[5]This result is a technical result of an unpublished paper. In this paper, we only use that result as a tool. If the reviewers think that adding the proof is relevant here, we are happy to provide it.

and

$$\varepsilon_k \leq \frac{1}{48}\sigma_r(\mathbf{X}_\star) \tag{30}$$

which we will use in the proof below. The latter follows from the fact that for $\tilde{r} = r$

$$\varepsilon_k = \min\left(\varepsilon_{k-1}, \sigma_{r+1}(\mathbf{X}^{(k)})\right) \leq \sigma_{r+1}(\mathbf{X}^{(k)}) \leq ||\mathbf{X}^{(k)} - \mathbf{X}_*|| \leq \sigma_r(\mathbf{X}_*)/48.$$

**Proof of Theorem 2.5:** First note, that by assumption $\tilde{r} = r$ and $\tilde{s} = s$. Furthermore, since $\frac{1}{48c_{\|\mathcal{A}\|_{2\to 2}^2}} \leq \frac{1}{2}$, the closeness assumption (29) implies that $\mathsf{H}_s(\mathbf{X}^{(k)})$ and $\mathbf{X}_\star$ share the same support due to Lemma B.2. Let $\mathbf{X}^{(k)}$ be the $k$-th iterate of Algorithm 1. Since the operator $\mathcal{A}\colon \mathbb{R}^{n_1\times n_2}\to\mathbb{R}^m$ has the $(r,s)$-RIP with $\delta\in(0,1)$, Lemma B.3 yields for all $\mathbf{U}\in\mathbb{R}^{n_1\times r}, \mathbf{V}\in\mathbb{R}^{n_2\times r}$ with $\mathrm{supp}(\mathbf{U}) = S, |S|\leq s$, that

$$\|\mathbf{\Xi}\|_F \leq c_{\|\mathcal{A}\|_{2\to 2}}\left\|\mathbb{P}_{T_{\mathbf{U},\mathbf{V},S}}^\perp(\mathbf{\Xi})\right\|_F,$$

for any $\mathbf{\Xi}\in\ker(\mathcal{A})$. Furthermore, due to our assumption that $\tilde{s} = s$ and $\tilde{r} = r$, the smoothing parameter update rules in (12), i.e., $\delta_k = \min\left(\delta_{k-1}, \rho_{s+1}(\mathbf{X}^{(k)})\right)$ and $\varepsilon_k = \min\left(\varepsilon_{k-1}, \sigma_{r+1}(\mathbf{X}^{(k)})\right)$, imply that $r_k \geq r$ and $s_k \geq s$ for all $k$. We can thus apply Lemma B.6 for $\mathbf{\Xi}^{(k)} := \mathbf{X}^{(k)} - \mathbf{X}_\star$ (note at this point that (29) implies the closeness assumption (25) of Lemma B.6) and obtain

$$\left\|\mathbf{\Xi}^{(k+1)}\right\| = \left\|\mathbf{X}^{(k+1)} - \mathbf{X}_\star\right\|$$

$$\leq 4c_{\|\mathcal{A}\|_{2\to 2}}^2\min\left\{\frac{\sigma_{r+1}(\mathbf{X}^{(k)})}{\varepsilon_k}, \frac{\rho_{s+1}(\mathbf{X}^{(k)})}{\delta_k}\right\}^2\left(\left\|W_{\mathbf{X}^{(k)},\varepsilon_k}^{lr}(\mathbf{X}_\star)\right\|_* + \left\|\mathbf{W}_{\mathbf{X}^{(k)},\delta_k}^{sp}\cdot\mathbf{X}_\star\right\|_{1,2}\right), \tag{31}$$

where $W_{\mathbf{X}^{(k)},\varepsilon_k}^{lr}\colon\mathbb{R}^{n_1\times n_2}\to\mathbb{R}^{n_1\times n_2}$ is the low-rank promoting part (9) of the weight operator associated to $\mathbf{X}^{(k)}$ and $\mathbf{W}_{\mathbf{X}^{(k)},\delta_k}^{sp}\in\mathbb{R}^{n_1\times n_1}$ the sparsity promoting part (10). Since by assumption

$$\max(\varepsilon_k, \|\mathbf{\Xi}^{(k)}\|) \leq \frac{1}{48}\sigma_r(\mathbf{X}_\star),$$

Lemma B.8 yields

$$\left\|W_{\mathbf{X}^{(k)},\varepsilon_k}^{lr}(\mathbf{X}_\star)\right\|_* \leq 0.995\frac{42}{40\sigma_r(\mathbf{X}_\star)}\left(\varepsilon_k^2 r + 2\varepsilon_k\|\mathbf{X}^{(k)} - \mathbf{X}_\star\| + 2\|\mathbf{X}^{(k)} - \mathbf{X}_\star\|^2\right). \tag{32}$$

Similarly, by assumption

$$\|\mathbf{\Xi}^{(k)}\|_{\infty,2} \leq \|\mathbf{\Xi}^{(k)}\| \leq \frac{1}{48s}\rho_s(\mathbf{X}_\star) \leq \frac{1}{48}\rho_s(\mathbf{X}_\star),$$

such that Lemma B.9 yields

$$\|\mathbf{W}_{\mathbf{X}^{(k)},\delta_k}^{sp}\cdot\mathbf{X}_\star\|_{1,2} \leq 0.995\frac{21s\delta_k^2}{20\rho_s(\mathbf{X}_\star)}. \tag{33}$$

Inserting (32) and (33) into (31) we obtain that

$$\left\|\mathbf{\Xi}^{(k+1)}\right\| \leq 0.995\cdot 4.2c_{\|\mathcal{A}\|_{2\to 2}}^2\min\left\{\frac{\sigma_{r+1}(\mathbf{X}^{(k)})}{\varepsilon_k}, \frac{\rho_{s+1}(\mathbf{X}^{(k)})}{\delta_k}\right\}^2$$

$$\cdot\left(\frac{r}{\sigma_r(\mathbf{X}_\star)}\left(\varepsilon_k^2 + 2\varepsilon_k\|\mathbf{\Xi}^{(k)}\| + 2\|\mathbf{\Xi}^{(k)}\|^2\right) + \frac{2s}{\rho_s(\mathbf{X}_\star)}\delta_k^2\right). \tag{34}$$

Due to the assertion that $r_k \geq r$, it holds that $\varepsilon_k \leq \sigma_{r+1}(\mathbf{X}^{(k)})$. Therefore, Lemma C.3 yields that

$$\left(\varepsilon_k^2 + 2\varepsilon_k\|\mathbf{\Xi}^{(k)}\| + 2\|\mathbf{\Xi}^{(k)}\|^2\right) \leq 5\|\mathbf{\Xi}^{(k)}\|^2.$$

and, since $s_k \geq s$, also that

$$\delta_k^2 \leq \|\mathbf{\Xi}^{(k)}\|_{\infty,2}^2 \leq \|\mathbf{\Xi}^{(k)}\|^2,$$

since $\delta_k \leq \rho_{s+1}(\mathbf{X}^{(k)})$ in this case.

Thus, using the assertion that one of the statements $\varepsilon_k = \sigma_{r+1}(\mathbf{X}^{(k)})$ or $\delta_k = \rho_{s+1}(\mathbf{X}^{(k)})$ is true, we obtain from (34) that

$$\left\|\mathbf{\Xi}^{(k+1)}\right\| \leq 0.995\cdot 4.2c_{\|\mathcal{A}\|_{2\to 2}}^2\cdot\left(\frac{5r}{\sigma_r(\mathbf{X}_\star)} + \frac{2s}{\rho_s(\mathbf{X}_\star)}\right)\left\|\mathbf{\Xi}^{(k)}\right\|^2. \tag{35}$$

For $\left\|\mathbf{\Xi}^{(k)}\right\| < \frac{1}{48}c_{\|\mathcal{A}\|_{2\to2}}^{-2} \min\{\frac{\sigma_r(\mathbf{X}_\star)}{r}, \frac{\rho_s(\mathbf{X}_\star)}{s}\}$ (as implied by (29)), this yields

$$\|\mathbf{\Xi}^{(k+1)}\| < 0.9\|\mathbf{\Xi}^{(k)}\| \tag{36}$$

and the quadratic error decay

$$\|\mathbf{\Xi}^{(k+1)}\| \leq \mu\|\mathbf{\Xi}^{(k)}\|^2$$

if we define $\mu = 4.179c_{\|\mathcal{A}\|_{2\to2}}^2 \left(\frac{5r}{\sigma_r(\mathbf{X}_\star)} + \frac{2s}{\rho_s(\mathbf{X}_\star)}\right)$.

To show the remaining statement, we need to argue that the assertions of Theorem 2.5 are satisfied not only for $k$, but for any $k + \ell$ with $\ell \geq 1$. For this, it is sufficient to show that

1. $r_{k+1} \geq r$,
2. $s_{k+1} \geq s$,
3. $\varepsilon_{k+1} \leq \frac{1}{48}\sigma_r(\mathbf{X}_\star)$,
4. (29) holds for $\mathbf{X}^{(k+1)}$, and that
5. one of the statements $\varepsilon_{k+1} = \sigma_{r+1}(\mathbf{X}^{(k+1)})$ or $\delta_{k+1} = \rho_{s+1}(\mathbf{X}^{(k+1)})$ is true,

as in this case, $\mathbf{X}^{(k+\ell)} \overset{\ell\to\infty}{\to} \mathbf{X}_\star$ follows by induction due to successive application of (36).

For 1. and 2., we see that this follows from the smoothing parameter update rules (12) which imply that $\varepsilon_{k+1} \leq \sigma_{r+1}(\mathbf{X}^{(k+1)})$ and $\delta_{k+1} \leq \rho_{s+1}(\mathbf{X}^{(k+1)})$.

3. follows from (30) and the fact that due to (12), $(\varepsilon_k)_{k\geq1}$ is non-increasing. 4. is satisfied due to (36) and (29). To show 5., we note that due to (29), the assertion (28) is satisfied, and therefore it follows from (35) and Lemma B.10 that

$$\left\|\mathbf{\Xi}^{(k+1)}\right\| \leq 4.179c_{\|\mathcal{A}\|_{2\to2}}^2 \left(\frac{5r}{\sigma_r(\mathbf{X}_\star)} + \frac{2s}{\rho_s(\mathbf{X}_\star)}\right) \left\|\mathbf{\Xi}^{(k)}\right\| \cdot 2\sqrt{2}\sqrt{n}c_{\|\mathcal{A}\|_{2\to2}} \sqrt{4\varepsilon_k^2 + \delta_k^2}.$$

We now distinguish the case (i) $\delta_k < \varepsilon_k$ and the case (ii) $\delta_k \geq \varepsilon_k$.

In case (i), it holds that

$$\sigma_{r+1}(\mathbf{X}^{(k+1)}) \leq \left\|\mathbf{\Xi}^{(k+1)}\right\| \leq 4.179c_{\|\mathcal{A}\|_{2\to2}}^2 \left(\frac{5r}{\sigma_r(\mathbf{X}_\star)} + \frac{2s}{\rho_s(\mathbf{X}_\star)}\right) 2\sqrt{10n}c_{\|\mathcal{A}\|_{2\to2}} \left\|\mathbf{\Xi}^{(k)}\right\| \varepsilon_k$$

$$= \mu 2\sqrt{10n}c_{\|\mathcal{A}\|_{2\to2}} \left\|\mathbf{\Xi}^{(k)}\right\| \varepsilon_k$$

$$< \varepsilon_k,$$

where the last inequality holds since by (29) the $k$-th iterate $\mathbf{X}^{(k)}$ additionally satisfies

$$\|\mathbf{X}^{(k)} - \mathbf{X}_\star\| < \frac{1}{2\mu\sqrt{10}c_{\|\mathcal{A}\|_{2\to2}}}. \tag{37}$$

In this case, due to the smoothing parameter update rule (12), we have that $\varepsilon_{k+1} = \sigma_{r+1}(\mathbf{X}^{(k+1)})$.

In case (ii), we have likewise that

$$\rho_{s+1}(\mathbf{X}^{(k+1)}) \leq \left\|\mathbf{\Xi}^{(k+1)}\right\| \leq \mu 2\sqrt{10n}c_{\|\mathcal{A}\|_{2\to2}} \left\|\mathbf{\Xi}^{(k)}\right\| \delta_k < \delta_k,$$

due to (35), Lemma B.10, and (37). Hence, $\delta_{k+1} = \rho_{s+1}(\mathbf{X}^{(k+1)})$ which shows the remaining statement 5. and concludes the proof of Theorem 2.5. ∎

## B.2    Proof of Theorem 2.6

1.) Let $\varepsilon, \delta > 0$ be arbitrary. Due to the additive structure of $\mathcal{F}_{\varepsilon,\delta}(\cdot)$, cf. (4), it is sufficient to establish that

$$\mathcal{F}_{sp,\delta}(\mathbf{Z}) \leq \mathcal{Q}_{sp,\delta}(\mathbf{Z}|\mathbf{X}) = \mathcal{F}_{sp,\delta}(\mathbf{X}) + \langle\nabla\mathcal{F}_{sp,\delta}(\mathbf{X}), \mathbf{Z} - \mathbf{X}\rangle + \frac{1}{2}\langle\mathbf{Z} - \mathbf{X}, \mathbf{W}_{\mathbf{X},\delta}^{sp}(\mathbf{Z} - \mathbf{X})\rangle \tag{38}$$

for any $\mathbf{Z}, \mathbf{X} \in \mathbb{R}^{n_1\times n_2}$, where $\mathbf{W}_{\mathbf{X},\delta}^{sp} : \mathbb{R}^{n_1\times n_2} \to \mathbb{R}^{n_1\times n_2}$ is defined analogously to (10) and

$$\mathcal{F}_{lr,\varepsilon}(\mathbf{Z}) \leq \mathcal{Q}_{lr,\varepsilon}(\mathbf{Z}|\mathbf{X}) = \mathcal{F}_{lr,\varepsilon}(\mathbf{X}) + \langle\nabla\mathcal{F}_{lr,\varepsilon}(\mathbf{X}), \mathbf{Z} - \mathbf{X}\rangle + \frac{1}{2}\langle\mathbf{Z} - \mathbf{X}, W_{\mathbf{X},\varepsilon}^{lr}(\mathbf{Z} - \mathbf{X})\rangle, \tag{39}$$

for any $\mathbf{Z}, \mathbf{X} \in \mathbb{R}^{n_1\times n_2}$, where $W_{\mathbf{X},\varepsilon}^{lr} : \mathbb{R}^{n_1\times n_2} \to \mathbb{R}^{n_1\times n_2}$ is defined analogously to (9).

The argument for (38) is standard in the IRLS literature [2, 73, 76] and is based on the facts that both $\mathcal{Q}_{sp,\delta}(\mathbf{Z}|\mathbf{X})$ and $\mathcal{F}_{sp,\delta}(\mathbf{Z})$ are row-wise separable, and that $t \mapsto f_{\sqrt{\tau}}(\sqrt{t})$ is concave and therefore majorized by its linearization: indeed, let $g_\tau : \mathbb{R} \to \mathbb{R}$ be such that

$$g_\tau(t) := \begin{cases} \frac{1}{2}\tau \log(e|t|/\tau), & \text{if } |t| > \tau, \\ \frac{1}{2}|t|, & \text{if } |t| \leq \tau. \end{cases}$$

The function $g_\tau(\cdot)$ is continuously differentiable with derivative $g'_\tau(t) = \frac{\tau}{2\max(|t|,\tau)}\text{sign}(t)$ and furthermore, concave restricted to the non-negative domain $\mathbb{R}_{\geq 0}$.

Therefore, it holds for any $t, t' \in \mathbb{R}_{\geq 0}$ that

$$g_\tau(t) \leq g_\tau(t') + g'_\tau(t')(t - t').$$

We recall the definition $f_\tau(t) = \frac{1}{2}\tau^2 \log(et^2/\tau^2)$ for $|t| > \tau$ and $f_\tau(t) = \frac{1}{2}t^2$ for $|t| \leq \tau$ from (2) with derivative $f'_\tau(t) = \frac{\max(t^2,\tau^2)t}{t^2} = \frac{\tau^2 t}{\max(t^2,\tau^2)}$. Thus, for any $x, z \in \mathbb{R}$, it follows that

$$f_\tau(z) = g_{\tau^2}(z^2) \leq g_{\tau^2}(x^2) + g'_{\tau^2}(x^2)(z^2 - x^2)$$
$$= f_\tau(x) + \frac{\tau^2}{2\max(x^2,\tau^2)}(z^2 - x^2),$$

and inserting $\tau = \delta$, $z = \|\mathbf{Z}_{i,:}\|_2$, $x = \|\mathbf{X}_{i,:}\|_2$ and summing over $i = 1, \dots n_1$ implies that

$$\mathcal{F}_{sp,\delta}(\mathbf{Z}) = \sum_{i=1}^{n_1} f_\delta(\|\mathbf{Z}_{i,:}\|_2) \leq \mathcal{F}_{sp,\delta}(\mathbf{X}) + \sum_{i=1}^{n_1} \frac{\delta^2}{2\max(\|\mathbf{X}_{i,:}\|_2^2, \delta^2)}(\|\mathbf{Z}_{i,:}\|_2^2 - \|\mathbf{X}_{i,:}\|_2^2)$$

$$= \mathcal{F}_{sp,\delta}(\mathbf{X}) + \sum_{i=1}^{n_1} \frac{\delta^2}{\max(\|\mathbf{X}_{i,:}\|_2^2, \delta^2)}\langle \mathbf{X}_{i,:}, \mathbf{Z}_{i,:} - \mathbf{X}_{i,:}\rangle + \frac{1}{2}\sum_{i=1}^{n_1} \frac{\|\mathbf{Z}_{i,:} - \mathbf{X}_{i,:}\|_2^2}{\max(\|\mathbf{X}_{i,:}\|_2^2/\delta^2, 1)}$$

From the chain rule, it follows that for all $i = 1, \dots, n_1$ for which $\mathbf{X}_{i,:} \neq 0$,

$$\frac{d}{d\mathbf{X}_{i,:}} f_\delta(\|\mathbf{X}_{i,:}\|_2) = f'_\delta(\|\mathbf{X}_{i,:}\|_2)\frac{d\|\mathbf{X}_{i,:}\|_2}{d\mathbf{X}_{i,:}} = \frac{\delta^2\|\mathbf{X}_{i,:}\|_2}{\max(\|\mathbf{X}_{i,:}\|_2^2, \delta^2)}\frac{\mathbf{X}_{i,:}}{\|\mathbf{X}_{i,:}\|_2}, = \frac{\delta^2\mathbf{X}_{i,:}}{\max(\|\mathbf{X}_{i,:}\|_2^2, \delta^2)} \tag{40}$$

and therefore

$$\mathcal{F}_{sp,\delta}(\mathbf{Z}) \leq \mathcal{F}_{sp,\delta}(\mathbf{X}) + \langle \nabla\mathcal{F}_{sp,\delta}(\mathbf{X}), \mathbf{Z} - \mathbf{X}\rangle + \frac{1}{2}\langle \mathbf{Z} - \mathbf{X}, \mathbf{W}_{\mathbf{X},\delta}^{sp}(\mathbf{Z} - \mathbf{X})\rangle$$

which shows the majorization of (38), recalling the definition $\mathbf{W}_{\mathbf{X},\delta}^{sp} = \text{diag}\left(\max\left(\|\mathbf{X}_{i,:}\|_2^2/\delta^2, 1)_{i=1}^{d_1}\right)^{-1}\right)$ of (10).

The majorization of (39) is non-trivial but follows in a straightforward way from [53, Theorem 2.4] as the objective $\mathcal{F}_{lr,\varepsilon}(\mathbf{Z})$ corresponds to the one of [53, Theorem 2.4] up to a multiplicative factor of $\varepsilon^2$ and constant additive factors, and since the weight operator $W_{\mathbf{X},\varepsilon}^{lr}$ corresponds to the weight operator used in [53, Chapter 2] for $p = 0$.

2.) Due to the definition (3) of $\mathcal{F}_{sp,\delta_k}(\cdot)$ and the derivative computation of (40), we observe that

$$\nabla\mathcal{F}_{sp,\delta_k}(\mathbf{X}^{(k)}) = \text{diag}\left(\left(\max\left(\|(\mathbf{X}^{(k)})_{i,:}\|_2^2/\delta_k^2, 1\right)^{-1}\right)_{i=1}^{d_1}\right)\mathbf{X}^{(k)} = \mathbf{W}_{\mathbf{X}^{(k)},\delta_k}^{sp} \cdot \mathbf{X}^{(k)},$$

comparing the resulting term with the definition of (10) of $\mathbf{W}_{\mathbf{X}^{(k)},\delta_k}^{sp}$. Furthermore, an analogue equality follows from the the formula

$$\nabla\mathcal{F}_{lr,\varepsilon_k}(\mathbf{X}^{(k)}) = [\mathbf{U} \quad \mathbf{U}_\perp]\text{diag}\left(\left(\sigma_i^{(k)}\max\left((\sigma_i^{(k)})^2/\varepsilon_k^2, 1\right)^{-1}\right)_{i=1}^d\right)\begin{bmatrix}\mathbf{V}^* \\ \mathbf{V}_\perp^*\end{bmatrix}$$

with $\sigma_i^{(k)} = \sigma_i(\mathbf{X}^{(k)})$ for any $i \leq d$, which is a direct consequence from the calculus of spectral functions Lemma C.1, and inserting into the low-rank promoting weight operator formula (9)

$$W_{\mathbf{X}^{(k)},\varepsilon_k}^{lr}(\mathbf{X}^{(k)}) = [\mathbf{U} \quad \mathbf{U}_\perp]\mathbf{\Sigma}_{\varepsilon_k}^{-1}\text{diag}\left(\left(\sigma_i^{(k)}\right)_{i=1}^d\right)\mathbf{\Sigma}_{\varepsilon_k}^{-1}\begin{bmatrix}\mathbf{V}^* \\ \mathbf{V}_\perp^*\end{bmatrix} = \nabla\mathcal{F}_{lr,\varepsilon_k}(\mathbf{X}^{(k)})$$

Inserting $\nabla \mathcal{F}_{sp,\delta_k}(\mathbf{X}^{(k)}) = \mathbf{W}^{sp}_{\mathbf{X}^{(k)},\delta_k} \cdot \mathbf{X}^{(k)}$ and $\nabla \mathcal{F}_{lr,\varepsilon_k}(\mathbf{X}^{(k)}) = W^{lr}_{\mathbf{X}^{(k)},\varepsilon_k}(\mathbf{X}^{(k)})$ into the definitions of $\mathcal{Q}_{lr,\varepsilon_k}(\mathbf{Z}|\mathbf{X}^{(k)})$ and $\mathcal{Q}_{sp,\delta_k}(\mathbf{Z}|\mathbf{X}^{(k)})$, we see that it holds that

$$\mathcal{Q}_{lr,\varepsilon_k}(\mathbf{Z}|\mathbf{X}^{(k)}) = \mathcal{F}_{lr,\varepsilon_k}(\mathbf{X}^{(k)}) + \frac{1}{2}\left(\langle \mathbf{Z}, W^{lr}_{\mathbf{X}^{(k)},\varepsilon_k}(\mathbf{Z})\rangle - \langle \mathbf{X}^{(k)}, W^{lr}_{\mathbf{X}^{(k)},\varepsilon_k}(\mathbf{X}^{(k)})\rangle\right)$$

and

$$\mathcal{Q}_{sp,\delta_k}(\mathbf{Z}|\mathbf{X}^{(k)}) = \mathcal{F}_{sp,\delta_k}(\mathbf{X}^{(k)}) + \frac{1}{2}\left(\langle \mathbf{Z}, \mathbf{W}^{sp}_{\mathbf{X}^{(k)},\delta_k}\mathbf{Z}\rangle - \langle \mathbf{X}^{(k)}, \mathbf{W}^{sp}_{\mathbf{X}^{(k)},\delta_k}\mathbf{X}^{(k)}\rangle\right).$$

Therefore, we see that the weighted least squares solution $\mathbf{X}^{(k+1)}$ of (11) for $k+1$ coincides with the minimizer of

$$\min_{\mathbf{Z}:\mathcal{A}(\mathbf{Z})=\mathbf{y}}\left[\mathcal{Q}_{lr,\varepsilon_k}(\mathbf{Z}|\mathbf{X}^{(k)}) + \mathcal{Q}_{sp,\delta_k}(\mathbf{Z}|\mathbf{X}^{(k)})\right]$$

$$= \min_{\mathbf{Z}:\mathcal{A}(\mathbf{Z})=\mathbf{y}}\left[\mathcal{F}_{lr,\varepsilon_k}(\mathbf{X}^{(k)}) + \mathcal{F}_{sp,\delta_k}(\mathbf{X}^{(k)}) \right. \tag{41}$$

$$\left. + \frac{1}{2}\left(\langle \mathbf{Z}, W_{\mathbf{X}^{(k)},\varepsilon_k,\delta_k}(\mathbf{Z})\rangle - \langle \mathbf{X}^{(k)}, W_{\mathbf{X}^{(k)},\varepsilon_k,\delta_k}(\mathbf{X}^{(k)})\rangle\right)\right]$$

with the weight operator $W_{\mathbf{X}^{(k)},\varepsilon_k,\delta_k}$ of (8), which implies that

$$\mathcal{Q}_{lr,\varepsilon_k}(\mathbf{X}^{(k+1)}|\mathbf{X}^{(k)}) + \mathcal{Q}_{sp,\delta_k}(\mathbf{X}^{(k+1)}|\mathbf{X}^{(k)}) \leq \mathcal{Q}_{lr,\varepsilon_k}(\mathbf{X}^{(k)}|\mathbf{X}^{(k)}) + \mathcal{Q}_{sp,\delta_k}(\mathbf{X}^{(k)}|\mathbf{X}^{(k)}). \tag{42}$$

Using the majorization (16) established in Statement 1 of Theorem 2.6 and (42), it follows that

$$\mathcal{F}_{\varepsilon_k,\delta_k}(\mathbf{X}^{(k+1)}) \leq \mathcal{Q}_{lr,\varepsilon_k}(\mathbf{X}^{(k+1)}|\mathbf{X}^{(k)}) + \mathcal{Q}_{sp,\delta_k}(\mathbf{X}^{(k+1)}|\mathbf{X}^{(k)})$$

$$\leq \mathcal{Q}_{lr,\varepsilon_k}(\mathbf{X}^{(k)}|\mathbf{X}^{(k)}) + \mathcal{Q}_{sp,\delta_k}(\mathbf{X}^{(k)}|\mathbf{X}^{(k)}) \tag{43}$$

$$= \mathcal{F}_{lr,\varepsilon_k}(\mathbf{X}^{(k)}) + \mathcal{F}_{sp,\delta_k}(\mathbf{X}^{(k)}) = \mathcal{F}_{\varepsilon_k,\delta_k}(\mathbf{X}^{(k)}),$$

using in the third line that $\mathcal{Q}_{lr,\varepsilon_k}(\mathbf{X}^{(k)}|\mathbf{X}^{(k)}) = \mathcal{F}_{lr,\varepsilon_k}(\mathbf{X}^{(k)})$ and $\mathcal{Q}_{sp,\delta_k}(\mathbf{X}^{(k)}|\mathbf{X}^{(k)}) = \mathcal{F}_{sp,\delta_k}(\mathbf{X}^{(k)})$.

To conclude, it suffices to show that $\varepsilon \mapsto \mathcal{F}_{\varepsilon,\delta_k}(\mathbf{X}^{(k+1)})$ and $\delta \mapsto \mathcal{F}_{\varepsilon_k,\delta}(\mathbf{X}^{(k+1)})$ are non-decreasing functions, since (43) then extends to

$$\mathcal{F}_{\varepsilon_{k+1},\delta_{k+1}}(\mathbf{X}^{(k+1)}) \leq \mathcal{F}_{\varepsilon_k,\delta_{k+1}}(\mathbf{X}^{(k+1)}) \leq \mathcal{F}_{\varepsilon_k,\delta_k}(\mathbf{X}^{(k+1)}) \leq \mathcal{F}_{\varepsilon_k,\delta_k}(\mathbf{X}^{(k)}),$$

where we used that the sequences $\varepsilon_k$ and $\delta_k$ defined in Algorithm 1 are decreasing. So let us prove this last claim. We define for $t \in \mathbb{R}$ the function $h_t : \mathbb{R}_{>0} \to \mathbb{R}$ such that $h_t(\tau) = f_\tau(t)$, i.e.,

$$h_t(\tau) = \begin{cases} \frac{1}{2}t^2, & \text{if } \tau \geq |t|, \\ \frac{1}{2}\tau^2 \log(et^2/\tau^2), & \text{if } \tau < |t|. \end{cases}$$

This function is continuously differentiable with $h'_t(\tau) = 0$ for all $\tau > |t|$ and

$$h'_t(\tau) = \tau\left(\log(et^2/\tau^2) - 1\right)$$

for $\tau < |t|$, which implies that $h'_t(\tau) \geq 0$ for all $\tau \geq 0$ and thus shows that $\varepsilon \mapsto \mathcal{F}_{\varepsilon,\delta_k}(\mathbf{X}^{(k+1)})$ and $\delta \mapsto \mathcal{F}_{\varepsilon_k,\delta}(\mathbf{X}^{(k+1)})$ are non-decreasing functions due to the additive structure of $\mathcal{F}_{\varepsilon,\delta}(\mathbf{X}^{(k+1)})$ and (4).

3.) First, we argue that $(\mathbf{X}^{(k)})_{k\geq 1}$ is a bounded sequence: Indeed, if $\overline{\varepsilon} := \lim_{k\to\infty} \varepsilon_k > 0$ and $\overline{\delta} := \lim_{k\to\infty} \delta_k > 0$, we note that

$$\frac{1}{2}\overline{\varepsilon}^2 \log(e\|\mathbf{X}^{(k)}\|^2/\overline{\varepsilon}^2) + \frac{1}{2}\overline{\delta}^2 \log(e \max_i \|\mathbf{X}^{(k)}\|_{\infty,2}/\overline{\delta}^2)$$

$$= \frac{1}{2}\overline{\varepsilon}^2 \log(e\sigma_1^2(\mathbf{X}^{(k)})/\overline{\varepsilon}^2) + \frac{1}{2}\overline{\delta}^2 \log(e \max_i \|\mathbf{X}^{(k)}_{i,:}\|_2/\overline{\delta}^2)$$

$$\leq \frac{1}{2}\varepsilon_k^2 \log(e\sigma_1^2(\mathbf{X}^{(k)})/\varepsilon_k^2) + \frac{1}{2}\delta_k^2 \log(e \max_i \|\mathbf{X}^{(k)}_{i,:}\|_2/\delta_k^2)$$

$$\leq \mathcal{F}_{lr,\varepsilon_k}(\mathbf{X}^{(k)}) + \mathcal{F}_{sp,\delta_k}(\mathbf{X}^{(k)}) = \mathcal{F}_{\varepsilon_k,\delta_k}(\mathbf{X}^{(k)}) \leq \mathcal{F}_{\varepsilon_1,\delta_1}(\mathbf{X}^{(1)})$$

$$\leq \frac{1}{2}\min(d_1,d_2)\sigma_1^2(\mathbf{X}^{(1)}) + \frac{1}{2}d_1 \max_i \|\mathbf{X}^{(1)}_{i,:}\|_2^2 =: C_{\mathbf{X}^{(1)}},$$

which implies that $\{\|\mathbf{X}^{(k)}\|\}_{k\geq 1}$ is bounded by a constant that depends on $C_{\mathbf{X}^{(1)}}$.

Furthermore, we note that the optimality condition of (11) (see Lemma B.5) implies that $\mathbf{X}^{(k+1)}$ satisfies
$$\langle W_{\mathbf{X}^{(k)},\varepsilon_k,\delta_k}(\mathbf{X}^{(k+1)}),\boldsymbol{\Xi}\rangle = 0 \text{ for all } \boldsymbol{\Xi} \in \ker \mathcal{A} \text{ and } \mathcal{A}(\mathbf{X}^{(k+1)}) = \mathbf{y}.$$

Choosing $\boldsymbol{\Xi} = \mathbf{X}^{(k+1)} - \mathbf{X}^{(k)}$ and using the notation $W^{(k)} = W_{\mathbf{X}^{(k)},\varepsilon_k,\delta_k}$ we see that

$$
\begin{aligned}
&\langle \mathbf{X}^{(k+1)}, W^{(k)}(\mathbf{X}^{(k+1)})\rangle - \langle \mathbf{X}^{(k)}, W^{(k)}(\mathbf{X}^{(k)})\rangle \\
&= \langle \mathbf{X}^{(k+1)}, W^{(k)}(\mathbf{X}^{(k+1)})\rangle - \langle \mathbf{X}^{(k)}, W^{(k)}(\mathbf{X}^{(k)})\rangle - 2\langle W^{(k)}(\mathbf{X}^{(k+1)}), \mathbf{X}^{(k+1)} - \mathbf{X}^{(k)}\rangle \\
&= -\left(\langle \mathbf{X}^{(k+1)}, W^{(k)}(\mathbf{X}^{(k+1)})\rangle - 2\langle W^{(k)}(\mathbf{X}^{(k)}), \mathbf{X}^{(k+1)}\rangle + \langle \mathbf{X}^{(k)}, W^{(k)}(\mathbf{X}^{(k)})\rangle\right) \\
&= -\langle (\mathbf{X}^{(k+1)} - \mathbf{X}^{(k)}), W^{(k)}(\mathbf{X}^{(k+1)} - \mathbf{X}^{(k)})\rangle.
\end{aligned}
\tag{44}
$$

Due to the definition of $W^{(k)}$, we note that its smallest singular value (interpreted as matrix operator) can be lower bounded by

$$
\begin{aligned}
\sigma_{\min}(W^{(k)}) &\geq \sigma_{\min}\left(W^{lr}_{\mathbf{X}^{(k)},\varepsilon_k}\right) + \sigma_{\min}\left(\mathbf{W}^{sp}_{\mathbf{X}^{(k)},\delta_k}\right) \geq \delta_k^2/\max_i \|\mathbf{X}^{(k)}_{i,:}\|_2^2 + \varepsilon_k^2/\sigma_1^2(\mathbf{X}^{(k)}) \\
&\geq \overline{\delta}^2/c_{\mathrm{sp},\mathbf{X}^{(1)}} + \overline{\varepsilon}^2/c_{\mathrm{lr},\mathbf{X}^{(1)}},
\end{aligned}
$$

where $c_{\mathrm{sp},\mathbf{X}^{(1)}}$ and $c_{\mathrm{lr},\mathbf{X}^{(1)}}$ are constants that satisfy $c_{\mathrm{sp},\mathbf{X}^{(1)}} \leq \overline{\delta}^2 \exp(C_{\mathbf{X}^{(1)}}/\overline{\delta}^2 - 1)$ and $c_{\mathrm{lr},\mathbf{X}^{(1)}} \leq \overline{\varepsilon}^2 \exp(C_{\mathbf{X}^{(1)}}/\overline{\varepsilon}^2 - 1)$.

Combining this with (44), the monotonicity according to Statement 2 of Theorem 2.6, and (41), it follows that

$$
\begin{aligned}
\mathcal{F}_{\varepsilon_k,\delta_k}(\mathbf{X}^{(k)}) - \mathcal{F}_{\varepsilon_{k+1},\delta_{k+1}}(\mathbf{X}^{(k+1)}) &\geq \frac{1}{2}\langle (\mathbf{X}^{(k)} - \mathbf{X}^{(k+1)}), W^{(k)}(\mathbf{X}^{(k)} - \mathbf{X}^{(k+1)})\rangle \\
&\geq \frac{1}{2}\left(\overline{\delta}^2/c_{\mathrm{sp},\mathbf{X}^{(1)}} + \overline{\varepsilon}^2/c_{\mathrm{lr},\mathbf{X}^{(1)}}\right)\|\mathbf{X}^{(k+1)} - \mathbf{X}^{(k)}\|_F^2.
\end{aligned}
$$

Summing over all $k$, this implies that $\lim_{k\to\infty}\|\mathbf{X}^{(k+1)} - \mathbf{X}^{(k)}\|_F = 0$.

Since $(\mathbf{X}^{(k)})_{k\geq 1}$ is bounded, each subsequence of $(\mathbf{X}^{(k)})_{k\geq 1}$ has a convergent subsequence. Let $(\mathbf{X}^{(k_\ell)})_{\ell\geq 1}$ be such a sequence with $\lim_{\ell\to\infty}\mathbf{X}^{(k_\ell)} = \bar{\mathbf{X}}$, i.e., $\bar{\mathbf{X}}$ is an accumulation point of the sequence. As the weight operator $W^{(k_\ell)}$ depends continuously on $\mathbf{X}^{(k_\ell)}$, there exists a weight operator $\bar{W} : \mathbb{R}^{n_1\times n_2} \to \mathbb{R}^{n_1\times n_2}$ such that $\bar{W} = \lim_{\ell\to\infty} W^{(k_\ell)}$.

Since $\lim_{k\to\infty}\|\mathbf{X}^{(k+1)} - \mathbf{X}^{(k)}\|_F = 0$, it also holds that $\mathbf{X}^{(k_\ell+1)} \to \bar{\mathbf{X}}$ and therefore
$$\langle \nabla \mathcal{F}_{\overline{\varepsilon},\overline{\delta}}(\bar{\mathbf{X}}), \boldsymbol{\Xi}\rangle = \langle \bar{W}(\bar{\mathbf{X}}), \boldsymbol{\Xi}\rangle = \lim_{\ell\to\infty}\langle W^{(k_\ell)}(\mathbf{X}^{(k_\ell+1)}), \boldsymbol{\Xi}\rangle = 0$$

for all $\boldsymbol{\Xi} \in \ker \mathcal{A}$. The statement is shown as this is equivalent to $\bar{\mathbf{X}}$ being a stationary point of $\mathcal{F}_{\overline{\varepsilon},\overline{\delta}}(\cdot)$ subject to the linear constraint $\{\mathbf{Z} \in \mathbb{R}^{n_1\times n_2} : \mathcal{A}(\mathbf{Z}) = \mathbf{y}\}$.

# C  Technical addendum

## C.1  Auxiliary Results

In the proof of Theorem 2.6, we use the following result about the calculus of spectral functions.

**Lemma C.1** ([63],[35, Proposition 7.4]). *Let $F : \mathbb{R}^{d_1\times d_2} \to \mathbb{R}$ be a spectral function $F = f \circ \sigma$ with an associated function $f : \mathbb{R}^d \to \mathbb{R}$ that is absolutely permutation symmetric. Then, $F$ is differentiable at $\mathbf{X} \in \mathbb{R}^{d_1\times d_2}$ if and only if $f$ is differentiable at $\sigma(\mathbf{X}) \in \mathbb{R}^d$.*

*In this case, the gradient $\nabla F$ of $F$ at $\mathbf{X}$ is given by*
$$\nabla F(\mathbf{X}) = \mathbf{U}\mathrm{diag}(\nabla f(\sigma(\mathbf{X}))\mathbf{V}^*$$

*if $\mathbf{X} = \mathbf{U}\mathrm{diag}(\sigma(\mathbf{X}))\mathbf{V}^*$ for unitary matrices $\mathbf{U} \in \mathbb{R}^{d_1\times d_1}$ and $\mathbf{V} \in \mathbb{R}^{d_2\times d_2}$.*[6]

---

[6]Here, for $\mathbf{v} \in \mathbb{R}^{\min(d_1,d_2)}$, $\mathrm{diag}(\mathbf{v}) \in \mathbb{R}^{d_1\times d_2}$ refers to the matrix with diagonal elements $\mathbf{v}_i$ on its main diagonal and zeros elsewhere.

## C.2 Proof of Lemma B.1

The projection operators for $\mathcal{M}_r^{n_1,n_2}$, $\mathcal{N}_s^{n_1,n_2}$, and $T_{\mathbf{U},\mathbf{V}}$ are well-known, see e.g. [8]. To see the final statement assume that $\mathbf{U}$ has row-support $S$ and note that $\mathbb{P}_{\mathbf{U},\mathbf{V},S}$ is idempotent, i.e.,

$$
\begin{aligned}
&\mathbb{P}_{\mathbf{U},\mathbf{V},S}\mathbb{P}_{\mathbf{U},\mathbf{V},S}\mathbf{Z} \\
&= \mathbf{U}\mathbf{U}^*(\mathbf{U}\mathbf{U}^*\mathbf{Z} + \mathbb{P}_S\mathbf{Z}\mathbf{V}\mathbf{V}^* - \mathbf{U}\mathbf{U}^*\mathbf{Z}\mathbf{V}\mathbf{V}^*) + \mathbb{P}_S(\mathbf{U}\mathbf{U}^*\mathbf{Z} + \mathbb{P}_S\mathbf{Z}\mathbf{V}\mathbf{V}^* - \mathbf{U}\mathbf{U}^*\mathbf{Z}\mathbf{V}\mathbf{V}^*)\mathbf{V}\mathbf{V}^* \\
&\quad - \mathbf{U}\mathbf{U}^*(\mathbf{U}\mathbf{U}^*\mathbf{Z} + \mathbb{P}_S\mathbf{Z}\mathbf{V}\mathbf{V}^* - \mathbf{U}\mathbf{U}^*\mathbf{Z}\mathbf{V}\mathbf{V}^*)\mathbf{V}\mathbf{V}^* \\
&= \mathbf{U}\mathbf{U}^*\mathbf{Z} + \mathbf{U}\mathbf{U}^*\mathbb{P}_S\mathbf{Z}\mathbf{V}\mathbf{V}^* - \mathbf{U}\mathbf{U}^*\mathbf{Z}\mathbf{V}\mathbf{V}^* + \mathbf{U}\mathbf{U}^*\mathbf{Z}\mathbf{V}\mathbf{V}^* + \mathbb{P}_S\mathbf{Z}\mathbf{V}\mathbf{V}^* - \mathbf{U}\mathbf{U}^*\mathbf{Z}\mathbf{V}\mathbf{V}^* \\
&\quad - \mathbf{U}\mathbf{U}^*\mathbf{Z}\mathbf{V}\mathbf{V}^* - \mathbf{U}\mathbf{U}^*\mathbb{P}_S\mathbf{Z}\mathbf{V}\mathbf{V}^* + \mathbf{U}\mathbf{U}^*\mathbf{Z}\mathbf{V}\mathbf{V}^* \\
&= \mathbf{U}\mathbf{U}^*\mathbf{Z} + \mathbb{P}_S\mathbf{Z}\mathbf{V}\mathbf{V}^* - \mathbf{U}\mathbf{U}^*\mathbf{Z}\mathbf{V}\mathbf{V}^* = \mathbb{P}_{\mathbf{U},\mathbf{V},S}\mathbf{Z}.
\end{aligned}
$$

One can easily check that $\mathbb{P}_{\mathbf{U},\mathbf{V},S}$ acts as identity when applied to matrices in $T_{\mathbf{U},\mathbf{V},S}$ and that $\mathbb{P}_{\mathbf{U},\mathbf{V},S} = \mathbb{P}^*_{\mathbf{U},\mathbf{V},S}$ since $\langle \mathbf{Z}', \mathbb{P}_{\mathbf{U},\mathbf{V},S}\mathbf{Z}\rangle_F = \langle \mathbb{P}_{\mathbf{U},\mathbf{V},S}\mathbf{Z}', \mathbf{Z}\rangle_F$, for any $\mathbf{Z}, \mathbf{Z}'$. This proves the claim.

## C.3 Proof of Lemma B.3

Let $\Xi \in \ker(\mathcal{A})$. Note that

$$
0 = \|\mathcal{A}(\Xi)\|_2 = \left\|\mathcal{A}(\mathbb{P}_{T_{\mathbf{U},\mathbf{V},S}}(\Xi) + \mathbb{P}^\perp_{T_{\mathbf{U},\mathbf{V},S}}(\Xi))\right\|_2 \geq \left\|\mathcal{A}(\mathbb{P}_{T_{\mathbf{U},\mathbf{V},S}}(\Xi))\right\|_2 - \left\|\mathcal{A}(\mathbb{P}^\perp_{T_{\mathbf{U},\mathbf{V},S}}(\Xi))\right\|_2
$$

By the RIP we hence get that

$$
\begin{aligned}
\left\|\mathbb{P}_{T_{\mathbf{U},\mathbf{V},S}}(\Xi)\right\|_F^2 &\leq \frac{1}{1-\delta}\left\|\mathcal{A}(\mathbb{P}_{T_{\mathbf{U},\mathbf{V},S}}(\Xi))\right\|_2^2 \leq \frac{1}{1-\delta}\left\|\mathcal{A}(\mathbb{P}^\perp_{T_{\mathbf{U},\mathbf{V},S}}(\Xi))\right\|_2^2 \\
&\leq \frac{\|\mathcal{A}\|^2_{2\to 2}}{(1-\delta)}\left\|\mathbb{P}^\perp_{T_{\mathbf{U},\mathbf{V},S}}(\Xi)\right\|_F^2.
\end{aligned}
$$

Consequently,

$$
\|\Xi\|_F^2 = \left\|\mathbb{P}_{T_{\mathbf{U},\mathbf{V},S}}(\Xi)\right\|_F^2 + \left\|\mathbb{P}^\perp_{T_{\mathbf{U},\mathbf{V},S}}(\Xi)\right\|_F^2 \leq \left(1 + \frac{\|\mathcal{A}\|^2_{2\to 2}}{(1-\delta)}\right)\left\|\mathbb{P}^\perp_{T_{\mathbf{U},\mathbf{V},S}}(\Xi)\right\|_F^2.
$$

## C.4 Proof of Lemma B.6

In the proof of Lemma B.6 we will use the following fact.

**Lemma C.2.** *Let $W^{lr}_{\mathbf{X}^{(k)},\varepsilon_k}$ be the weight operator defined in (9), which is based on the matrices $\mathbf{U} \in \mathbb{R}^{n_1 \times r_k}$ and $\mathbf{V} \in \mathbb{R}^{n_2 \times r_k}$ of leading $r_k$ left and right singular vectors of $\mathbf{X}^{(k)}$. If $\mathbf{M} \in T_{\mathbf{U},\mathbf{V}}$, then $W^{lr}_{\mathbf{X}^{(k)},\varepsilon_k}(\mathbf{M}) \in T_{\mathbf{U},\mathbf{V}}$. If $\mathbf{M} \in T^\perp_{\mathbf{U},\mathbf{V}}$, then $W^{lr}_{\mathbf{X}^{(k)},\varepsilon_k}(\mathbf{M}) \in T^\perp_{\mathbf{U},\mathbf{V}}$.*

**Proof :** If $\mathbf{M} \in T_{\mathbf{U},\mathbf{V}}$, there exist $\mathbf{M}_1 \in \mathbb{R}^{r_k \times r_k}$, $\mathbf{M}_2 \in \mathbb{R}^{r_k \times (n_2-r_k)}$, $\mathbf{M}_3 \in \mathbb{R}^{(n_1-r_k)\times r_k}$ such that

$$
\mathbf{M} = \begin{bmatrix}\mathbf{U} & \mathbf{U}_\perp\end{bmatrix}\begin{bmatrix}\mathbf{M}_1 & \mathbf{M}_2 \\ \mathbf{M}_3 & 0\end{bmatrix}\begin{bmatrix}\mathbf{V}^* \\ \mathbf{V}^*_\perp\end{bmatrix},
$$

e.g., see [90, Proposition 2.1]. We thus observe that the weight operator $W^{lr}_{\mathbf{X}^{(k)},\varepsilon_k} : \mathbb{R}^{n_1 \times n_2} \to \mathbb{R}^{n_1 \times n_2}$ from (18) satisfies

$$
\begin{aligned}
&W^{lr}_{\mathbf{X}^{(k)},\varepsilon_k}(\mathbf{M}) \\
&= \begin{bmatrix}\mathbf{U} & \mathbf{U}_\perp\end{bmatrix}\left(\mathbf{H}(\boldsymbol{\sigma}^{(k)},\varepsilon_k)\circ\left(\begin{bmatrix}\mathbf{U}^* \\ \mathbf{U}^*_\perp\end{bmatrix}\begin{bmatrix}\mathbf{U} & \mathbf{U}_\perp\end{bmatrix}\begin{bmatrix}\mathbf{M}_1 & \mathbf{M}_2 \\ \mathbf{M}_3 & 0\end{bmatrix}\begin{bmatrix}\mathbf{V}^* \\ \mathbf{V}^*_\perp\end{bmatrix}\begin{bmatrix}\mathbf{V} & \mathbf{V}_\perp\end{bmatrix}\right)\right)\begin{bmatrix}\mathbf{V}^* \\ \mathbf{V}^*_\perp\end{bmatrix} \\
&= \begin{bmatrix}\mathbf{U} & \mathbf{U}_\perp\end{bmatrix}\left(\mathbf{H}(\boldsymbol{\sigma}^{(k)},\varepsilon_k)\circ\begin{bmatrix}\mathbf{M}_1 & \mathbf{M}_2 \\ \mathbf{M}_3 & 0\end{bmatrix}\right)\begin{bmatrix}\mathbf{V}^* \\ \mathbf{V}^*_\perp\end{bmatrix} \\
&= \begin{bmatrix}\mathbf{U} & \mathbf{U}_\perp\end{bmatrix}\begin{bmatrix}\mathbf{H}_1^{(k)}\circ\mathbf{M}_1 & \mathbf{H}_2^{(k)}\circ\mathbf{M}_2 \\ \mathbf{H}_3^{(k)}\circ\mathbf{M}_3 & 0\end{bmatrix}\begin{bmatrix}\mathbf{V}^* \\ \mathbf{V}^*_\perp\end{bmatrix} \in T_{\mathbf{U},\mathbf{V}}.
\end{aligned}
$$

Similarly, if $\mathbf{M} \in T^\perp_{\mathbf{U},\mathbf{V}}$, there exists $\mathbf{M}_4 \in \mathbb{R}^{(n_1-r_k)\times(n_2-r_k)}$ such that $\mathbf{M} = \mathbf{U}_\perp\mathbf{M}_4(\mathbf{V}_\perp)^*$ and

$$
\begin{aligned}
W^{lr}_{\mathbf{X}^{(k)},\varepsilon_k}(\mathbf{M}) &= \begin{bmatrix}\mathbf{U} & \mathbf{U}_\perp\end{bmatrix}\left(\mathbf{H}(\boldsymbol{\sigma}^{(k)},\varepsilon_k)\circ\begin{bmatrix}0 & 0 \\ 0 & \mathbf{M}_4\end{bmatrix}\right)\begin{bmatrix}\mathbf{V}^* \\ \mathbf{V}^*_\perp\end{bmatrix} \\
&= \begin{bmatrix}\mathbf{U} & \mathbf{U}_\perp\end{bmatrix}\begin{bmatrix}0 & 0 \\ 0 & \mathbf{M}_4/\varepsilon_k^2\end{bmatrix}\begin{bmatrix}\mathbf{V}^* \\ \mathbf{V}^*_\perp\end{bmatrix} \in T^\perp_{\mathbf{U},\mathbf{V}}. \qquad \blacksquare
\end{aligned}
$$

**Proof of Lemma B.6:** Let $\Xi \in \mathbb{R}^{n_1 \times n_2}$ be arbitrary. We start with some simple but technical observations. First note that by Lemma B.1, if $T_k = T_{\widetilde{\mathbf{U}}, \widetilde{\mathbf{V}}, S}$,

$$
\begin{aligned}
\mathbb{P}_{T_k^\perp} \Xi &= (\mathbf{Id} - \mathbb{P}_{T_k})(\Xi) \\
&= \Xi - \left( \widetilde{\mathbf{U}} \widetilde{\mathbf{U}}^* \Xi + \mathbb{P}_S \Xi \widetilde{\mathbf{V}} \widetilde{\mathbf{V}}^* - \widetilde{\mathbf{U}} \widetilde{\mathbf{U}}^* \Xi \widetilde{\mathbf{V}} \widetilde{\mathbf{V}}^* \right) \\
&= (\mathbf{Id} - \mathbb{P}_{\widetilde{\mathbf{U}}, \widetilde{\mathbf{V}}}) \Xi + (\mathbf{Id} - \mathbb{P}_S) \Xi \widetilde{\mathbf{V}} \widetilde{\mathbf{V}}^* \\
&= \mathbb{P}_{\widetilde{\mathbf{U}}, \widetilde{\mathbf{V}}}^\perp \Xi + \mathbb{P}_{S^c} \Xi \widetilde{\mathbf{V}} \widetilde{\mathbf{V}}^*,
\end{aligned}
\tag{45}
$$

with

$$
\left\langle \mathbb{P}_{\widetilde{\mathbf{U}}, \widetilde{\mathbf{V}}}^\perp \Xi, \mathbb{P}_{S^c} \Xi \widetilde{\mathbf{V}} \widetilde{\mathbf{V}}^* \right\rangle = \left\langle \widetilde{\mathbf{U}}_\perp \widetilde{\mathbf{U}}_\perp^* \Xi \widetilde{\mathbf{V}}_\perp \widetilde{\mathbf{V}}_\perp^*, \mathbb{P}_{S^c} \Xi \widetilde{\mathbf{V}} \widetilde{\mathbf{V}}^* \right\rangle = 0,
\tag{46}
$$

where we used that $\mathbb{P}_{\widetilde{\mathbf{U}}, \widetilde{\mathbf{V}}}^\perp \Xi = (\mathbf{Id} - \mathbb{P}_{\widetilde{\mathbf{U}}, \widetilde{\mathbf{V}}}) \Xi = \widetilde{\mathbf{U}}_\perp \widetilde{\mathbf{U}}_\perp^* \Xi \widetilde{\mathbf{V}}_\perp \widetilde{\mathbf{V}}_\perp^*$, for $\widetilde{\mathbf{U}}_\perp \in \mathbb{R}^{n_1 \times (n_1 - r)}$ and $\widetilde{\mathbf{V}}_\perp \in \mathbb{R}^{n_2 \times (n_2 - r)}$ being the complementary orthonormal bases of $\widetilde{\mathbf{U}}$ and $\widetilde{\mathbf{V}}$, and that $\widetilde{\mathbf{V}}_\perp^* \widetilde{\mathbf{V}} = \mathbf{0}$.

Second, let now $\mathbf{U} \in \mathbb{R}^{n_1 \times r}$ and $\mathbf{V} \in \mathbb{R}^{n_2 \times r}$ be matrices with $r$ leading left and right singular vectors of $\mathbf{X}^{(k)}$ in their columns which coincide with the matrices $\mathbf{U}$ and $\mathbf{V}$ from Definition 2.1 in their first $r$ columns. Then it follows from (45) and (46) that

$$
\begin{aligned}
&\left\| \mathbb{P}_{T_k^\perp}(\Xi) \right\|_F^2 \\
&= \left\| \mathbb{P}_{\widetilde{\mathbf{U}}, \widetilde{\mathbf{V}}}^\perp \Xi \right\|_F^2 + \left\| \mathbb{P}_{S^c} \Xi \widetilde{\mathbf{V}} \widetilde{\mathbf{V}}^* \right\|_F^2 \\
&= \left\| \mathbb{P}_{\mathbf{U}, \mathbf{V}}^\perp \Xi + (\mathbb{P}_{\widetilde{\mathbf{U}}, \widetilde{\mathbf{V}}}^\perp - \mathbb{P}_{\mathbf{U}, \mathbf{V}}^\perp) \Xi \right\|_F^2 + \left\| \mathbb{P}_{S^c} \Xi (\mathbf{V} \mathbf{V}^* + (\widetilde{\mathbf{V}} \widetilde{\mathbf{V}}^* - \mathbf{V} \mathbf{V}^*)) \right\|_F^2 \\
&\leq 2 \left( \left\| \mathbb{P}_{\mathbf{U}, \mathbf{V}}^\perp \Xi \right\|_F^2 + \left\| (\mathbb{P}_{\widetilde{\mathbf{U}}, \widetilde{\mathbf{V}}}^\perp - \mathbb{P}_{\mathbf{U}, \mathbf{V}}^\perp) \Xi \right\|_F^2 + \| \mathbb{P}_{S^c} \Xi \mathbf{V} \mathbf{V}^* \|_F^2 + \left\| \mathbb{P}_{S^c} \Xi (\widetilde{\mathbf{V}} \widetilde{\mathbf{V}}^* - \mathbf{V} \mathbf{V}^*) \right\|_F^2 \right).
\end{aligned}
\tag{47}
$$

By an argument analogous to (46), we observe that

$$
\left\| \mathbb{P}_{\mathbf{U}, \mathbf{V}}^\perp \Xi \right\|_F^2 + \| \mathbb{P}_{S^c} \Xi \mathbf{V} \mathbf{V}^* \|_F^2 = \langle \mathbb{P}_{\mathbf{U}, \mathbf{V}}^\perp \Xi + \mathbb{P}_{S^c} \Xi \mathbf{V} \mathbf{V}^*, \mathbb{P}_{\mathbf{U}, \mathbf{V}}^\perp \Xi + \mathbb{P}_{S^c} \Xi \mathbf{V} \mathbf{V}^* \rangle = \langle \widetilde{\Xi}, \widetilde{\Xi} \rangle,
$$

where $\widetilde{\Xi} = \mathbb{P}_{\mathcal{M}}(\Xi)$ is an element of the subspace $\mathcal{M} = \mathcal{M}_1 \oplus \mathcal{M}_2 \subset \mathbb{R}^{n_1 \times n_2}$ that is the direct sum of the subspaces $\mathcal{M}_1 := \{ \mathbb{P}_{\mathbf{U}, \mathbf{V}}^\perp \mathbf{Z} : \mathbf{Z} \in \mathbb{R}^{n_1 \times n_2} \}$ and $\mathcal{M}_2 := \{ \mathbb{P}_{S^c} \mathbf{Z} \mathbf{V} \mathbf{V}^* : \mathbf{Z} \in \mathbb{R}^{n_1 \times n_2} \}$.

Let now $W_{\mathbf{X}^{(k)}, \varepsilon_k}^{lr}$ be the rank promoting part of the weight operator from (9) and $\mathbf{W}_{\mathbf{X}^{(k)}, \delta_k}^{sp}$ be the row-sparsity promoting part from (10). Note that the restriction of

$$
\bar{W} := \mathbb{P}_{\mathbf{U}, \mathbf{V}}^\perp W_{\mathbf{X}^{(k)}, \varepsilon_k}^{lr} \mathbb{P}_{\mathbf{U}, \mathbf{V}}^\perp + \mathbb{P}_{S^c} \mathbf{W}_{\mathbf{X}^{(k)}, \delta_k}^{sp} \mathbb{P}_{S^c}
$$

to $\mathcal{M}$ is invertible as its first summand is invertible on $\mathcal{M}_1 = T_{\mathbf{U}, \mathbf{V}}^\perp$, its second summand is invertible on $\mathcal{M}_2$ (recall that the weight operators are positive definite), and $\mathcal{M}_1 \perp \mathcal{M}_2$. Therefore it holds that

$$
\begin{aligned}
&\left\| \mathbb{P}_{\mathbf{U}, \mathbf{V}}^\perp \Xi \right\|_F^2 + \| \mathbb{P}_{S^c} \Xi \mathbf{V} \mathbf{V}^* \|_F^2 \\
&= \langle \widetilde{\Xi}, \widetilde{\Xi} \rangle = \left\langle \bar{W}_{|\mathcal{M}}^{1/2} \widetilde{\Xi}, \bar{W}_{|\mathcal{M}}^{-1} \bar{W}_{|\mathcal{M}}^{1/2} \widetilde{\Xi} \right\rangle \\
&\leq \sigma_1 \left( \bar{W}_{|\mathcal{M}}^{-1} \right) \left\langle \widetilde{\Xi}, \bar{W}_{|\mathcal{M}} \widetilde{\Xi} \right\rangle = \frac{1}{\sigma_{\min} \left( \bar{W}_{|\mathcal{M}} \right)} \left\langle \widetilde{\Xi}, \bar{W} \widetilde{\Xi} \right\rangle \\
&\leq \frac{1}{\sigma_{\min} \left( \left( \mathbb{P}_{\mathbf{U}, \mathbf{V}}^\perp W_{\mathbf{X}^{(k)}, \varepsilon_k}^{lr} \mathbb{P}_{\mathbf{U}, \mathbf{V}}^\perp \right)_{|\mathcal{M}} \right) + \sigma_{\min} \left( \left( \mathbb{P}_{S^c} \mathbf{W}_{\mathbf{X}^{(k)}, \delta_k}^{sp} \mathbb{P}_{S^c} \right)_{|\mathcal{M}} \right)} \left\langle \widetilde{\Xi}, \bar{W} \widetilde{\Xi} \right\rangle \\
&\leq \frac{1}{\frac{\varepsilon_k^2}{\sigma_{r+1}^2(\mathbf{X}^{(k)})} + \frac{\delta_k^2}{\rho_{s+1}^2(\mathbf{X}^{(k)})}} \left\langle \Xi, \bar{W} \Xi \right\rangle.
\end{aligned}
$$

In the first inequality, we used that $\bar{W}_{|\mathcal{M}}$ is positive definite. In the second inequality, we used that $\sigma_{\min}(A + B) \geq \sigma_{\min}(A) + \sigma_{\min}(B)$, for any positive semidefinite operators $A$ and $B$, and and in the third inequality that $\langle \widetilde{\Xi}, \bar{W} \widetilde{\Xi} \rangle \leq \langle \Xi, \bar{W} \Xi \rangle$. The latter observation can be deduced as follows: Note that, by the self-adjointness of $\bar{W}$,

$$
\begin{aligned}
\langle \Xi, \bar{W} \Xi \rangle &= \langle \mathbb{P}_{\mathcal{M}}(\Xi), \bar{W} \mathbb{P}_{\mathcal{M}}(\Xi) \rangle + \langle \mathbb{P}_{\mathcal{M}}(\Xi), \bar{W} \mathbb{P}_{\mathcal{M}}^\perp(\Xi) \rangle + \langle \mathbb{P}_{\mathcal{M}}^\perp(\Xi), \bar{W} \mathbb{P}_{\mathcal{M}}(\Xi) \rangle + \langle \mathbb{P}_{\mathcal{M}}^\perp(\Xi), \bar{W} \mathbb{P}_{\mathcal{M}}^\perp(\Xi) \rangle \\
&= \langle \widetilde{\Xi}, \bar{W} \widetilde{\Xi} \rangle + 2 \langle \mathbb{P}_{\mathcal{M}}^\perp(\Xi), \bar{W} \mathbb{P}_{\mathcal{M}}(\Xi) \rangle + \langle \mathbb{P}_{\mathcal{M}}^\perp(\Xi), \bar{W} \mathbb{P}_{\mathcal{M}}^\perp(\Xi) \rangle.
\end{aligned}
$$

Since $\bar{W}$ is positive semi-definite (due to the fact that both $W^{lr}_{\mathbf{X}^{(k)},\varepsilon_k}$ and $\mathbf{W}^{sp}_{\mathbf{X}^{(k)},\delta_k}$ are positive definite), all that remains is to argue that the mixed term on the right-hand side vanishes. To this end, note that $\mathbb{P}^{\perp}_{\mathbf{U},\mathbf{V}}\mathbf{Z} = \mathbb{P}_{S^c}\mathbf{Z}\mathbf{V}\mathbf{V}^* = 0$, for any $\mathbf{Z} \in \mathcal{M}_{\perp}$ and compute

$$
\begin{aligned}
\langle \mathbb{P}^{\perp}_{\mathcal{M}}(\boldsymbol{\Xi}), \bar{W}\mathbb{P}_{\mathcal{M}}(\boldsymbol{\Xi})\rangle &= \langle \mathbb{P}^{\perp}_{\mathcal{M}}(\boldsymbol{\Xi}), \mathbb{P}^{\perp}_{\mathbf{U},\mathbf{V}}(W^{lr}_{\mathbf{X}^{(k)},\varepsilon_k}(\mathbb{P}^{\perp}_{\mathbf{U},\mathbf{V}}(\mathbb{P}_{\mathcal{M}_1}(\boldsymbol{\Xi})))) \rangle + \langle \mathbb{P}^{\perp}_{\mathcal{M}}(\boldsymbol{\Xi}), \mathbb{P}_{S^c}\mathbf{W}^{sp}_{\mathbf{X}^{(k)},\delta_k}\mathbb{P}_{S^c}\mathbb{P}_{\mathcal{M}_2}(\boldsymbol{\Xi})\rangle \\
&= \langle \mathbb{P}^{\perp}_{\mathbf{U},\mathbf{V}}(\mathbb{P}^{\perp}_{\mathcal{M}}(\boldsymbol{\Xi})), W^{lr}_{\mathbf{X}^{(k)},\varepsilon_k}(\mathbb{P}^{\perp}_{\mathbf{U},\mathbf{V}}(\mathbb{P}_{\mathcal{M}_1}(\boldsymbol{\Xi}))) \rangle + \langle \mathbb{P}_{S^c}\mathbb{P}^{\perp}_{\mathcal{M}}(\boldsymbol{\Xi})\mathbf{V}\mathbf{V}^*, \mathbf{W}^{sp}_{\mathbf{X}^{(k)},\delta_k}\mathbb{P}_{S^c}\boldsymbol{\Xi}\rangle \\
&= 0.
\end{aligned}
$$

We can now continue by estimating

$$
\begin{aligned}
\left\langle \boldsymbol{\Xi}, \bar{W}\boldsymbol{\Xi}\right\rangle &= \left\langle \boldsymbol{\Xi}, \mathbb{P}^{\perp}_{\mathbf{U},\mathbf{V}}W^{lr}_{\mathbf{X}^{(k)},\varepsilon_k}\mathbb{P}^{\perp}_{\mathbf{U},\mathbf{V}}\boldsymbol{\Xi}\right\rangle + \left\langle \boldsymbol{\Xi}, \mathbb{P}_{S^c}\mathbf{W}^{sp}_{\mathbf{X}^{(k)},\delta_k}\mathbb{P}_{S^c}\boldsymbol{\Xi}\right\rangle \\
&= \left\langle \mathbb{P}^{\perp}_{\mathbf{U},\mathbf{V}}(\boldsymbol{\Xi}), W^{lr}_{\mathbf{X}^{(k)},\varepsilon_k}\mathbb{P}^{\perp}_{\mathbf{U},\mathbf{V}}(\boldsymbol{\Xi})\right\rangle + \left\langle \mathbb{P}_{S^c}\boldsymbol{\Xi}, \mathbf{W}^{sp}_{\mathbf{X}^{(k)},\delta_k}\mathbb{P}_{S^c}\boldsymbol{\Xi}\right\rangle \\
&\leq \left\langle \boldsymbol{\Xi}, W^{lr}_{\mathbf{X}^{(k)},\varepsilon_k}\boldsymbol{\Xi}\right\rangle + \left\langle \boldsymbol{\Xi}, \mathbf{W}^{sp}_{\mathbf{X}^{(k)},\delta_k}\boldsymbol{\Xi}\right\rangle \\
&= \left\langle \boldsymbol{\Xi}, W_{\mathbf{X}^{(k)},\varepsilon_k,\delta_k}\boldsymbol{\Xi}\right\rangle,
\end{aligned}
\tag{48}
$$

using the positive semidefiniteness of $W^{lr}_{\mathbf{X}^{(k)},\varepsilon_k}$ and $\mathbf{W}^{sp}_{\mathbf{X}^{(k)},\delta_k}$ in the last inequality. To be precise, the last inequality can be argued as follows: Due to complimentary supports $S$ and $S^c$, we see that

$$
\begin{aligned}
\langle \boldsymbol{\Xi}, \mathbf{W}^{sp}_{\mathbf{X}^{(k)},\delta_k}\boldsymbol{\Xi}\rangle &= \langle \mathbb{P}_S\boldsymbol{\Xi}, \mathbf{W}^{sp}_{\mathbf{X}^{(k)},\delta_k}\mathbb{P}_S\boldsymbol{\Xi}\rangle + \langle \mathbb{P}_{S^c}\boldsymbol{\Xi}, \mathbf{W}^{sp}_{\mathbf{X}^{(k)},\delta_k}\mathbb{P}_{S^c}\boldsymbol{\Xi}\rangle \\
&\quad + \underbrace{\langle \mathbb{P}_S\boldsymbol{\Xi}, \mathbf{W}^{sp}_{\mathbf{X}^{(k)},\delta_k}\mathbb{P}_{S^c}\boldsymbol{\Xi}\rangle}_{=0} + \underbrace{\langle \mathbb{P}_{S^c}\boldsymbol{\Xi}, \mathbf{W}^{sp}_{\mathbf{X}^{(k)},\delta_k}\mathbb{P}_S\boldsymbol{\Xi}\rangle}_{=0} \\
&\geq \langle \mathbb{P}_{S^c}\boldsymbol{\Xi}, \mathbf{W}^{sp}_{\mathbf{X}^{(k)},\delta_k}\mathbb{P}_{S^c}\boldsymbol{\Xi}\rangle.
\end{aligned}
\tag{49}
$$

Similarly, we note that $W^{lr}_{\mathbf{X}^{(k)},\varepsilon_k}$ acts diagonally on $T_{\mathbf{U},\mathbf{V}}$ and $T^{\perp}_{\mathbf{U},\mathbf{V}}$. Indeed, we have by Lemma C.2 that if $\mathbf{M} \in T_{\mathbf{U},\mathbf{V}}$, then $W^{lr}_{\mathbf{X}^{(k)},\varepsilon_k}(\mathbf{M}) \in T_{\mathbf{U},\mathbf{V}}$ and if $\mathbf{M} \in T^{\perp}_{\mathbf{U},\mathbf{V}}$, then $W^{lr}_{\mathbf{X}^{(k)},\varepsilon_k}(\mathbf{M}) \in T^{\perp}_{\mathbf{U},\mathbf{V}}$, which implies

$$
\langle \mathbb{P}_{\mathbf{U},\mathbf{V}}\boldsymbol{\Xi}, W^{lr}_{\mathbf{X}^{(k)},\varepsilon_k}(\mathbb{P}^{\perp}_{\mathbf{U},\mathbf{V}}\boldsymbol{\Xi})\rangle = 0 \quad \text{and} \quad \langle \mathbb{P}^{\perp}_{\mathbf{U},\mathbf{V}}\boldsymbol{\Xi}, W^{lr}_{\mathbf{X}^{(k)},\varepsilon_k}(\mathbb{P}_{\mathbf{U},\mathbf{V}}\boldsymbol{\Xi})\rangle = 0
$$

due to the orthogonality of elements in $T^{\perp}_{\mathbf{U},\mathbf{V}}$ and $T_{\mathbf{U},\mathbf{V}}$, respectively, and therefore it follows from $\boldsymbol{\Xi} = T_{\mathbf{U},\mathbf{V}}(\boldsymbol{\Xi}) + T^{\perp}_{\mathbf{U},\mathbf{V}}(\boldsymbol{\Xi})$ that

$$
\begin{aligned}
\langle \boldsymbol{\Xi}, W^{lr}_{\mathbf{X}^{(k)},\varepsilon_k}(\boldsymbol{\Xi})\rangle &= \langle \mathbb{P}_{\mathbf{U},\mathbf{V}}\boldsymbol{\Xi}, W^{lr}_{\mathbf{X}^{(k)},\varepsilon_k}(\mathbb{P}_{\mathbf{U},\mathbf{V}}\boldsymbol{\Xi})\rangle + \langle \mathbb{P}^{\perp}_{\mathbf{U},\mathbf{V}}\boldsymbol{\Xi}, W^{lr}_{\mathbf{X}^{(k)},\varepsilon_k}(\mathbb{P}^{\perp}_{\mathbf{U},\mathbf{V}}\boldsymbol{\Xi})\rangle \\
&\quad + \underbrace{\langle \mathbb{P}_{\mathbf{U},\mathbf{V}}\boldsymbol{\Xi}, W^{lr}_{\mathbf{X}^{(k)},\varepsilon_k}(\mathbb{P}^{\perp}_{\mathbf{U},\mathbf{V}}\boldsymbol{\Xi})\rangle}_{=0} + \underbrace{\langle \mathbb{P}^{\perp}_{\mathbf{U},\mathbf{V}}\boldsymbol{\Xi}, W^{lr}_{\mathbf{X}^{(k)},\varepsilon_k}(\mathbb{P}_{\mathbf{U},\mathbf{V}}\boldsymbol{\Xi})\rangle}_{=0} \\
&\geq \langle \mathbb{P}^{\perp}_{\mathbf{U},\mathbf{V}}\boldsymbol{\Xi}, W^{lr}_{\mathbf{X}^{(k)},\varepsilon_k}(\mathbb{P}^{\perp}_{\mathbf{U},\mathbf{V}}\boldsymbol{\Xi})\rangle.
\end{aligned}
\tag{50}
$$

Combining the previous estimates with (47) and noticing that

$$
\frac{1}{\frac{\varepsilon_k^2}{\sigma^2_{r+1}(\mathbf{X}^{(k)})} + \frac{\delta_k^2}{\rho^2_{s+1}(\mathbf{X}^{(k)})}} \leq \min\left\{ \frac{\sigma^2_{r+1}(\mathbf{X}^{(k)})}{\varepsilon_k^2}, \frac{\rho^2_{s+1}(\mathbf{X}^{(k)})}{\delta_k^2}\right\},
$$

we obtain

$$
\begin{aligned}
\left\|\mathbb{P}_{T_k^{\perp}}(\boldsymbol{\Xi})\right\|_F^2 &\leq 2\left( \min\left\{ \frac{\sigma^2_{r+1}(\mathbf{X}^{(k)})}{\varepsilon_k^2}, \frac{\rho^2_{s+1}(\mathbf{X}^{(k)})}{\delta_k^2}\right\} \left\langle \boldsymbol{\Xi}, W_{\mathbf{X}^{(k)},\varepsilon_k,\delta_k}\boldsymbol{\Xi}\right\rangle \right) \\
&\quad + 2\left\|(\mathbb{P}^{\perp}_{\widetilde{\mathbf{U}},\widetilde{\mathbf{V}}} - \mathbb{P}^{\perp}_{\mathbf{U},\mathbf{V}})\boldsymbol{\Xi}\right\|_F^2 + 2\left\|\mathbb{P}_{S^c}\boldsymbol{\Xi}(\widetilde{\mathbf{V}}\widetilde{\mathbf{V}}^* - \mathbf{V}\mathbf{V}^*)\right\|_F^2.
\end{aligned}
\tag{51}
$$

Next, we control the last two summands in (51) using matrix perturbation results. Recall that $\mathbf{U}_\star \in \mathbb{R}^{n_1 \times r}$ and $\mathbf{V}_\star \in \mathbb{R}^{n_2 \times r}$ are the singular vector matrices of the reduced singular value decomposition of $\mathbf{X}_\star$. First observe that

$$
\begin{aligned}
&\left\|(\mathbb{P}_{\widetilde{\mathbf{U}},\widetilde{\mathbf{V}}} - \mathbb{P}_{\mathbf{U}_\star,\mathbf{V}_\star})\boldsymbol{\Xi}\right\|_F \\
&= \left\|(\widetilde{\mathbf{U}}\widetilde{\mathbf{U}}^* - \mathbf{U}_\star\mathbf{U}_\star^*)\boldsymbol{\Xi}(\mathbf{Id} - \mathbf{V}_\star\mathbf{V}_\star^*) + (\mathbf{Id} - \widetilde{\mathbf{U}}\widetilde{\mathbf{U}}^*)\boldsymbol{\Xi}(\widetilde{\mathbf{V}}\widetilde{\mathbf{V}}^* - \mathbf{V}_\star\mathbf{V}_\star^*)\right\|_F \\
&\leq \left\|\widetilde{\mathbf{U}}\widetilde{\mathbf{U}}^* - \mathbf{U}_\star\mathbf{U}_\star^*\right\| \|\boldsymbol{\Xi}\|_F \|\mathbf{Id} - \mathbf{V}_\star\mathbf{V}_\star^*\| + \left\|\mathbf{Id} - \widetilde{\mathbf{U}}\widetilde{\mathbf{U}}^*\right\| \|\boldsymbol{\Xi}\|_F \left\|\widetilde{\mathbf{V}}\widetilde{\mathbf{V}}^* - \mathbf{V}_\star\mathbf{V}_\star^*\right\| \\
&\leq \left( \left\|\widetilde{\mathbf{U}}\widetilde{\mathbf{U}}^* - \mathbf{U}_\star\mathbf{U}_\star^*\right\| + \left\|\widetilde{\mathbf{V}}\widetilde{\mathbf{V}}^* - \mathbf{V}_\star\mathbf{V}_\star^*\right\|\right) \|\boldsymbol{\Xi}\|_F.
\end{aligned}
$$

Now note that, by [12, Lemma 1] and [65, Theorem 3.5], we obtain

$$\left\|\widetilde{\mathbf{U}}\widetilde{\mathbf{U}}^* - \mathbf{U}_\star\mathbf{U}_\star^*\right\| + \left\|\widetilde{\mathbf{V}}\widetilde{\mathbf{V}}^* - \mathbf{V}_\star\mathbf{V}_\star^*\right\| \le 2\left(\|\mathbf{U}_\star\widetilde{\mathbf{U}}_\perp^*\| + \|\mathbf{V}_\star\widetilde{\mathbf{V}}_\perp^*\|\right) \le 4\frac{\|\mathsf{H}_s(\mathbf{X}^{(k)}) - \mathbf{X}_\star\|}{\sigma_r(\mathbf{X}_\star) - \sigma_{r+1}(\mathsf{H}_s(\mathbf{X}^{(k)}))}$$

$$\le 4\frac{\|\mathbf{X}^{(k)} - \mathbf{X}_\star\|}{\sigma_r(\mathbf{X}_\star) - \sigma_{r+1}(\mathbf{X}^{(k)})} \le 4\frac{\|\mathbf{X}^{(k)} - \mathbf{X}_\star\|}{(1 - 1/48)\sigma_r(\mathbf{X}_\star)},$$

(52)

where we used some small observations in the last two inequalities: First, $\sigma_{r+1}(\mathsf{H}_s(\mathbf{X}^{(k)})) \le \sigma_{r+1}(\mathbf{X}^{(k)})$, which follows from the rectangular Cauchy interlacing theorem [27, Theorem 23]. Second, according to Lemma B.2 and (25), the row-support $S$ of $\mathsf{H}_s(\mathbf{X}^{(k)})$ coincides with the row-support $S_\star = \{i \in [n_1] : \|(\mathbf{X}_\star)_{i,:}\|_2 \neq 0\}$ of $\mathbf{X}_\star$ and hence $\mathsf{H}_s(\mathbf{X}^{(k)}) - \mathbf{X}_\star$ is a submatrix of $\mathbf{X}^{(k)} - \mathbf{X}_\star$. Finally, $\sigma_{r+1}(\mathbf{X}^{(k)}) = \|\mathsf{T}_r(\mathbf{X}^{(k)}) - \mathbf{X}^{(k)}\| \le \|\mathbf{X}_\star - \mathbf{X}^{(k)}\| \le \frac{1}{48}\sigma_r(\mathbf{X}_\star)$ due to (25). Consequently,

$$\left\|(\mathbb{P}_{\widetilde{\mathbf{U}},\widetilde{\mathbf{V}}} - \mathbb{P}_{\mathbf{U}_\star,\mathbf{V}_\star})\mathbf{\Xi}\right\|_F \le 4\frac{\|\mathbf{X}^{(k)} - \mathbf{X}_\star\|}{(1 - 1/48)\sigma_r(\mathbf{X}_\star)}\|\mathbf{\Xi}\|_F$$

and, by a similar argument,

$$\|(\mathbb{P}_{\mathbf{U}_\star,\mathbf{V}_\star} - \mathbb{P}_{\mathbf{U},\mathbf{V}})\mathbf{\Xi}\|_F \le 4\frac{\|\mathbf{X}^{(k)} - \mathbf{X}_\star\|}{(1 - 1/48)\sigma_r(\mathbf{X}_\star)}\|\mathbf{\Xi}\|_F,$$

such that it follows from $\mathbb{P}_{\mathbf{U},\mathbf{V}}^\perp = \mathbf{Id} - \mathbb{P}_{\mathbf{U},\mathbf{V}}$ and $\mathbb{P}_{\widetilde{\mathbf{U}},\widetilde{\mathbf{V}}}^\perp = \mathbf{Id} - \mathbb{P}_{\widetilde{\mathbf{U}},\widetilde{\mathbf{V}}}$ that

$$\left\|(\mathbb{P}_{\widetilde{\mathbf{U}},\widetilde{\mathbf{V}}}^\perp - \mathbb{P}_{\mathbf{U},\mathbf{V}}^\perp)\mathbf{\Xi}\right\|_F = \left\|(\mathbb{P}_{\widetilde{\mathbf{U}},\widetilde{\mathbf{V}}} - \mathbb{P}_{\mathbf{U},\mathbf{V}})\mathbf{\Xi}\right\|_F \le \left\|(\mathbb{P}_{\widetilde{\mathbf{U}},\widetilde{\mathbf{V}}} - \mathbb{P}_{\mathbf{U}_\star,\mathbf{V}_\star})\mathbf{\Xi}\right\|_F + \left\|(\mathbb{P}_{\mathbf{U}_\star,\mathbf{V}_\star} - \mathbb{P}_{\mathbf{U},\mathbf{V}})\mathbf{\Xi}\right\|_F$$

$$\le 8\frac{\|\mathbf{X}^{(k)} - \mathbf{X}_\star\|}{(1 - 1/48)\sigma_r(\mathbf{X}_\star)}\|\mathbf{\Xi}\|_F.$$

(53)

To estimate the fourth summand in (51), we argue analogously that

$$\left\|\mathbb{P}_{S^c}\mathbf{\Xi}(\widetilde{\mathbf{V}}\widetilde{\mathbf{V}}^* - \mathbf{V}\mathbf{V}^*)\right\|_F \le \left\|\mathbf{\Xi}(\widetilde{\mathbf{V}}\widetilde{\mathbf{V}}^* - \mathbf{V}\mathbf{V}^*)\right\|_F \le \left(\|\widetilde{\mathbf{V}}\widetilde{\mathbf{V}}^* - \mathbf{V}_\star\mathbf{V}_\star^*\| + \|\mathbf{V}_\star\mathbf{V}_\star^* - \mathbf{V}\mathbf{V}^*\|\right)\|\mathbf{\Xi}\|_F$$

$$\le 2\left(\|\mathbf{V}_\star\widetilde{\mathbf{V}}_\perp^*\| + \|\mathbf{V}_\star\mathbf{V}_\perp^*\|\right)\|\mathbf{\Xi}\|_F$$

$$\le 2\left(\frac{\|\mathsf{H}_s(\mathbf{X}^{(k)}) - \mathbf{X}_\star\|}{\sigma_r(\mathbf{X}_\star) - \sigma_{r+1}(\mathsf{H}_s(\mathbf{X}^{(k)}))} + \frac{\|\mathbf{X}^{(k)} - \mathbf{X}_\star\|}{\sigma_r(\mathbf{X}_\star) - \sigma_{r+1}(\mathbf{X}^{(k)})}\right)\|\mathbf{\Xi}\|_F$$

$$\le 4\frac{\|\mathbf{X}^{(k)} - \mathbf{X}_\star\|}{\sigma_r(\mathbf{X}_\star) - \sigma_{r+1}(\mathbf{X}^{(k)})}\|\mathbf{\Xi}\|_F \le 4\frac{\|\mathbf{X}^{(k)} - \mathbf{X}_\star\|}{(1 - 1/48)\sigma_r(\mathbf{X}_\star)}\|\mathbf{\Xi}\|_F,$$

(54)

using again that $\sigma_{r+1}(\mathbf{X}^{(k)}) \le \|\mathbf{X}_\star - \mathbf{X}^{(k)}\| \le \frac{1}{48}\sigma_r(\mathbf{X}_\star)$ due to (25).

Let now $\mathbf{\Xi}^{(k+1)} = \mathbf{X}^{(k+1)} - \mathbf{X}_\star$. Combining (24) and (51)-(52) we can proceed to estimate that

$$\|\mathbf{\Xi}^{(k+1)}\|_F^2$$

$$\le c^2\left\|\mathbb{P}_{T_k^\perp}(\mathbf{\Xi}^{(k+1)})\right\|_F^2$$

$$\le 2c^2\min\left\{\frac{\sigma_{r+1}(\mathbf{X}^{(k)})}{\varepsilon_k}, \frac{\rho_{s+1}(\mathbf{X}^{(k)})}{\delta_k}\right\}^2\left\langle\mathbf{\Xi}^{(k+1)}, W_{\mathbf{X}^{(k)},\varepsilon_k,\delta_k}\mathbf{\Xi}^{(k+1)}\right\rangle$$

$$+ 2\left[\left(\frac{8}{47/48}\right)^2 + \left(\frac{4}{47/48}\right)^2\right]c^2\|\mathbf{\Xi}^{(k+1)}\|_F^2\frac{\|\mathbf{X}^{(k)} - \mathbf{X}_\star\|^2}{\sigma_r^2(\mathbf{X}_\star)}$$

$$\le 2c^2\min\left\{\frac{\sigma_{r+1}(\mathbf{X}^{(k)})}{\varepsilon_k}, \frac{\rho_{s+1}(\mathbf{X}^{(k)})}{\delta_k}\right\}^2\left\langle\mathbf{\Xi}^{(k+1)}, W_{\mathbf{X}^{(k)},\varepsilon_k,\delta_k}(\mathbf{\Xi}^{(k+1)})\right\rangle + 167c^2\|\mathbf{\Xi}^{(k+1)}\|_F^2\frac{1}{(19c)^2}$$

$$\le 2c^2\min\left\{\frac{\sigma_{r+1}(\mathbf{X}^{(k)})}{\varepsilon_k}, \frac{\rho_{s+1}(\mathbf{X}^{(k)})}{\delta_k}\right\}^2\left\langle\mathbf{\Xi}^{(k+1)}, W_{\mathbf{X}^{(k)},\varepsilon_k,\delta_k}(\mathbf{\Xi}^{(k+1)})\right\rangle + \frac{1}{2}\|\mathbf{\Xi}^{(k+1)}\|_F^2,$$

(55)

where the third inequality follows from (8) and (25). Hence, rearranging (55) yields

$$\|\mathbf{\Xi}^{(k+1)}\|_F^2 \le 4c^2\min\left\{\frac{\sigma_{r+1}(\mathbf{X}^{(k)})}{\varepsilon_k}, \frac{\rho_{s+1}(\mathbf{X}^{(k)})}{\delta_k}\right\}^2\left\langle\mathbf{\Xi}^{(k+1)}, W_{\mathbf{X}^{(k)},\varepsilon_k,\delta_k}(\mathbf{\Xi}^{(k+1)})\right\rangle.$$

By Lemma B.5, we know that $\mathbf{X}^{(k+1)}$ fulfills

$$0 = \left\langle \mathbf{\Xi}^{(k+1)}, W_{\mathbf{X}^{(k)}, \varepsilon_k, \delta_k}(\mathbf{X}^{(k+1)}) \right\rangle = \left\langle \mathbf{\Xi}^{(k+1)}, W_{\mathbf{X}^{(k)}, \varepsilon_k, \delta_k}(\mathbf{\Xi}^{(k+1)}) \right\rangle + \left\langle \mathbf{\Xi}^{(k+1)}, W_{\mathbf{X}^{(k)}, \varepsilon_k, \delta_k}(\mathbf{X}_\star) \right\rangle$$

such that we conclude that

$$\left\| \mathbf{\Xi}^{(k+1)} \right\|^2 \leq \left\| \mathbf{\Xi}^{(k+1)} \right\|_F^2$$

$$\leq 4c^2 \min \left\{ \frac{\sigma_{r+1}(\mathbf{X}^{(k)})}{\varepsilon_k}, \frac{\rho_{s+1}(\mathbf{X}^{(k)})}{\delta_k} \right\}^2 \left\langle \mathbf{\Xi}^{(k+1)}, W_{\mathbf{X}^{(k)}, \varepsilon_k, \delta_k}(\mathbf{\Xi}^{(k+1)}) \right\rangle$$

$$= -4c^2 \min \left\{ \frac{\sigma_{r+1}(\mathbf{X}^{(k)})}{\varepsilon_k}, \frac{\rho_{s+1}(\mathbf{X}^{(k)})}{\delta_k} \right\}^2 \left\langle \mathbf{\Xi}^{(k+1)}, W_{\mathbf{X}^{(k)}, \varepsilon_k, \delta_k}(\mathbf{X}_\star) \right\rangle$$

$$= -4c^2 \min \left\{ \frac{\sigma_{r+1}(\mathbf{X}^{(k)})}{\varepsilon_k}, \frac{\rho_{s+1}(\mathbf{X}^{(k)})}{\delta_k} \right\}^2 \left( \left\langle \mathbf{\Xi}^{(k+1)}, W_{\mathbf{X}^{(k)}, \varepsilon_k}^{lr}(\mathbf{X}_\star) \right\rangle + \left\langle \mathbf{\Xi}^{(k+1)}, \mathbf{W}_{\mathbf{X}^{(k)}, \delta_k}^{sp} \cdot \mathbf{X}_\star \right\rangle \right)$$

$$\leq 4c^2 \min \left\{ \frac{\sigma_{r+1}(\mathbf{X}^{(k)})}{\varepsilon_k}, \frac{\rho_{s+1}(\mathbf{X}^{(k)})}{\delta_k} \right\}^2 \left( \left\| W_{\mathbf{X}^{(k)}, \varepsilon_k}^{lr}(\mathbf{X}_\star) \right\|_* \left\| \mathbf{\Xi}^{(k+1)} \right\| + \left\| \mathbf{\Xi}^{(k+1)} \right\| \left\| \mathbf{W}_{\mathbf{X}^{(k)}, \delta_k}^{sp} \cdot \mathbf{X}_\star \right\|_{1,2} \right)$$

$$= 4c^2 \min \left\{ \frac{\sigma_{r+1}(\mathbf{X}^{(k)})}{\varepsilon_k}, \frac{\rho_{s+1}(\mathbf{X}^{(k)})}{\delta_k} \right\}^2 \left( \left\| W_{\mathbf{X}^{(k)}, \varepsilon_k}^{lr}(\mathbf{X}_\star) \right\|_* + \left\| \mathbf{W}_{\mathbf{X}^{(k)}, \delta_k}^{sp} \cdot \mathbf{X}_\star \right\|_{1,2} \right) \left\| \mathbf{\Xi}^{(k+1)} \right\|,$$

which completes the proof. We used in the penultimate line Hölder's inequality and that

$$|\langle \mathbf{A}, \mathbf{B} \rangle_F| = \left| \sum_{i,j} A_{i,j} B_{i,j} \right| \leq \sum_i \|\mathbf{A}_{i,:}\|_2 \|\mathbf{B}_{i,:}\|_2 \leq \left( \max_i \|\mathbf{A}_{i,:}\|_2 \right) \cdot \sum_i \|\mathbf{B}_{i,:}\|_2 \leq \|\mathbf{A}\| \|\mathbf{B}\|_{1,2},$$

for all matrices $\mathbf{A}, \mathbf{B}$. ∎

### C.5 Proof of Lemma B.9

Note that by Lemma B.2 and (27), $S := \mathrm{supp}(\mathsf{H}_s(\mathbf{X}^{(k)})) = \mathrm{supp}(\mathbf{X}_\star)$. Since by assumption $\delta_k \leq \rho_s(\mathbf{X}^{(k)})$ we have by definition of $\mathbf{W}_{\mathbf{X}^{(k)}, \delta_k}^{sp}$ that $\mathbf{Z} := \mathbf{W}_{\mathbf{X}^{(k)}, \delta_k}^{sp} \cdot \mathbf{X}_\star$ is a matrix with row-support $S$ and rows

$$\mathbf{Z}_{i,:} = \min \left\{ \frac{\delta_k^2}{\|(\mathbf{X}^{(k)})_{i,:}\|_2^2}, 1 \right\} (\mathbf{X}_\star)_{i,:} = \frac{\delta_k^2}{\|(\mathbf{X}^{(k)})_{i,:}\|_2^2} (\mathbf{X}_\star)_{i,:}$$

for $i \in S$. Now note that if (27) holds, then

$$\|\mathbf{Z}_{i,:}\|_2 = \frac{\delta_k^2 \|(\mathbf{X}_\star)_{i,:}\|_2}{\|(\mathbf{X}^{(k)})_{i,:}\|_2^2} \leq \frac{\delta_k^2}{(1 - \zeta)^2 \rho_s(\mathbf{X}_\star)},$$

where we used in the last estimate that with (27) and $\|(\mathbf{X}_\star)_{i,:}\|_2 \geq \rho_s(\mathbf{X}_\star)$, for $i \in S$, we have

$$\|(\mathbf{X}^{(k)})_{i,:}\|_2^2 \geq (\|(\mathbf{X}_\star)_{i,:}\|_2 - \|(\mathbf{X}^{(k)})_{i,:} - (\mathbf{X}_\star)_{i,:}\|_2)^2 \geq (\|(\mathbf{X}_\star)_{i,:}\|_2 - \zeta \rho_s(\mathbf{X}_\star))^2$$

$$\geq \left( (1 - \zeta) \rho_s(\mathbf{X}_\star) \right) \left( (1 - \zeta) \|(\mathbf{X}_\star)_{i,:}\|_2 \right),$$

for all $i \in S$. The claim easily follows since $\mathbf{Z}$ has only $s$ non-zero rows.

### C.6 Proof of Lemma B.10

In the proof of Lemma B.10, we use a simple technical observation.

**Lemma C.3.** *Let $\mathbf{X}_\star \in \mathcal{M}_{r,s}$, let $\mathbf{X}^{(k)}$ be the $k$-th iterate of Algorithm 1, and abbreviate $\mathbf{\Xi}^{(k)} = \mathbf{X}^{(k)} - \mathbf{X}_\star$. Then the following two statements hold true:*

    *1. If $\varepsilon_k \leq \sigma_{r+1}(\mathbf{X}^{(k)})$, then $\varepsilon_k \leq \|\mathbf{\Xi}^{(k)}\|$.*

    *2. If $\delta_k \leq \rho_{s+1}(\mathbf{X}^{(k)})$, then $\delta_k \leq \|\mathbf{\Xi}^{(k)}\|_{\infty,2}$.*

**Proof:** By defining $[\mathbf{X}^{(k)}]_r$ to be the best rank-$r$ approximation of $\mathbf{X}^{(k)}$ in any unitarily invariant norm, we bound

$$\varepsilon_k \leq \sigma_{r+1}(\mathbf{X}^{(k)}) = \|\mathbf{X}^{(k)} - [\mathbf{X}^{(k)}]_r\| \leq \|\mathbf{X}^{(k)} - \mathbf{X}_\star\| = \|\mathbf{\Xi}^{(k)}\|,$$

where the inequality follows the fact that $\mathbf{X}_\star$ is a rank-$r$ matrix.

Similarly, for the second statement, we have that

$$\delta_k \leq \rho_{s+1}(\mathbf{X}^{(k)}) = \|\mathbf{X}^{(k)} - \mathsf{H}_s(\mathbf{X}^{(k)})\|_{\infty,2} \leq \|\mathbf{X}^{(k)} - \mathbf{X}_\star\|_{\infty,2} = \|\mathbf{\Xi}^{(k)}\|_{\infty,2},$$

using that $\mathbf{X}_\star$ is $s$-row sparse. ∎

**Proof of Lemma B.10:** First, we note that, using Lemma B.3, the observation in Remark B.7 yields

$$\|\mathbf{\Xi}\|_F^2 \leq 4c_{\|\mathcal{A}\|_{2\to2}}^2 \min\left\{\frac{\sigma_{r+1}(\mathbf{X}^{(k)})}{\varepsilon_k}, \frac{\rho_{s+1}(\mathbf{X}^{(k)})}{\delta_k}\right\}^2 \tag{56}$$
$$\cdot \left\langle \mathbf{\Xi}, \left(\mathbb{P}_{\mathbf{U},\mathbf{V}}^\perp W_{\mathbf{X}^{(k)},\varepsilon_k}^{lr}\mathbb{P}_{\mathbf{U},\mathbf{V}}^\perp + \mathbb{P}_{S^c}\mathbf{W}_{\mathbf{X}^{(k)},\delta_k}^{sp}\mathbb{P}_{S^c}\right)\mathbf{\Xi}\right\rangle$$

for all $\mathbf{\Xi} \in \mathbb{R}^{n_1\times n_2}$ as the assumption (28) implies $\|\mathbf{X}^{(k)} - \mathbf{X}_\star\| \leq \min\left\{\frac{1}{48}, \frac{1}{19c}\right\}\sigma_r(\mathbf{X}_\star)$. Thus, this holds in particular also for $\mathbf{\Xi}^{(k)} = \mathbf{X}^{(k)} - \mathbf{X}_\star$. (Recall that $\mathbf{U}$ and $\mathbf{V}$ contain the leading singular vectors of $\mathbf{X}^{(k)}$, see Definition 2.1.) We estimate that

$$\sqrt{\langle \mathbb{P}_{\mathbf{U},\mathbf{V}}^\perp\mathbf{\Xi}^{(k)}, W_{\mathbf{X}^{(k)},\varepsilon_k}^{lr}\mathbb{P}_{\mathbf{U},\mathbf{V}}^\perp\mathbf{\Xi}^{(k)}\rangle} = \left\|(W_{\mathbf{X}^{(k)},\varepsilon_k}^{lr})^{1/2}(\mathbb{P}_{\mathbf{U},\mathbf{V}}^\perp(\mathbf{\Xi}^{(k)}))\right\|_F$$
$$\leq \left\|(W_{\mathbf{X}^{(k)},\varepsilon_k}^{lr})^{1/2}(\mathbb{P}_{\mathbf{U},\mathbf{V}}^\perp(\mathbf{X}_\star))\right\|_F + \left\|(W_{\mathbf{X}^{(k)},\varepsilon_k}^{lr})^{1/2}(\mathbb{P}_{\mathbf{U},\mathbf{V}}^\perp(\mathbf{X}^{(k)}))\right\|_F$$
$$\leq \left\|(W_{\mathbf{X}^{(k)},\varepsilon_k}^{lr})^{1/2}(\mathbf{X}_\star)\right\|_F + \sqrt{\sum_{i=r+1}^{\min(n_1,n_2)} \frac{\sigma_i^2}{\max\left(\frac{\sigma_i^2}{\varepsilon_k^2}, 1\right)}} \leq \left\|(W_{\mathbf{X}^{(k)},\varepsilon_k}^{lr})^{1/2}(\mathbf{X}_\star)\right\|_F + \sqrt{n-r}\,\varepsilon_k$$

Furthermore, since $\max(\varepsilon_k, \|\mathbf{\Xi}^{(k)}\|) \leq \frac{1}{48}\sigma_r(\mathbf{X}_\star)$ by assumption and $\varepsilon_k \leq \|\mathbf{\Xi}^{(k)}\|$ by Lemma C.3, we can use a variant of Lemma B.8 to obtain

$$\left\|(W_{\mathbf{X}^{(k)},\varepsilon_k}^{lr})^{1/2}(\mathbf{X}_\star)\right\|_F \leq \frac{48}{47}\left(\sqrt{r}\varepsilon_k + 2\varepsilon_k\frac{\|\mathbf{\Xi}^{(k)}\|_F}{\sigma_r(\mathbf{X}_\star)} + 2\frac{\|\mathbf{\Xi}^{(k)}\|\|\mathbf{\Xi}^{(k)}\|_F}{\sigma_r(\mathbf{X}_\star)}\right)$$
$$\leq 1.04\sqrt{r}\varepsilon_k + \frac{4.16\|\mathbf{\Xi}^{(k)}\|}{\sigma_r(\mathbf{X}_\star)}\|\mathbf{\Xi}^{(k)}\|_F.$$

On the other hand, we note that $\mathbf{\Xi}^{(k)}$ restricted to $S^c$ coincides with the restriction of $\mathbf{X}^{(k)}$ to $S^c$ under assumption (28), cf. Lemma B.2, and therefore

$$\langle \mathbb{P}_{S^c}\mathbf{\Xi}^{(k)}, \mathbf{W}_{\mathbf{X}^{(k)},\delta_k}^{sp}\mathbb{P}_{S^c}\mathbf{\Xi}^{(k)}\rangle = \langle \mathbb{P}_{S^c}\mathbf{X}^{(k)}, \mathbf{W}_{\mathbf{X}^{(k)},\delta_k}^{sp}\mathbb{P}_{S^c}\mathbf{X}^{(k)}\rangle$$
$$= \sum_{i=s+1}^{n_1} \frac{\|(\mathbf{X}^{(k)})_{i,:}\|_2^2}{\max\{\|(\mathbf{X}^{(k)})_{i,:}\|_2^2/\delta_k^2, 1\}} \leq (n_1-s)\delta_k^2.$$

With the estimate of above, this implies that

$$\left\langle \mathbf{\Xi}, \left(\mathbb{P}_{\mathbf{U},\mathbf{V}}^\perp W_{\mathbf{X}^{(k)},\varepsilon_k}^{lr}\mathbb{P}_{\mathbf{U},\mathbf{V}}^\perp + \mathbb{P}_{S^c}\mathbf{W}_{\mathbf{X}^{(k)},\delta_k}^{sp}\mathbb{P}_{S^c}\right)\mathbf{\Xi}\right\rangle$$
$$\leq \left(\left\|(W_{\mathbf{X}^{(k)},\varepsilon_k}^{lr})^{1/2}(\mathbf{X}_\star)\right\|_F + \sqrt{n-r}\varepsilon_k\right)^2 + (n_1-s)\delta_k^2$$
$$\leq \frac{13}{4}r\varepsilon_k^2 + \frac{52\|\mathbf{\Xi}^{(k)}\|^2}{\sigma_r^2(\mathbf{X}_\star)}\|\mathbf{\Xi}^{(k)}\|_F^2 + 3(n-r)\varepsilon_k^2 + (n_1-s)\delta_k^2.$$

Inserting these estimates into (56), we obtain

$$\|\mathbf{\Xi}^{(k)}\|_F^2 \leq 4c_{\|\mathcal{A}\|_{2\to2}}^2\min\left\{\frac{\sigma_{r+1}(\mathbf{X}^{(k)})}{\varepsilon_k}, \frac{\rho_{s+1}(\mathbf{X}^{(k)})}{\delta_k}\right\}^2\left(\frac{13}{4}n\varepsilon_k^2 + n_1\delta_k^2 + \frac{52\|\mathbf{\Xi}^{(k)}\|^2}{\sigma_r^2(\mathbf{X}_\star)}\|\mathbf{\Xi}^{(k)}\|_F^2\right).$$

If now either one of the two equations $\varepsilon_k = \sigma_{r+1}(\mathbf{X}^{(k)})$ or $\delta_k = \rho_{s+1}(\mathbf{X}^{(k)})$ is true, it follows that

$$\|\mathbf{\Xi}^{(k)}\|_F^2 \leq c_{\|\mathcal{A}\|_{2\to2}}^2\left(13n\varepsilon_k^2 + 4n_1\delta_k^2\right) + c_{\|\mathcal{A}\|_{2\to2}}^2\frac{208\|\mathbf{\Xi}^{(k)}\|^2}{\sigma_r^2(\mathbf{X}_\star)}\|\mathbf{\Xi}^{(k)}\|_F^2$$
$$\leq c_{\|\mathcal{A}\|_{2\to2}}^2\left(13n\varepsilon_k^2 + 4n_1\delta_k^2\right) + \frac{1}{2}\|\mathbf{\Xi}^{(k)}\|_F^2$$

if the proximity condition $\|\mathbf{\Xi}^{(k)}\| = \|\mathbf{X}^{(k)} - \mathbf{X}_\star\| \leq \frac{1}{21c_{\|\mathcal{A}\|_{2\to2}}}\sigma_r(\mathbf{X}_\star)$ is satisfied. Rearranging the latter inequality yields the conclusion of Lemma B.10. ∎

