# OpenReview forum: "Recovering Simultaneously Structured Data via Non-Convex Iteratively Reweighted Least Squares"
_NeurIPS.cc/2023/Conference — NeurIPS 2023 poster_

### Official Review · Reviewer_P5td · 2023-06-08

**Soundness:** 3 good
**Presentation:** 4 excellent
**Contribution:** 3 good
**Rating:** 6
**Confidence:** 4

**Summary:**

This paper proposes an IRLS method for recovering data with multiple, heterogeneous low-dimensional structures from linear observations. It combines non-convex surrogates for row-sparsity and rank, to identify simultaneously row-sparse and low-rank matrices from limited measurements. Theoretical results are provided that show locally quadratic convergence. The experiments demonstrate favorable empirical convergence and prove its efficacy in handling challenging data recovery scenarios.

**Strengths:**

The paper is well-written overall and the contributions are clear. The challenge of the combination of structures is made apparent. The related work discussion is thorough. The IRLS is a popular approach and is practical even for large-scale systems. Experiments suggest the method is also robust to the choice of parameters r and s.

**Weaknesses:**

Although it is also true of other results in this area, only local convergence is guaranteed and practically it may be challenging to guarantee an initialization within the required radius. The results would be strengthened if other matrices besides Gaussian were shown to satisfy this RIP, and if others were even just used empirically and shown to still offer convergence.

**Questions:**

It would be good to remind the reader what \cal{F}_ functions are in (13).

Are there other matrices besides Gaussian that are known to satisfy the RIP given in Definition 2.3?

**Limitations:**

The authors discuss the non-optimality of the bounds in Theorem 2.4, arguing that experimentally the convergence radius appears to be much larger. The authors also discuss future work about generalizing to other combined structures, which the reviewer agrees is interesting future work but certainly not a limitation to the results in this paper.

---

> ### Author Rebuttal · Authors · 2023-08-10
>
> We appreciate your constructive and very detailed feedback to our submission. A point-by-point response to your comments follows below:
>
> > Although it is also true of other results in this area, only local convergence is guaranteed and practically it may be challenging to guarantee an initialization within the required radius. The results would be strengthened if other matrices besides Gaussian were shown to satisfy this RIP, and if others were even just used empirically and shown to still offer convergence.
>
> First, we would like to mention that our supplementary material includes experiments with rank-1 and subsampled Fourier measurements, which show similar outcomes as in the Gaussian case. Extending the theory to these measurement operators would be attractive. As long as the RIP holds for such measurements our theory directly applies. Extending the RIP, e.g., from Gaussian to subgaussian measurements is straight-forward.
>
> Furthermore, in [52, ``Blind recovery of sparse signals from subsampled convolution''], Lee et al. used that, for rank one matrices, subsampled Fourier measurements of a certain shape satisfy the RIP we use (requiring additional log-factors in the sample complexity). Since extending such results to more general settings requires substantial additional effort, this is however beyond the scope of our present work.
>
> **Questions**:
> > It would be good to remind the reader what $\mathcal{F}$ functions are in (13).
>
> Thanks for this good suggestion. In the final version of the manuscript, we will be happy to remind the reader that Equation (13), which defines the quadratic models minimized in the weighted least squares steps, uses as functions $\mathcal{F}$ the $\varepsilon_k$-smoothed log-determinant function and the $\delta_k$-smoothed sum of logarithmic row-wise $\ell_2$-norm, which had been previously defined in Equation (2)-(3).
>
> > Are there other matrices besides Gaussian that are known to satisfy the RIP given in Definition 2.3?
>
> Apart from the straight-forward generalization to subgaussian measurement matrices and the subsampled Fourier measurements by [52, Lee et al.] (where the RIP is only needed for rank one matrices), we are not aware of further matrix types which provably satisfy our RIP with near-optimal rates.
>
> Please let us know if you have further questions or concerns regarding our submission. If we could clarify your questions, we would appreciate it very much if you considered an adjustment of your rating. Thank you!

---

> > ### Comment · Reviewer_P5td · 2023-08-11
> >
> > I have read the rebuttal and thank the authors for their responses.

---

### Official Review · Reviewer_3C1j · 2023-07-04

**Soundness:** 3 good
**Presentation:** 3 good
**Contribution:** 3 good
**Rating:** 7
**Confidence:** 4

**Summary:**

This article introduces an algorithm for recovering jointly row-sparse and low-rank matrices from (underdetermined) linear measurements. The algorithm is based on iteratively reweighted least squares for a non-convex objective. The method is theoretically analysed, establishing local quadratic convergence rates under the restricted isometry property. The second theoretical result relates the algorithm more directly to IRLS to show that the objective is non-increasing and proves convergence to a stationary point.

**Strengths:**

I found the paper well-written, the problem of simultaneously recovering sparse and low rank matrices is challenging one cannot simply rely on standard convex regularisation. This work provides a nice solution to this problem via IRLS and the theoretical results are interesting: the authors establish both 'compressed sensing' type results that guarantee recovery under near optimal sampling complexity, as well as results that provide an understanding of the optimisation properties of the algorithm.

**Weaknesses:**

- The numerical results are somewhat limited, it would have been nice to have a discussion on the applications of the proposed method and more realistic numerical examples.
- the result of Theorem 2.4 is restricted to the setting of exact measurements and does not cover robustness results, i.e. what if y has been corrupted with noise? Also, what if $X_*$ is only approximately sparse and low rank? Both of these are the more realistic settings, so it would have been nice to see a more complete result. Also, theory aside, equation (1) mentions additive noise, but it is not clear from the algorithm and results that the proposed method can handle additive noise.
- Given that the title mentions "simultaneously structured data", I was expecting more than just low rank + sparse. It would be interesting to see the method extended to other kind of structures such as sparsity with respect to some dictionary.


**Questions:**

- can you mention the per-iteration complexity of your method and contrast it with existing methods?
- the introduction mentions simultaneously low rank and both column and row sparse. Can your results be extended to this setting?

**Limitations:**

Yes.

---

> ### Author Rebuttal · Authors · 2023-08-10
>
> We appreciate your constructive and very detailed feedback to our submission. A point-by-point
> response to your comments follows below:
>
> > The numerical results are somewhat limited, it would have been nice to have a discussion on the applications of the proposed method and more realistic numerical examples.
>
> We agree that additional experiments on some of the real-world applications mentioned in our introduction (like sparse phase retrieval, blind deconvolution, or hyperspectral imaging) could enrich the paper. Since the focus of this work was on algorithm development and analysis, and an efficient implementation of IRLS will depend on the specific problem instance (i.e., type of measurement operator), we refrained from heading into this direction to keep the exposition concise.
>
> > The result of Theorem 2.4 is restricted to the setting of exact measurements and does not cover robustness results, i.e. what if $\mathbf{y}$ has been corrupted with noise? Also, what if $\mathbf{X}_*$ is only approximately sparse and low rank? Both of these are the more realistic settings, so it would have been nice to see a more complete result. Also, theory aside, equation (1) mentions additive noise, but it is not clear from the algorithm and results that the proposed method can handle additive noise.
>
> Certainly, the noise robustness of the proposed methods if of interest both theoretically and empirically. We restricted ourselves to setup of exact measurements in the theory as even for the case of only low-rank structures, IRLS has less developed theory in this case. However, from an empirical point of view, IRLS (without modification of the optimization problem) returns a solution whose reconstruction error is of the order of a constant times the magnitude of the additive perturbation. We add a plot exploring this behavior in the rank-$1$ case of Figure 1 of the paper in the attached PDF above.
>
> > Given that the title mentions "simultaneously structured data", I was expecting more than just low rank + sparse. It would be interesting to see the method extended to other kind of structures such as sparsity with respect to some dictionary.
>
> This is a good point. Indeed, one could add a fixed dictionary to the problem formulation which can be absorbed into the measurement operator in the analysis. The only point where we would expect some additional work to be necessary would be in proving that the concatenation of measurement operator and fixed dictionary still satisfy the RIP. For Gaussian $\mathcal{A}$ this should be possible. Also note that the paper [55, Lefkimmiatis et al.], which we mentioned in our related work discussion and which restricts itself to single structures, combines IRLS with dictionary learning techniques, i.e., it addresses the question of how to learn a dictionary on the fly.
>
> > The introduction mentions simultaneously low rank and both column and row sparse. Can your results be extended to this setting?
>
>  Thanks for mentioning this important aspect of our setting! While we restricted the presentation to the one-sided sparsity case for simplicity, it is indeed possible to adapt our IRLS method to handle sparsity on both sides of the matrix, i.e., column- and row-sparsity, by adding a second sparsity weighting term to the combined weight operator $W_{\mathbf{X}^{(k)},\varepsilon_{k},\delta_{k}}$.
>
> For instance, in the symmetric setting $\mathbf X_\star = \mathbf X_\star^T$ (naturally occurring in sparse phase retrieval) one would define the weight operator $W_{X^{(k)},\varepsilon_{k},\delta_{k}}$ as in (8), but with an additional term that multiplies $W_{X^{(k)},\delta_{k}}^{sp}$ _from the right_ to $Z$, which corresponds to minimizing the sum of _three_ smoothed logarithmic surrogates.

---

> > ### Comment · Reviewer_3C1j · 2023-08-12
> >
> > I have read the rebuttal and thank the authors for their responses.

---

### Official Review · Reviewer_W2en · 2023-07-07

**Soundness:** 3 good
**Presentation:** 3 good
**Contribution:** 3 good
**Rating:** 5
**Confidence:** 4

**Summary:**

Paper proposes to solve inverse problems on matrices subject to multiple types of sparsity (e.g. low-rank and element-wise sparsity) using an algorithm designed for the non-convex objective functions involved. The algorithm is a re-weighted least squares, which despite not solving a convex problem, authors prove a local convergence and a global consistency theorem for. They illustrate the performance of their algorithm on synthetic data.

**Strengths:**

The algorithm proposed solves a difficult non-convex problem, and the theoretical results by authors are original and interesting.

**Weaknesses:**

1. The paper contains statistical results of the type "if at least m measurements are available then the algorithm achieves good performance", but is missing a more thorough discussion on algorithmic complexity. In particular, I would be curious to know if the truncated SVD in the Update Smoothing step is the heaviest computational piece of the algorithm or if the computational bottleneck is somewhere else.
2. Numerical experiments are run on very small data. To really show the benefit of two sparsity types I would believe that bigger values of n1 n2 result in more convincing evidence that there is benefit in mixing the two prior knowledges


**Questions:**

1. what is the sensitivity of the algorithm to performing approximate SVDs
2. Figure 3: fewer iterates are needed, OK, but what's the FLOP in each iterate? what's the overall computational cost of achieving epsilon close results as compared to others? is anything parallelizable?
3. A related stream of research (see Tight Convex Relaxations for Sparse Matrix Factorization, NIPS 2014) characterizes the atoms that form the basis in which a doubly sparse matrix is sparse. How does this work, especially the lower bound Omega(r(s1+s2)), compare to those methods which directly minimize for the desired structure to recover?

**Limitations:**

because the computational complexity is not deeply discussed, it's hard to know how applicable the methods are to larger data

---

> ### Author Rebuttal · Authors · 2023-08-10
>
> We appreciate your constructive and very detailed feedback to our submission. A point-by-point
> response to your comments follows below:
>
> > The paper contains statistical results of the type ”if at least m measurements are available
> then the algorithm achieves good performance”, but is missing a more thorough discussion
> on algorithmic complexity. In particular, I would be curious to know if the truncated SVD
> in the Update Smoothing step is the heaviest computational piece of the algorithm or if the
> computational bottleneck is somewhere else.
>
> > Numerical experiments are run on very small data. To really show the benefit of two sparsity types I would believe that bigger values of $n_1$ and $n_2$ result in more convincing evidence that there is benefit in mixing the two prior knowledges.
>
> On the one hand, we agree with the reviewer that a larger scale of experiments is always preferable. On the other hand, we would like to emphasize that the theoretical results on locally quadratic convergence rates and monotonic decrease of the objective function are guaranteed to hold in higher dimensions as well. In evaluating these, the scale of the specific simulation does not matter. Finally, note that the scales of the problem instances we use are comparable to the ones used in the papers [24] and [53] describing the state-of-the-art
> methods RiemAdaIHT and SPF, respectively.
>
> **Questions**:
>
> > (1) What is the sensitivity of the algorithm to performing approximate SVDs?
>
> This is an interesting point. In fact, we did not test the present IRLS method for robustness against using approximate SVDs. However, in [47], Kümmerle et al. used a related IRLS method for low-rank recovery (i.e., a setting with only a single parsimonuous structure) in which case high-accuracy low-rank solutions were obtained using a truncated SVD implemented by a randomized block Krylov method [NIPS 2015, "Randomized Block Krylov Methods for Stronger and Faster Approximate Singular Value Decomposition", Musco et al.]. We therefore expect that this behavior extends to the simultaneously structured case considered here.
>
> > (2) Figure 3: fewer iterates are needed, OK, but what's the FLOP in each iterate? What's the overall computational cost of achieving epsilon close results as compared to others? Is anything parallelizable?
>
> We refer to our general response above for a discussion of the per-iteration computational cost.
> Regarding a potential way to parallelize the main computational steps, it would be possible to use a conjugate gradient method to solve the main linear system of the IRLS step whose complexity is dominated by matrix-vector and matrix-matrix multiplications (similar to previous work on low-rank IRLS of [47]), which parallelize well. Second, the matrix multiplications needed to obtained the truncated SVD via block Krylov could also be parallelized.
>
> > (3) A related stream of research (see Tight Convex Relaxations for Sparse Matrix Factorization, NIPS 2014) characterizes the atoms that form the basis in which a doubly sparse matrix is sparse. How does this work, especially the lower bound Omega(r(s1+s2)), compare to those methods which directly minimize for the desired structure to recover?
>
> Thanks for pointing us to this interesting work! We will be happy to include a reference to the NIPS 2014 paper by Richard et al. to the literature review in the final version. From a practical point of view, however, the paper does not propose a polynomial time algorithm for the problem, and the proposed heuristic algorithm focuses exclusively on Sparse PCA, which is a different problem setting than the underdetermined matrix recovery we consider.
>
> Therefore, it is hard to compare the numerical performance to our work right away. Regarding the theory, the estimates on the statistical dimension of the proposed (k, q)-trace norm provided by Richard et al. are restricted to (atomic) rank-1 matrices, whereas our local convergence theory for IRLS hold for arbitrary rank.
>
> Please let us know if you have further questions or concerns regarding our submission. If we could clarify your questions, we would appreciate it very much if you considered an adjustment of your rating. Thank you!

---

> > ### Author Response · Authors · 2023-08-13
> > **Addendum to rebuttal**
> >
> > Dear reviewer,
> >
> > Due to a technical issue, we could not submit a general rebuttal. The area chair kindly agreed to allow us to address your remaining question regarding the per-iteration and total time cost: We address this question in our "Addendum to rebuttal" to the review of  Reviewer bSFz. Thank you for your understanding!

---

### Official Review · Reviewer_bSFz · 2023-07-07

**Soundness:** 3 good
**Presentation:** 2 fair
**Contribution:** 2 fair
**Rating:** 5
**Confidence:** 3

**Summary:**

This work studies the problem of recovering a low-rank and row-sparse matrix from its compressed linear observations. A method based on iteratively re-weighted least squares is proposed, in which the sparsity inducing function is a non-convex log function. For theoretical contributions, this work provides a local quadratic convergence analysis of the iterates of the proposed algorithm under a restricted isometry property assumption, and it also studies the convergence of function value for any linear measurement operator using a Majorize-Minimize algorithm framework. Numerical experiments on synthetic datasets show that the proposed method can recover the ground truth with fewer measurements than some state-of-the-art methods.

**Strengths:**

1. This work provides convergence analyses not only for linear operators satisfying restricted isometry property but also for more general linear operators. Furthermore, it mentions when the restricted isometry property can be satisfied, and it also discusses the radius of the neighborhood in the local quadratic convergence.

2. The proposed algorithm is well explained by identifying that the underlying family of objectives that are minimized during the iterations is (2).

3. There are experiments for cases where the rank and sparsity are not accurately known as a prior, and thus the sensitivity to hyper-parameter choice is discussed, and also it provides a better simulation of potential real-world scenarios.

**Weaknesses:**

(1) The algorithm requires estimates of the rank and row sparsity, and the theoretical conclusions require that these estimates are accurate, i.e., $\tilde r = r$ and $\tilde s = s$. These priors and assumptions can limit the application of this method and its theoretical convergence guarantees.

(2) There are some assumptions on the iterates $\varepsilon_k$ and $\delta_k$ in Theorem 2.4 and Theorem 2.5, but it is not clear whether these assumptions will ever be satisfied. Please refer to question (1) and (3) for details.

(3) In numerical experiments, it would be better to report the time cost of different methods, including both the per iteration time cost and the total time cost.



**Questions:**

(1) In Theorem 2.4, will the assumptions $\varepsilon_k = \sigma_{r+1} (X^{(k)})$, $\delta_k = \rho_{s+1}(X^{k})$, and $\varepsilon_k \leq \sigma_r(X_\star) / 48$ ever be satisfied?

(2) The inequality at the end of page 6 only holds for certain $k$ satisfying the assumptions, so how do we arrive at the conclusion $X^{(k+l)} \to X_\star$ from such an inequality for one step?

(3) In Theorem 2.5, how strong/restricted is the assumption that $\delta_k$ and $\varepsilon_k$ have limits, and the limits are positive?

**Limitations:**

As far as I see, there is no potential negative societal impact of this work.

---

> ### Author Rebuttal · Authors · 2023-08-10
>
> We appreciate your constructive and very detailed feedback to our submission. A point-by-point
> response and clarification with regards to your comments follows below:
>
> > (1) The algorithm requires estimates of the rank and row sparsity, and the theoretical conclusions require that these estimates are accurate, i.e., $\widetilde{r} = r$ and $\widetilde{s}=s$. These priors and assumptions can limit the application of this method and its theoretical convergence guarantees.
>
> We agree that our theoretical convergence analysis is limited to this choice of hyperparameters. We, however, would like to point out that competing methods we are aware of, including the ones of papers [24] and [53], likewise need rank and row sparsity estimates. Furthermore, the experiments we performed in Fig.\ 2, in which the rank parameter was overestimated by $100\%$ and the sparsity parameter was overestimated by $50\%$, suggest that IRLS is more robust to poor tuning of the hyperparameters than other methods such as RiemAdaIHT and SPF of papers [24] and [53]. Finally, we would like to point out that the IRLS method we propose does not require any additional hyperparameters except rank and row sparsity estimates such as a step size choice, unlike, e.g., RiemAdaIHT.
>
> > (3) In numerical experiments, it would be better to report the time cost of different methods, including both the per iteration time cost and the total time cost.
>
> We agree that this would be insightful. We decided against including direct timing comparisons with other methods as a fair such comparison would require efficient implementations of all algorithms involved; since we were more interested in the _quality_ of the returned solutions, we focussed less on efficiency. We refer to our general answer above for a discussion of the per iteration cost.
>
> > (2) There are some assumptions on the iterates $\varepsilon_k$ and $\delta_k$ in Theorem 2.4 and Theorem 2.5, but it is not clear whether these assumptions will ever be satisfied. Please refer to question (1) and (3) for details.
>
> **Questions**:
>
> > (1) In Theorem 2.4, will the assumptions $\varepsilon_k = \sigma_{r+1}(\mathbf{X}^{(k)})$, $\delta_k=\rho_{s+1}(\mathbf{X}^{(k)})$, and $\varepsilon_k \leq \sigma_r(\mathbf{X}_*)$ ever be satisfied?
>
> We note that it suffices if $\varepsilon_k = \sigma_{k+1}(\mathbf{X}^{(k)})$ \emph{or} $\delta_k = \rho_{s+1}(\mathbf{X}^{(k)})$ at some step $k$, i.e., we do  not require that both of these statements are satisfied at the same time.
>
> > (2) The inequality at the end of page 6 only holds for certain $k$ satisfying the assumptions, so how do we arrive at the conclusion $\mathbf{X}^{(k+\ell)} \to \mathbf{X}_*$ from such an inequality for one step?
>
> Note that this condition is always fulfilled in the first step of Algorithm 1 since both parameters are initialized with $\infty$. Whether the condition still holds in later steps depends on the quality of initialization. The proof of Theorem 2.4 works via induction and relies on initializing sufficiently close to the ground truth. Given such an initialization, the proof shows that $\varepsilon_k = \sigma_{k+1}(\mathbf{X}^{(k)})$ or $\delta_k = \rho_{s+1}(\mathbf{X}^{(k)})$ holds for all consequent steps.
>
> Regarding the condition $\varepsilon_k \leq \sigma_{r}(X_{\star}) /48 $, we noticed that this condition is actually superfluous since the locality assumption on $\mathbf{X}^{(k)}$ in (15) automatically implies that $\varepsilon_k \leq \sigma_r(\mathbf{X}_*)/48$. Just note that
>
>  $\varepsilon_k = \min\left(\varepsilon_{k-1}, \sigma_{r+1}(\mathbf{X}^{(k)})\right) \leq \sigma_{r+1}(\mathbf{X}^{(k)})$
>         $\leq || \mathbf{X}^{(k)} - X_*|| \leq \sigma_r(X_*)/48$
>     if $\widetilde{r} = r$ as in the assumptions of Theorem 2.4. In the second inequality we used that $\mathbf{X}_*$ is of rank $r$ and in the last inequality we used (15). We will remove (14) in the revised version. Thanks for catching this!
>
> > (3) In Theorem 2.5, how strong/restricted is the assumption that $\delta_k$ and $\varepsilon_k$ have limits, and the limits are positive?
>
> Regarding Question 3, note that the sequences $\delta_k$ and $\varepsilon_k$ are, by definition, decreasing and bounded from below, i.e., by the monotone convergence theorem they converge to a limit in any case.
>
>
> Please let us know if you have further questions or concerns regarding our submission. If we
> could clarify your questions, we would appreciate it very much if you considered an adjustment of
> your rating. Thank you!

---

> > ### Author Response · Authors · 2023-08-13
> > **Addendum to rebuttal**
> >
> > Dear reviewer,
> >
> > Due to a technical issue, we could not submit a general rebuttal. The area chair kindly agreed to allow us to address your remaining question regarding the per-iteration and total time cost:
> >
> > In Appendix B of the submission, we discuss the per-iteration cost of the proposed IRLS algorithm on a basic level and mention that by applying the Sherman-Morrison-Woodbury formula, it is possible to rewrite the weighted least squares problem (18) such that the computational bottleneck is the inversion of an $O(r (n_1+n_2)) x O(r (n_1 + n_2))$, symmetric linear system.
> >
> > We note that in general, this can be done in a time complexity of $O(r^3 max(n_1,n_2)^3)$ using standard linear algebra. However, this system itself has a close-to-diagonal structure and is positive definite, so that high quality solutions could arguably found within few inner iterations (which would cost $(O(r (n_1 * n_2))$ each). If such a an implementation avenue is taken, the structure of the measurement operator might dominate the per-iteration cost of IRLS; for dense Gaussian measurements, the computation of one auxiliary matrix that is needed would cost $O(m n_1 n_2)$ flops. If rank-one or Fourier-type measurements are taken, this cost can significantly be reduced, see [24, Table 1 and Section 3] for an analogous discussion.
> >
> > We will include a more thorough discussion of the computational aspects into a final version of the manuscript, explaining also the details of our current Matlab implementation. Overall, we concede that additional work on the implementation will be needed to make the framework applicable to large-scale data such as in blind deconvolution and hyperspectral imaging problems.
> >
> > As for the total time cost of our method, we do not have a theoretical result due to the fact that we only analyze the convergence rate _close to the solution_, so the theory does not tell us how long it needs to takes to get into the local neighborhood starting from which we can quantify the complexity. However, we tried to illustrate the generic behavior of the algorithm in Figures 3 and Figures 6, which indicate that we find ourselves within a neighborhood within which quadratic convergence can be observed after only few iterations in many cases.

---

### Decision · Program_Chairs · 2023-09-21

**Decision:**

Accept (poster)

**Comment:**

The paper proposed an iteratively reweighted least squares (IRLS) algorithm to recover data matrices that are simultaneously row-sparse and low-rank, which achieves a locally quadratic convergence under minimal sample complexity, outperforming methods that use a combination of convex surrogates. The paper received unanimous support from the reviewers.